# NuA4 and H2A.Z control environmental responses and autotrophic growth in *Arabidopsis*

Tomasz Bieluszewski [1,4], Weronika Sura [1,4], Wojciech Dziegielewski [1,4], Anna Bieluszewska [1], Catherine Lachance[2], Michał Kabza[1], Maja Szymanska-Lejman [1], Mateusz Abram [1], Piotr Wlodzimierz [1], Nancy De Winne[3], Geert De Jaeger[3], Jan Sadowski[1], Jacques Côté [2✉] & Piotr A. Ziolkowski [1✉]

Nucleosomal acetyltransferase of H4 (NuA4) is an essential transcriptional coactivator in eukaryotes, but remains poorly characterized in plants. Here, we describe Arabidopsis homologs of the NuA4 scaffold proteins Enhancer of Polycomb-Like 1 (AtEPL1) and Esa1-Associated Factor 1 (AtEAF1). Loss of AtEAF1 results in inhibition of growth and chloroplast development. These effects are stronger in the *Atepl1* mutant and are further enhanced by loss of Golden2-Like (GLK) transcription factors, suggesting that NuA4 activates nuclear plastid genes alongside GLK. We demonstrate that AtEPL1 is necessary for nucleosomal acetylation of histones H4 and H2A.Z by NuA4 in vitro. These chromatin marks are diminished genome-wide in *Atepl1*, while another active chromatin mark, H3K9 acetylation (H3K9ac), is locally enhanced. Expression of many chloroplast-related genes depends on NuA4, as they are downregulated with loss of H4ac and H2A.Zac. Finally, we demonstrate that NuA4 promotes H2A.Z deposition and by doing so prevents spurious activation of stress response genes.

[1] Laboratory of Genome Biology, Institute of Molecular Biology and Biotechnology, Adam Mickiewicz University, 61-614 Poznan, Poland. [2] Laval University Cancer Research Center, CHU de Québec-UL Research Center-Oncology Division, Quebec City, QC G1R 3S3, Canada. [3] VIB-UGent Center for Plant Systems Biology, 9052 Ghent, Belgium. [4] These authors contributed equally: Tomasz Bieluszewski, Weronika Sura, Wojciech Dziegielewski.
✉email: jacques.cote@crchudequebec.ulaval.ca; pzio@amu.edu.pl

One of the characteristic features of plants is the presence of chloroplasts, which carry out photosynthesis[1]. As most chloroplast proteins are encoded by nuclear genes, chromatin should, in principle, influence their expression. Molecular and genetic studies have suggested that such a link indeed exists. For example, changes in the acetylation status of histones H3 and H4 associated with genes encoding plastid proteins coincide with changes in light and nutrient availability as well as with the developmental stage[2,3]. Likewise, increased levels of H3K9 acetylation (H3K9ac) accompany increased transcriptional activity of these genes[4]. In addition, mutations that disrupt genes involved in controlling histone modifications and chromatin remodeling, such as *General Control Non-repressed 5* (*GCN5*), *Histone Acetyltransferase of the TAFII250 Family 2* (*HAF2*; one of *Arabidopsis thaliana* homologs of TBP-ASSOCIATED FACTOR 1), *Histone Deacetylase 15* (*HDA15*) and *Brahma* (*BRM*), affect the transcript levels of genes encoding plastid proteins[5–7]. Light-dependent cell differentiation involves extensive and rapid changes in the chromatin landscape and genome topology in photosynthetic cells[8,9]. Mutants of GCN5 and other chromatin factors involved in H2B ubiquitination, H3K36 trimethylation and H2A.Z removal display defects in photomorphogenesis and adaptation to light[5,10–13]. Interestingly, the chromatin perturbations described thus far produce only mild photosynthetic phenotypes, despite their often extreme impacts on other processes[14]. Here, we show that the expression of the photosynthetic apparatus of the model plant *A. thaliana* depends on histone acetylation by a highly conserved transcriptional coactivator, the nucleosomal acetyltransferase of H4 (NuA4) complex.

Previous studies on Arabidopsis homologs of NuA4 subunits by affinity purification followed by mass spectrometry (AP-MS/MS) suggested that plant NuA4 resembles its yeast counterpart[15–18]. Importantly, the scaffold subunits Esa1-Associated Factor 1A/B (AtEAF1A/B), Transcription-Associated Protein 1 (AtTRA1) and Enhancer of Polycomb-Like 1A/B (AtEPL1A/B) were shown to be physically associated with the catalytic subunit homologs Histone Acetyltransferase of the MYST family 1/2 (HAM1/2)[16–19]. Notably, AtTRA1 was also copurified with components of complexes responsible for H2A-H2A.Z exchange, suggesting a link between NuA4-dependent histone acetylation and H2A.Z deposition[20,21]. AtEPL1 and AtEAF1 seem to be specific for NuA4, as they were not copurified with other chromatin complexes; thus, they are key to understanding NuA4 function[15,22,23]. In genetic studies, plant homologs of NuA4 subunits have been linked to gametogenesis, ABA responses, flowering time regulation, chlorophyll synthesis, cell growth, and ploidy[17,24–26]. Some of these functions of plant NuA4 have been confirmed to be mediated by histone acetylation[16,17,26,27].

In this work, we introduce a set of genetic tools that allowed us to explore the consequences of complete loss of NuA4 complex integrity without compromising viability, as occurs with the catalytic subunit mutants described in eukaryotic models[24,28,29]. Mutants lacking *AtEPL1A/B* or *AtEAF1A/B* consistently display a pleiotropic phenotype that is much more severe than the NuA4 mutant phenotypes described in plants to date. We show that the photosynthetic component of this phenotype is similar to the effects observed in the absence of Golden2-Like (GLK) transcription factors, which specifically promote chloroplast development[30]. Our genetic and transcriptomic analyses argue against the idea that NuA4 and GLKs act in the same pathway. Instead, we show that loss of NuA4 complex integrity leads to a dramatic global decrease in H4K5 and H2A.Z acetylation, with limited impact on H3K9ac. These chromatin changes strongly correlate with global transcriptomic changes. We provide direct evidence that plant NuA4 is able to efficiently acetylate both H4

and H2A.Z in nucleosomes. In addition, we show that in the absence of NuA4, H2A.Z is lost from nucleosomes across the genome. Loss of gene body H2A.Z leads to upregulation of stress genes, which normally contain large amounts of this histone variant. Transcriptional activation of these genes in the NuA4 mutants may require H3 acetylation to compensate for the lack of H4/H2A.Z acetylation in the +1 nucleosome. Finally, our results suggest a direct link between NuA4 activity and the expression of nuclear-encoded plastid genes, which depends on high levels of H2A.Z in the +1 nucleosome. Based on these findings, we propose that NuA4, by controlling both the acetylation and deposition of H2A.Z, contributes to the switch between the stress response and autotrophic growth.

## Results

**NuA4 supports chloroplast development.** The yeast NuA4 complex consists of the catalytic module piccolo NuA4 (picNuA4) and a regulatory module attached to picNuA4 through the platform subunit EAF1[31] (Fig. 1a). The EPL1 subunit enables both picNuA4 formation and its binding to the regulatory portion of the NuA4 holocomplex[23,32]. Accordingly, a yeast strain lacking the C-terminal enhancer of polycomb B (EPcB) domain of EPL1, through which it binds to EAF1, phenocopied strains lacking functional EAF1[23].

To generate a loss-of-function mutant, we disrupted both tandem copies of the *AtEAF1* genes simultaneously by using Cas9-gRNA (Fig. 1b). During mutant screening, we observed the effects of sectorial somatic double knockout of *AtEAF1*, which manifested as green/pale green mosaicism (Fig. 1c). Microscopic observation revealed a reduction in chloroplast size in the pale green sectors (Fig. 1c), suggesting that the function of AtEAF1 is cell autonomous. We observed Cas9-gRNA-induced mosaicism in plants carrying all three independent Cas9-gRNA combinations, although the construct targeting the region downstream of the Swi3, Ada2, N-Cor, and TFIIIB (SANT) domain produced a weaker effect, suggesting the minor role of this region in chloroplast development (Fig. 1c). The association between the chloroplastic phenotype and the loss of AtEAF1 was confirmed in the *Ateaf1-1* homozygous mutant, which showed even, pale green coloration (Fig. 1d). In this mutant line, the 11-kb deletion produced a chimeric *AtEAF1A/B* gene containing a frameshift upstream of the SANT domain (Fig. 1b and Supplementary Fig. 1). The SANT domain is vital to the function of yeast EAF1;[23] therefore, we considered *Ateaf1-1* to be a loss-of-function mutant.

The Arabidopsis genome encodes two EPL1 orthologs, *AtEPL1A* (At1g16690) and *AtEPL1B* (At1g79020) which have not been studied genetically. To investigate the function of AtEPL1, we generated a double T-DNA insertion mutant (hereafter *Atepl1-1*). The insertions in *Atepl1a-1* and *Atepl1b-1* are located within the EPcA domain and between the EPcA/EPcB domains, respectively (Fig. 1b). While full-length transcripts were undetectable in the double mutant, both loci still produced aberrant transcripts encoding a polypeptide homologous to the N-terminal portion of EPL1 (Fig. 1b), which is sufficient for picNuA4 activity in yeast but not for binding to the NuA4 regulatory portion[23] (Fig. 1a).

To obtain null alleles, we used Cas9-gRNA to introduce frameshift deletions in the 5' portions of the coding regions of both genes (Fig. 1b and Supplementary Fig. 2). As the double homozygous mutation was expected to be lethal, we first generated two single mutants (*Atepl1a-2* and *Atepl1b-2*) and crossed them to obtain the *Atepl1-2* double mutant. In addition, we generated an in-frame deletion spanning the EPcB domain of AtEPL1B (*Atepl1b-3*, Fig. 1b and Supplementary Fig. 2) and crossed this mutant with *Atepl1-2* to obtain the *Atepl1a-2*

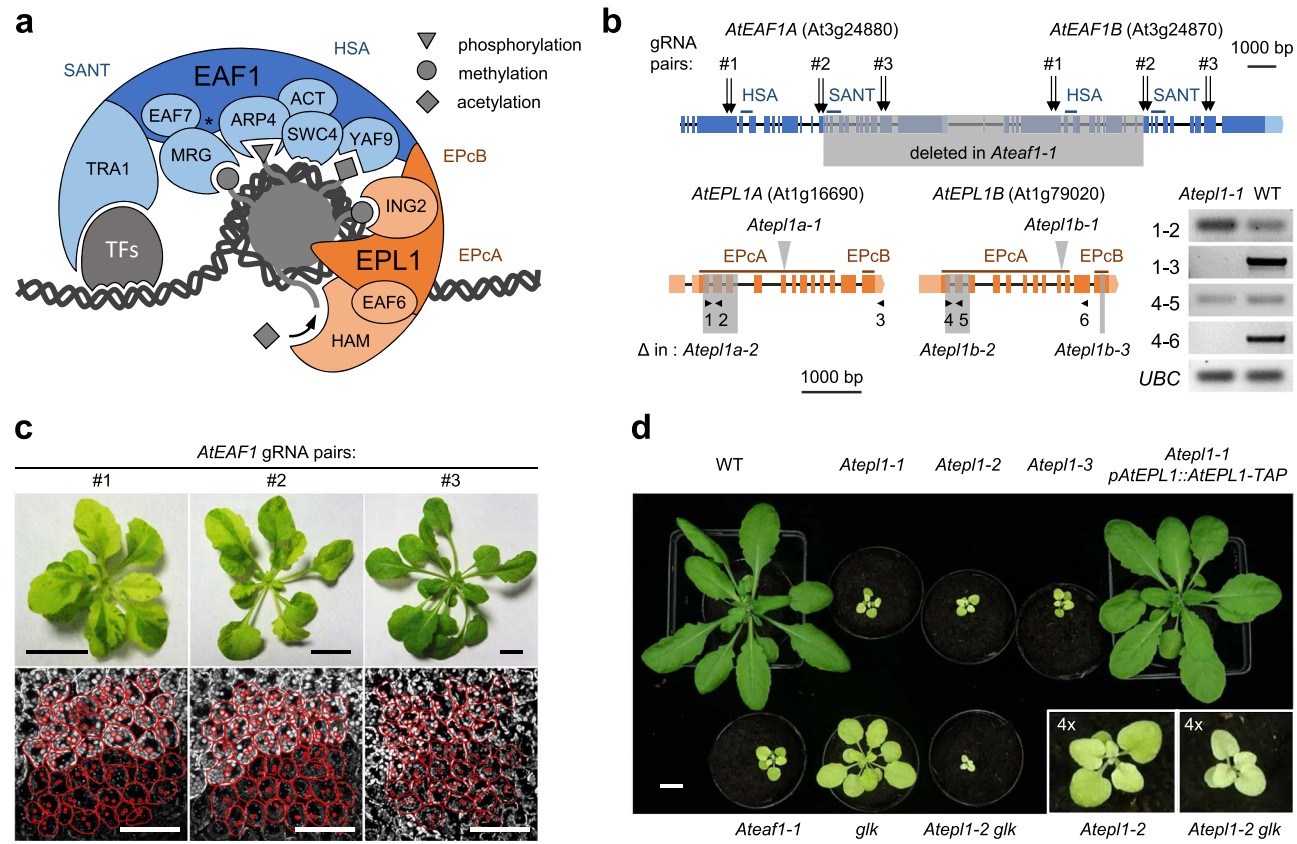

**Fig. 1 Ateaf1 and Atepl1 mutants as tools for genetic analysis of the plant NuA4 complex. a** Schematic overview of the NuA4 complex engaging chromatin. The catalytic picNuA4 module and the regulatory module are colored orange and blue, respectively. The asterisk between EAF7 and ARP4 indicates the lack of an EAF5 homolog in plants. The approximate positions of the crucial protein domains are indicated by the labels outside the shapes representing the subunits. **b** Gene structure diagrams of the AtEAF1 and AtEPL1 loci. The target sites for the three double gRNA constructs (gRNA pairs) used for AtEAF1 mutagenesis are denoted by arrows. The shaded areas delimit Cas9-gRNA-induced deletions (Δ) characterizing particular alleles. The gray arrowheads indicate T-DNA insertion sites. The small black arrowheads indicate the sites of primers used for the RT–PCR analysis presented in the panel on the right. **c** Transgenic plants carrying the three different Cas9-gRNA constructs. The scale bar represents 10 mm in all images. The lower panels show confocal images of chlorophyll autofluorescence in the palisade mesophyll cell layer of a leaf, centered on the border regions between green (upper part) and pale green (lower part) sectors. The red outlines delimit selected cells and chloroplasts. The scale bar represents 50 µm in all micrographs. **d** Four-week-old WT and mutant plants grown under the same conditions. The insets show the same images at ×4 magnification. The scale bar represents 10 mm.

Atepl1b-3 mutant (hereafter Atepl1-3). Since the genomic deletion present at the AtEPL1A locus in the Atepl1-2 mutant ends in an intron, the exact transcript sequence cannot be predicted with confidence. To check whether the splicing of the Atepl1a-2 allele leads to a frameshift, we sequenced 6 independent cDNA clones derived from a homozygous Atepl1-2 mutant. All clones displayed an identical splicing pattern of the truncated intron IV, which led to a frameshift; therefore, this allele is unlikely to produce considerable amounts of an even partially functional AtEPL1A protein (Supplementary Fig. 2). Importantly, the strong pleiotropic Ateaf1-like phenotype shared by the Atepl1 homozygotes was fully rescued in Atepl1-1 homozygotes expressing tagged AtEPL1B (pAtEPL1B::AtEPL1B-TAP, Fig. 1d).

**AtEPL1 is not essential for viability but is key to NuA4 function.** Unlike in yeast and animals, where EPL1 is an essential gene[33,34], all double Atepl1 mutants described above were viable, although they were completely sterile and clearly exhibited more defects than other plant NuA4 subunit mutants characterized to date (Supplementary Fig. 3). The viability of the Atepl1 mutants gave us an opportunity to compare the relative contributions made by the EAF1-dependent NuA4 holocomplex and EPL1-dependent picNuA4 in plants. Additionally, we included in our analyses the well-characterized glk1-1 glk2-1 mutant (hereafter

glk), in which a cell-autonomous reduction in chloroplast size and a significant reduction in rosette diameter have previously been observed[30,35].

We measured the chloroplast and cell sizes and found them to be significantly smaller in all mutants than in wild-type (WT) plants (Fig. 2a and Supplementary Table 1). The NuA4 mutants indeed showed similarity to the glk mutant in terms of chloroplast size but were less similar in terms of cell size (Fig. 2a). Notably, the Atepl1-2 null mutant displayed a greater reduction in chloroplast size than the Ateaf1-1 mutant, possibly owing to residual nontargeted activity of picNuA4 in this line. It remains unclear why the Atepl1-3 (picNuA4-separated) mutant is more phenotypically similar to Atepl1-2 than to Ateaf1-1.

The pale green coloration of the NuA4 and glk mutants indicates loss of chlorophyll. Indeed, spectrophotometric analysis showed a significant reduction in total chlorophyll content in all mutants, with the strongest effect in glk, followed by Atepl1-2. The Ateaf1-1 mutant again displayed the weakest effect (Fig. 2b and Supplementary Table 3). Interestingly, the significant increase in the chlorophyll a/b ratio reported previously in the glk mutant[36] was not observed in the NuA4 mutants (Fig. 2b and Supplementary Table 4).

As histone acetylation is known to be important in photomorphogenesis, we characterized the light adaptation phenotype

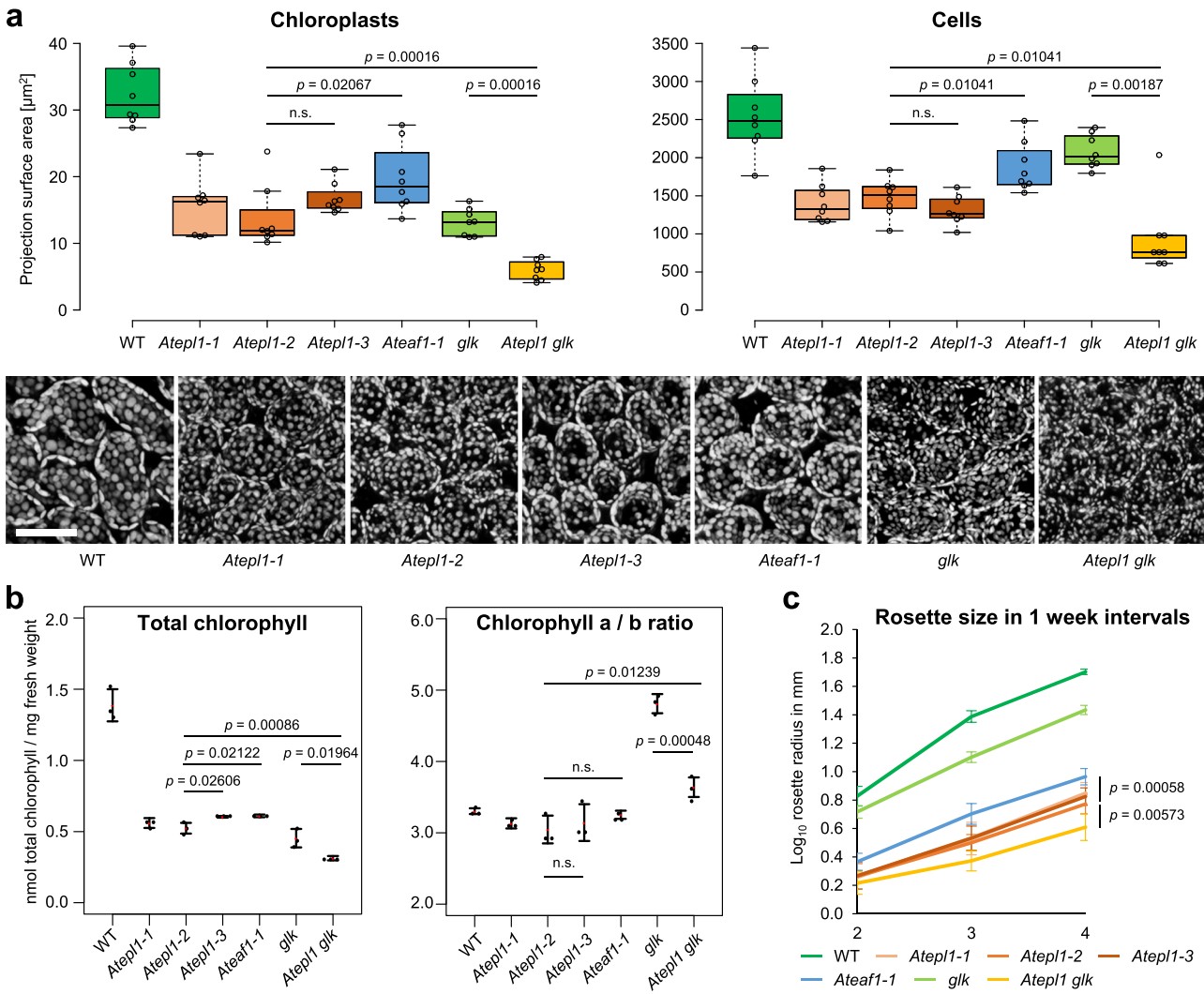

**Fig. 2 Growth- and chloroplast development-related phenotypes of the NuA4 mutants are enhanced by *glk* mutations. a** Quantification of the chloroplast and cell size data acquired by microscopic imaging of the palisade mesophyll cell layer. The center line of a boxplot indicates the median; the upper and lower bounds indicate the 75th and 25th percentiles, respectively; and the whiskers indicate the minimum and maximum. Each circle represents a median surface area of 20 chloroplasts or 9 cells measured in a single biological replicate (a single confocal stack from a separate plant). Statistical significance was determined by a two-tailed Mann–Whitney *U* test. The chlorophyll autofluorescence images below the plots are projections of example stacks used for the measurements. All of the images are on the same scale. The scale bar represents 50 μm. **b** Determination of the total chlorophyll content and the chlorophyll a/b ratio by spectroscopic analysis of extracted chlorophyll. The dots indicate biologically independent replicates (three per each genotype), while the red dots indicate means. The whiskers delimit 95% confidence intervals. Statistical significance was determined by a two-tailed *t* test. **c** Growth quantification of the mutants. The nodes indicate the mean rosette radius of 10 or 20 (*Atepl1-2*) plants grown in parallel. The whiskers delimit 95% confidence intervals. *p* values are given for the *Ateaf1-1* vs. *Atepl1-2* (upper) and *Atepl1-2* vs. *Atepl1 glk* (lower) comparisons. Statistical significance was determined by a two-tailed *t* test. Source data are provided as a Source Data file.

of the *Atepl1-2* mutant. We measured hypocotyl length, apical curvature, and cotyledon development (surface area and angle) at different time points following a dark-to-light transition. *Atepl1-2* showed a reduced hypocotyl length and cotyledon surface area (Supplementary Fig. 4). This phenotype was in contrast to other histone acetyltransferase mutants, which showed elongated hypocotyls when grown in light or enhancement of the long-hypocotyl phenotype of light receptor mutants[5,37]. However, this result must be considered with caution, as the severe growth phenotypes of NuA4 mutants can affect hypocotyl and cotyledon development.

To characterize the vegetative growth phenotype of the mutants, we measured the rosette radius of plants after 2, 3, and 4 weeks of growth under long-day conditions. In this assay, the *glk* mutant performed much better than the NuA4 mutants,

suggesting that the dwarfism of *Atepl1* and *Ateaf1* cannot be fully explained by chloroplast dysfunction (Fig. 2c and Supplementary Tables 5–7). Compared with the WT plants, all lines that contained mutations in NuA4 subunits showed a severe reduction in size at all time points (Fig. 2c and Supplementary Tables 5–7). Consistent with its relatively weak effect on cell size, the *Ateaf1-1* mutation had a significantly smaller effect on growth than the *Atepl1-2* mutation (Fig. 2c and Supplementary Tables 1–7). Collectively, these results support a model in which AtEPL1 cooperates with AtEAF1 for its functionality as a picNuA4 subunit.

**NuA4 and GLKs play parallel roles in transcription.** Yeast NuA4 requires transcription factors for improved targeting to

specific genomic loci[38,39]. We hypothesized that the phenotypic similarity between NuA4 subunit and GLK transcription factor mutants was due to the targeting of NuA4 by GLKs. To verify our hypothesis, we generated a quadruple *Atepl1a-2 Atepl1b-2 glk1-1 glk2-1* mutant (hereafter *Atepl1 glk*) by crossing the *Atepl1-2* sesquimutant with the previously characterized *glk1-1 glk2-1* homozygous mutant[30]. The quadruple mutant was viable but displayed significantly exacerbated phenotypic effects in most phenotypic assays (Figs. 1–2 and Supplementary Tables 1–7). Notably, the increase in the chlorophyll a/b ratio observed in the *glk* mutant but not in the NuA4 mutants was significantly diminished in the *Atepl1 glk* mutant (Fig. 2b and Supplementary Table 4). We also examined the expression of six selected photosynthesis-related genes in the quadruple mutant and found that their expression levels were generally lower than they were in any of the double mutants (Supplementary Fig. 5). These results suggest that NuA4 and GLK have mostly independent functions in the transcription of genes related to growth and photosynthesis.

Seeking an explanation for the mutant phenotypes and genetic interactions described above, we performed transcriptome analysis of *Atepl1-2*, *Ateaf1-1*, *glk* and WT plants. Overall, 4022, 3474, and 1235 genes showed at least a 1.5-fold decrease (downregulated genes) or increase (upregulated genes) in the steady-state transcript level in the *Atepl1-2*, *Ateaf1-1*, and *glk* mutants, respectively (Supplementary Data 1–6). For *Atepl1-2* and *Ateaf1-1*, Gene ontology analysis of the downregulated gene groups (1945 and 1771 uniquely mapped IDs, respectively) revealed significant enrichment with genes associated with chloroplasts, photosynthesis and growth (Fig. 3a and Supplementary Data 7–9). The upregulated gene groups (2077 and 1703 IDs for *Atepl1-2* and *Ateaf1-1*, respectively) were significantly enriched in genes functionally related to responses to biotic and abiotic stimuli (Fig. 3a, b and Supplementary Data 10–12). Biotic stress marker genes were also induced in *Atepl1-1* seedlings, which rationally rules out the possibility of their activation by deteriorating external conditions over extended growth periods (Supplementary Fig. 6). Of the genes downregulated and upregulated in *Ateaf1-1*, 72.7% and 77.6% were also downregulated and upregulated in *Atepl1-2*, respectively (Fig. 3b). In contrast to the differential expression pattern in the NuA4 mutants, the genes that were differentially expressed in *glk* relative to WT control plants were predominantly downregulated (807 vs. 428 IDs, Fig. 3b). The overlap between the genes with changed expression in *Atepl1-2* and *glk* was relatively small: only 31.7% and 46.7% of the genes down- and upregulated, respectively, in *glk* were also adequately deregulated in *Atepl1-2*. Surprisingly, the genes that were downregulated in *Atepl1-2* and *Ateaf1-2* showed a higher association with chloroplasts and photosynthesis than the genes that were downregulated in *glk* (Fig. 3b and Supplementary Fig. 7). Upon closer examination of four manually curated groups of functionally related genes, we found that both NuA4 and GLKs contribute to the expression of nuclear genes encoding proteins involved in chlorophyll biosynthesis and light harvesting, whereas the expression of nuclear plastid ribosomal protein genes (npRPGs) depends only on NuA4 (Fig. 3c). Overall, these observations are in agreement with the results of our genetic analyses, suggesting largely independent roles for NuA4 and GLKs in chloroplast development and maintenance.

**AtEPL1 is a subunit of the plant NuA4 catalytic module**. To further explore the molecular basis of the observed phenotypes of plant NuA4 mutants, we first had to test whether the physical interactions of AtEPL1 and AtEAF1 follow the predictions based

on sequence homology to yeast proteins. In yeast, EPL1 interacts directly with the catalytic subunit of the complex, which in Arabidopsis is represented by HAM1 and HAM2[40]. We produced recombinant GST-AtEPL1A, GST-AtEPL1B, His-HAM1, and His-HAM2 in bacteria and performed a pull-down assay. Both AtEPL1 were specifically pulled down by both HAM, as no GST signal was detected in the eluate either after His-GFP was used as bait or when no bait was used (Fig. 4a). We confirmed all interactions detected by pulldown using far western blotting (FWB), which indicated that AtEPL1A and AtEPL1B interact directly and interchangeably with HAM1 and HAM2 (Fig. 4b).

To determine whether Arabidopsis picNuA4 is capable of nucleosomal histone acetylation, we coexpressed HAM1, AtEPL1A, ING2, and AtEAF6 and used the recombinant complex for an in vitro histone acetylation (HAT) assay (Fig. 4c–g). Notably, the use of HAM1, AtEPL1A, or ING2 as bait for complex purification resulted in the recovery of all recombinant picNuA4 subunits, which confirmed that these four plant proteins interact physically to form a complex homologous to yeast picNuA4 (Fig. 4c and Supplementary Fig. 8a). The in vitro HAT assay confirmed the activity of Arabidopsis picNuA4 on native short oligonucleosomes (SONs). As expected, tritium-labeled acetyl groups were more efficiently transferred to histones H2A and H4, showing a significant preference for the latter, seemingly more in the Arabidopsis complex than in the yeast complex (Fig. 4e). When the complex was isolated based on ING2, HAM1 was recovered from both AtEPL1A-expressing bacterial strains and samples lacking this protein (Fig. 4d and Supplementary Fig. 8a). However, HAT assays on the obtained isolates showed that the presence of AtEPL1A is essential for the activity of the complex on chromatin (SONs) and that the complex shows significantly reduced activity on free core histones (CHs) (Fig. 4e and Supplementary Fig. 8b).

Next, we checked whether plant picNuA4 is also capable of performing H2A.Z acetylation. As mammalian H2A.Z differs from its Arabidopsis counterpart in the N-terminal tail, which is the target for acetylation (Supplementary Fig. 9), we generated a stable K562 cell line expressing the major Arabidopsis H2A.Z gene, HTA9, tagged with 3×FLAG, from the *AAVS1* locus[41]. Fractionation of chromatin followed by affinity purification with an anti-FLAG antibody proved that Arabidopsis H2A.Z is effectively incorporated into nucleosomes in the mammalian system (Fig. 4f). HAT assays showed that Arabidopsis picNuA4 can efficiently target H2A.Z in chromatin and requires AtEPL1 for this acetyltransferase activity (Fig. 4g). All results of the HAT assays were consistent with observations made in yeast[42,43] and provide strong evidence for the evolutionary conservation of the subunit composition and the activity of the picNuA4 complex between budding yeast and plants.

**AtEPL1 interacts with the regulatory module of NuA4**. PicNuA4 interacts with the targeting module of yeast NuA4 through contact between the EPcB domain of EPL1 and EAF1. Our phenotypic analyses of the *Atepl1-3* and *Ateaf1-1* mutants suggested that AtEAF1 and the EPcB domain of AtEPL1 are crucial elements of the plant NuA4 complex. To obtain biochemical confirmation of this observation, we performed tandem AP-MS/MS, in which we used the *Atepl1-1* mutant complemented with the *pAtEPL1B::AtEPL1B-TAP* construct. The experiment was performed twice on hydroponically grown seedlings. Notably, among the 15 proteins that were specifically copurified with AtEPL1B in both experiments, 12 were homologs of yeast NuA4 subunits, including EAF1 and ESA1 (Table 1 and Supplementary Data 13–16). This result confirms and improves on previously published findings regarding potential interacting

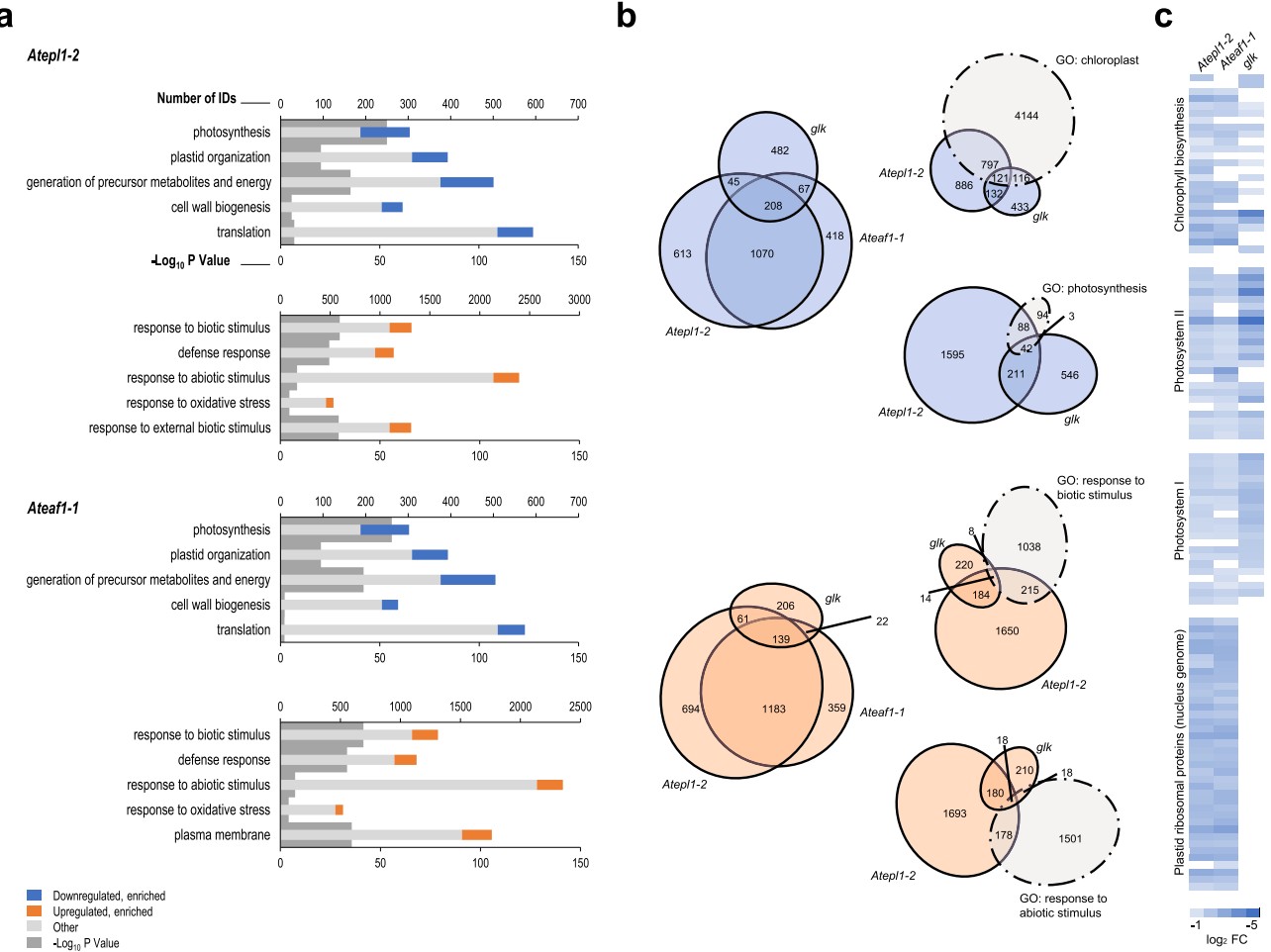

**Fig. 3 NuA4 influences the expression of chloroplast-related and stress-responsive genes. a** Gene ontology enrichment analysis of the down- and upregulated genes in *Atepl1-2* and *Ateaf1-1*. The five representative GO categories displayed in each graph were selected manually from the lists presented in Supplementary Data 8 and 11. Gene enrichment analyses were performed using the PANTHER overrepresentation test. **b** Area-proportional Venn diagrams illustrating the overlap between genes exhibiting an expression change in the same direction in the *Atepl1-2*, *Ateaf1-1* and *glk1* mutants as well as the gene sets representing particular GO categories. All numeric values indicate the numbers of genes present in a single sector of the diagram rather than the total values for the sets. **c** Relative transcript levels (log2 FC) of the genes encoding four selected groups of chloroplast proteins in *Atepl1-2*, *Ateaf1-1*, and *glk*. Empty fields indicate a lack of significant change. Source data are provided as a Source Data file.

proteins of AtEPL1B[18] and other putative subunits of *A. thaliana* NuA4[16–18,44] (Table 1). In particular, tandem AP-MS/MS resulted in a low total number of interacting proteins but a high number of NuA4 subunit homologs, indicating high specificity (Table 1 and Supplementary Data 13–14). In fact, our analysis identified only 5 proteins that showed no homology to yeast NuA4 subunits as potential AtEPL1B interacting proteins. Two of these proteins, AOG2 and BRD6, were previously copurified with *A. thaliana* NuA4 subunits, albeit within much larger pools of additional interacting proteins (Table 1)[16,18,45]. Interestingly, the uncharacterized bromodomain protein BRD6[40], which was previously copurified with SWC4[16] and HAM1[18] (Table 1), is orthologous to BRD8, a subunit of the human homolog of NuA4[46], according to the PANTHER classification system[47,48], indicating a possible role of BRD6 as an accessory subunit of plant NuA4.

To test whether AtEPL1 interacts with the targeting region of NuA4 through direct binding to AtEAF1, we performed a pulldown assay in which a His-tagged fragment of AtEAF1B corresponding to AAs 430-1322, His-HAM1 (positive control), or His-EGFP (negative control) was used as bait and GST-AtEPL1B was used as prey. Only His-AtEAF1B and His-HAM1 were able

to pull-down GST-EPL1B, which confirmed the direct physical interaction between AtEPL1B and AtEAF1B (Fig. 4h). These results support our homology-based model in which AtEPL1 and AtEAF1 interact directly to provide a scaffold for the assembly of the plant NuA4 complex.

**NuA4 acetylates H4 and H2A.Z and stimulates H2A.Z deposition.** To determine whether *Atepl1-2* plants display decreased acetylation of H4 (H4K5ac, H4K8ac, H4K12ac, and H4K16ac) and H2A.Z in vivo, we performed ChIP-qPCR on selected genes whose expression was decreased in *Atepl1-2* (Supplementary Fig. 10). As expected, all tested forms of acetylation were significantly reduced in the mutant. Among NuA4-acetylated H4 residues, H4K5ac was the most enriched.

In yeast, H2A.Z incorporation into nucleosomes can be stimulated by NuA4-dependent H4 acetylation[49]. Therefore, we sought to determine whether the lack of functional AtEPL1 results in H2A.Z loss from chromatin, which could affect gene expression. To this end, we checked the levels of HTA9, the major isoform of H2A.Z in Arabidopsis, by ChIP-qPCR at several loci (Supplementary Fig. 10). We found that HTA9 levels were significantly reduced in the *Atepl1-2* mutant (Supplementary

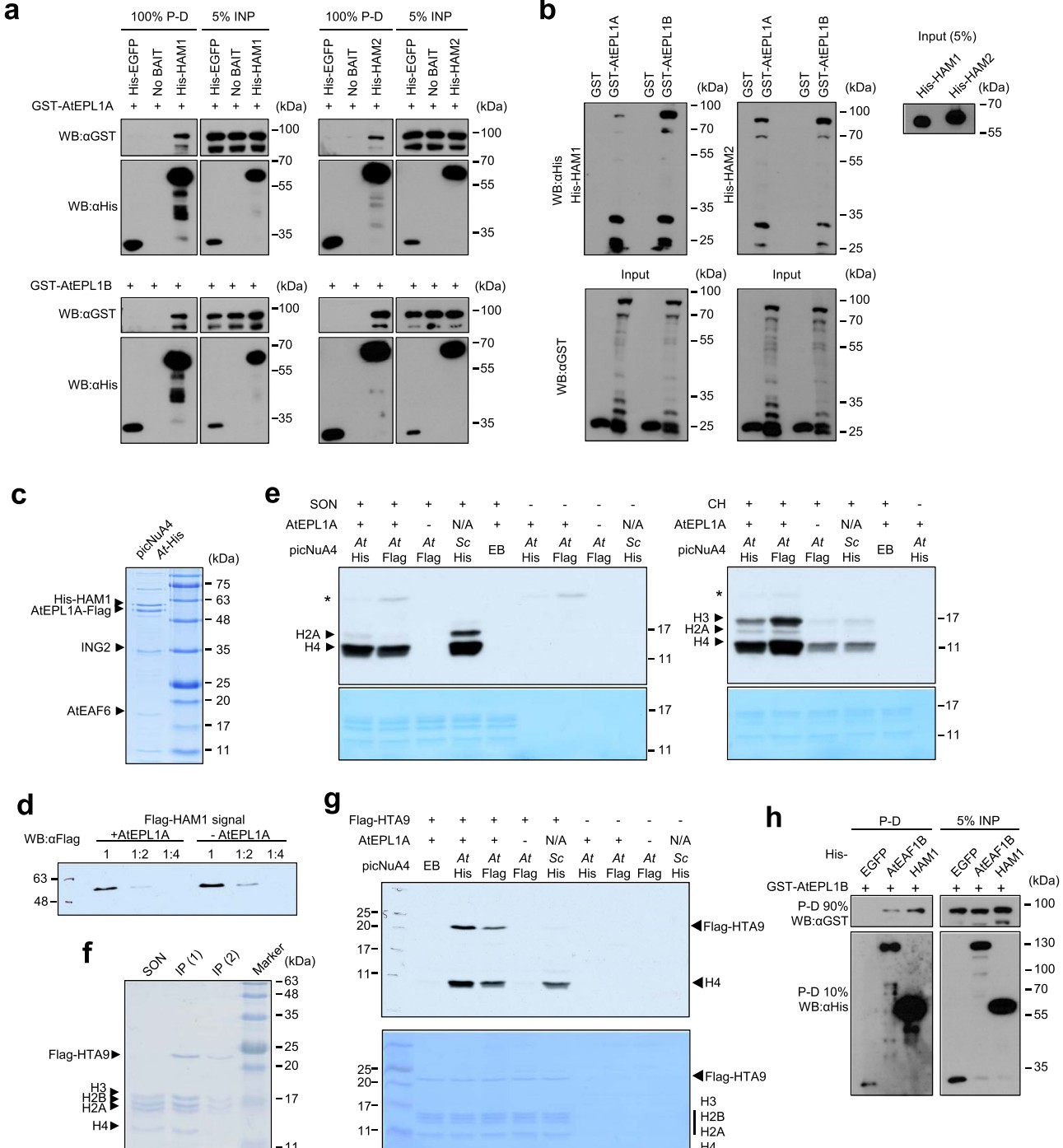

Fig. 10). We also crossed *Atepl1* with mutants of two genes directly involved in H2A.Z deposition, *ARP6* and *MBD9*[20,21,50,51]; however, both triple mutations (*Atepl1-2 arp6* and *Atepl1-2 mbd9*) were synthetically lethal (Supplementary Table 8).

To obtain insight into global chromatin changes, we performed chromatin immunoprecipitation followed by DNA sequencing (ChIP-seq) of *Atepl1-2* and WT plants grown under the same conditions as the plants used for RNA-seq. We profiled two modifications that we showed are catalyzed by plant NuA4 (H4K5ac and H2A.Zac) and one activating modification catalyzed primarily by other HATs (H3K9ac). To profile H2A.Zac, we used an antibody raised against human H2A.Zac, which was shown to recognize also Arabidopsis H2A.Zac (Crevillén et al.[17]; Supplementary Fig. 8C). In

addition, we profiled both total H2A.Z to better understand the relationship between its acetylation and chromatin deposition and unmodified H3 to observe potential alterations in nucleosome occupancy. Due to expected global changes in epitope abundance, we included a predetermined amount of mouse chromatin in our ChIP experiments as a reference (spike-in).

In the *Atepl1-2* mutant, the total H2A.Z, H4K5ac, and H2A.Zac levels were dramatically reduced (Fig. 5a). In contrast, the profile of H3K9ac in *Atepl1-2* overlapped well with that in WT plants, with only a slight increase within the peak downstream of the TSS (Fig. 5a). Interestingly, the average H3 occupancy in *Atepl1-2* showed an increase on both sides of the TSS (Fig. 5a).

**Fig. 4 AtEPL1 interacts directly with the catalytic and platform subunits of Arabidopsis NuA4 and supports nucleosomal H4 and H2A.Z acetylation in vitro. a** Pulldown assay showing the interaction between the recombinant AtEPL1 and HAM paralog pairs expressed in bacteria. Pull-down and input are abbreviated P-D and INP, respectively. **b** Interactions between AtEPL1s and HAMs were analyzed by far western blotting. After SDS–PAGE and western blotting of free GST and GST-AtEPL1A/B, the upper membranes were sequentially incubated with His-HAM1/2 and anti-His antibodies, whereas the lower membranes were incubated with an anti-GST antibody. The lower bands in the upper panels most likely indicate the interaction of HAM1/2 with partially degraded AtEPL1B. **c** Purification of recombinant *A. thaliana* picNuA4 from bacteria (Coomassie staining). A 6×His tag fused to HAM1 was used for purification on cobalt affinity resin. The numbers in the brackets indicate the expected molecular weights of the recombinant picNuA4 subunits. **d** Western blot showing Flag-HAM1 normalized protein levels in purified fractions from bacteria producing At picNuA4 complexes with and without AtEPL1A. Both fractions retained the Flag-HAM1 signal after ING2-His purification, indicating that AtEPL1A is not required for HAM1 binding. **e** In vitro assay of the HAT activity of recombinant *A. thaliana* (*At*) picNuA4 toward short oligonucleosomes (SONs) and free core histones (CHs). Recombinant *S. cerevisiae* (*Sc*) picNuA4 was used as a positive control. *At*-His corresponds to the purified His-HAM1 complex, and *At*-Flag indicates two versions of the Flag-HAM1-purified complex (see Supplementary Fig. 8a), with and without AtEPL1A, as shown in **d**. Autoacetylation on AtEAF6 is highlighted with an asterisk. **f** Immunoprecipitation (IP) of Flag-conjugated Arabidopsis H2A.Z protein (Flag-HTA9) from chromatin extract resulted in the isolation of human nucleosomes containing Arabidopsis H2A.Z. IP (1) and IP (2) correspond to Flag peptide eluates 1 and 2, respectively. SONs were used as a control. **g** In vitro assay of the HAT activity of recombinant *A. thaliana* picNuA4 toward Arabidopsis H2A.Z histone (FLAG-HTA9)-containing oligonucleosomes. Three versions of At picNuA4 were used, similar to the procedure in **f**. Only complexes containing AtEPL1A efficiently acetylated nucleosomal HTA9. *Sc* picNuA4 was used as a control with no activity towards HTA9. **h** Pull-down assay showing the interaction between AtEPL1B and a fragment of AtEAF1B corresponding to AAs 430-1322. HAM1 was used for comparison. All proteins were expressed in bacteria as either GST or 6×His fusions. Source data are provided as a Source Data file.

As the H2A.Z histone variant and the three acetyl marks were highly enriched downstream of the TSS in WT plants (Fig. 5a), we averaged the signal over the first 500 bp of each transcribed region and used these values to perform comparisons between individual genes. Total H2A.Z showed a global reduction in this window, although it was uneven, likely because of extensive differences among genes in the enrichment of this histone variant (Fig. 5b). Consequently, H2A.Zac also showed an extensive and uneven reduction for most genes (Fig. 5b). These results indicate that NuA4 is critical for H2A.Z deposition onto chromatin. While H4K5ac in *Atepl1-2* showed a strong and almost uniform reduction in this window, H3K9ac was preserved in most genes, with few genes showing large changes (Fig. 5b).

Using the same data, we checked the interrelationships between individual histone modifications (Fig. 5d). All three histone acetylations correlate well with each other in WT, with the strongest correlation observed between H2A.Zac and H4K5ac, which is consistent with their dependence on NuA4. On the other hand, we did not observe any significant correlation between total H2A.Z enrichment and histone acetylations (including H2A.Zac). In *Atepl1-2*, the correlations between histone acetylations generally weakened (Fig. 5e).

**Loss of H4ac/H2A.Zac and gain of H3ac correlate with the *Atepl1-2* transcriptome changes.** Given the general role of histone acetylation in transcriptional activity, we hypothesized that genes dysregulated in *Atepl1-2* should be especially noticeable in terms of their acetylation distribution, enrichment or change in the mutant. H4K5ac, H2A.Zac, and H3K9ac showed similar average genic distributions in both the WT and the *Atepl1-2* mutant, peaking at the +1 nucleosome, according to our ChIP-seq data. The average levels of all three forms of acetylation in WT plants were lower in dysregulated genes (log₂ fold change < −1.5 or log₂ > 1.5) than in all expressed genes (Fig. 6a). Notably, in the mutant background, the H3K9ac level was significantly increased in upregulated genes, but slightly decreased in downregulated genes (Fig. 6a and Supplementary Fig. 11). In addition, all expressed genes exhibited a moderate increase in H3K9ac in the mutant background. This pattern indicates that in the context of decreased acetylation of H4 and H2A.Z, increased H3 acetylation may maintain or even increase the expression of some genes.

In parallel, we observed that all three gene groups (all expressed, downregulated and upregulated) displayed different H4K5ac and H2A.Zac levels in WT plants, with upregulated genes showing the lowest enrichment of the two modifications. In the *Atepl1-2* mutant background, all gene groups lost both acetylation marks, but the decreases were greatest in the downregulated genes (Fig. 6 and Supplementary Fig. 11). This finding suggests that in the absence of NuA4, the levels of nucleosomal H4K5ac and H2A.Zac drop below the threshold levels in certain genes, decreasing their transcriptional activity.

We then further investigated how gene expression changes in the *Atepl1-2* mutant corresponded to changes in H3, H4K5ac, H2A.Zac, total H2A.Z and H3K9ac enrichment. To this end, genes dysregulated in *Atepl1-2* (*n* = 5135) were sorted based on their fold change (FC) and divided into percentiles. For genes in each percentile, the averaged spike-in normalized signal over the first 500 bp of each transcribed region was used as histone modification enrichment measure. We observed an inverse correlation between histone acetylation and expression FC (Fig. 6b). Interestingly, this relationship was present when the histone modification enrichment was assessed both in WT and in the *Atepl1-2* background, and did not depend on the direction of the expression change (Fig. 6b and Supplementary Fig. 12). Based on these findings, we concluded that genes dysregulated in *Atepl1-2* are characterized by low basal level of histone acetylation in the +1 nucleosome of WT plants.

On the other hand, the level of total H2A.Z showed no significant correlation with the expression change. As H2A.Z can have both activating and repressing effects on gene transcriptional activity[52], we decided to use our data for total- and acetylated-H2A.Z to estimate the amount of non-acetylated H2A.Z in gene percentiles (Fig. 6b and Supplementary Fig. 12). This showed that genes dysregulated in *Atepl1-2* are enriched for non-acetylated H2A.Z in the +1 nucleosome.

Based on the average values of chromatin modifications per each percentile, we calculated the difference between the levels of H3, H3K9ac, H4K5ac and both acetylated and unacetylated H2A.Z in WT and the *Atepl1-2* mutant (Fig. 6c). *Atepl1* − WT signal differences for H3 occupancy showed negative correlation, while all three histone acetylations showed strong positive correlations with gene expression change: For H2A.Zac and H4K5ac, the decrease in modification level was greatest for downregulated genes and smallest for upregulated genes (Fig. 6c). In the case of H3K9ac, we observed an increase in signal strength in the mutant for most percentiles (positive Y-axis values in

**Table 1 TAP-MS data for AtEPL1B combined with metaanalysis of the previously published AP-MS experiments using NuA4 subunits as baits.**

| | Sc name | At name | At gene | AtARP4 | AtARP4 | AtSWC4 | HAM1 | HAM2 | AtEPL1A | AtEPL1B | AtYAF9A | AtEPL1B |
|---|---|---|---|---|---|---|---|---|---|---|---|---|
| Complexes | - | - | - | NuA4, SWRI, SNF, INO80 | NuA4, SWRI, SNF, INO80 | NuA4, SWRI | NuA4, PEAT | NuA4, PEAT | NuA4 | NuA4 | NuA4, SWRI | NuA4 |
| Model | | - | - | Cell culture | Cell culture | Cell culture | Seedlings | Seedlings | Seedlings | Seedlings | | **Seedlings** |
| Promoter | | - | - | CaMV35S | UBQ10 | UBQ10 | Native | Native | Native | Native | CaMV35S | **Native** |
| Tag | | - | - | GS TAP tag | Strep | Strep | Myc | Myc | Flag | Flag | GH TAP tag | **GS TAP tag** |
| Reference | | - | - | Vercruyssen | Bieluszewski | Bieluszewski | Tan | Tan | Tan | Tan | Crevillen | **This work** |
| Conserved NuA4 subunits | **ARP4** | AtARP4 | AT1G18450 | + | + | + | + | + | + | + | + | **2/1.78** |
| | | AtARP4A | AT1G73910 | - | + | + | + | + | + | + | - | **2/1.63** |
| | **SWC4** | AtSWC4 | AT2G47210 | + | + | + | + | + | + | + | + | **2/1.42** |
| | **YAF9** | AtYAF9A | AT5G45600 | + | + | + | + | + | + | + | + | - |
| | | AtYAF9B | AT2G18000 | - | - | + | - | - | - | - | - | **2/0.42** |
| | **EAF1** | AtEAF1A | AT3G24880 | - | + | + | + | + | + | + | + | **2/0.43** |
| | | AtEAF1B | AT3G24870 | - | + | + | + | - | - | + | - | **2/0.37** |
| | **TRA1** | AtTRA1A | AT2G17930 | - | - | - | - | - | - | - | - | **2/0.44** |
| | | AtTRA1B | AT4G36080 | - | - | - | - | - | - | - | - | **2/0.44** |
| | **ESA1** | HAM1 | AT5G64610 | - | + | + | + | + | + | + | + | **2/0.95** |
| | | HAM2 | AT5G09740 | - | - | - | - | + | - | - | - | 1/0.16 |
| | **EPL1** | AtEPL1A | AT1G16690 | - | - | + | + | + | + | + | + | **2/2.22** |
| | | AtEPL1B | AT1G79020 | - | + | + | + | - | + | + | - | **2/1.50** |
| | **YNG2** | AtING2 | AT5G54390 | - | + | + | + | - | + | + | - | **2/1.96** |
| | **EAF6** | AtEAF6 | AT4G14385 | - | + | + | + | - | + | + | - | **2/0.29** |
| | **EAF3** | MRG1 | AT4G37280 | - | - | + | - | - | - | - | - | - |
| | | MRG2 | AT1G02740 | - | - | - | - | - | - | - | - | - |
| | **EAF5** | - | - | | | | | | | | | |
| | **EAF7** | - | - | - | - | + | - | - | - | - | - | - |
| **Additional AtEPL1B interactors detected here** | | AtEAF7 | AT1G26470 | - | - | - | - | - | - | - | - | **2/0.21** |
| | | AHASS1 | AT2G31810 | - | - | - | - | - | - | - | - | **2/0.18** |
| | | AOG2 | AT3G07810 | - | - | + | + | - | - | - | - | **2/0.06** |
| | | BRD6 | AT3G60110 | - | - | - | + | - | - | - | - | 1/0.06 |
| | | GATB | AT1G48520 | - | - | - | - | - | - | - | - | 1/0.04 |
| | | UGD1 | AT1G26570 | - | - | - | - | - | - | - | 114 | 0 |
| **Number of additional bait-specific hits** | | | | 2 | 454 | 1104 | 331 | 280 | 96 | 241 | 114 | |

*NSAF* normalized spectral abundance factor.
Numbers in the rightmost column indicate the number of biological replicates (out of two) in which the protein was detected as well as its relative abundance in the final eluate analyzed by mass spectrometry.
Values in bold indicate proteins for which peptides were detected in both biological replicates.

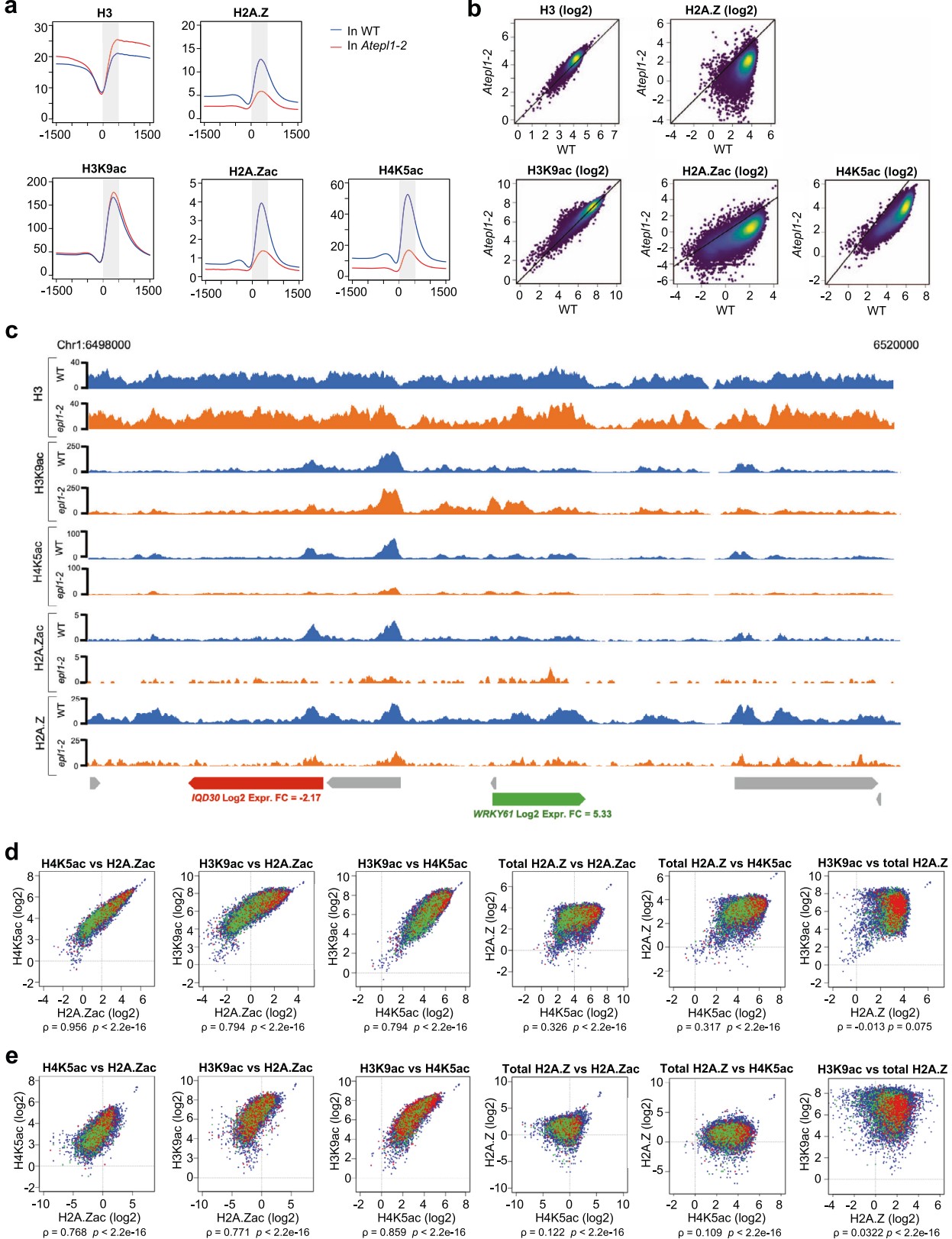

Fig. 6c), confirming the compensatory role of this acetylation. In turn, we observed an inverse correlation between expression fold change and non-acetylated H2A.Z change (Fig. 6c). This suggests that the dual influence of H2A.Z on gene expression is due to its presence in the acetylated (transcriptional activation) or non-acetylated (repression) form.

**Nuclear plastid gene expression depends on NuA4**. As shown above, loss of AtEPL1 affected the acetylation levels of H4 and H2A.Z, confirmed herein as genuine NuA4 targets, as well as acetylation of H3 catalyzed by other histone acetyltransferases. We interpreted H3K9ac changes at individual loci as indicative of a transcriptional response to physiological and developmental

**Fig. 5 NuA4 acetylates H4 and H2A.Z and stimulates H2A.Z deposition on chromatin. a** TSS-centered profiles of spike-in-normalized occupancy of H3, H2A.Z, H3K9ac, H4K5ac and H2A.Zac averaged over all genes expressed in WT Col-0 ($n = 18,620$). The values on the vertical axes are relative. The values on the horizontal axes indicate the chromosomal position relative to the TSS (i.e., negative values represent the promoter region, whereas positive values represent the gene body). To profile H2A.Zac we used an antibody raised against human H2A.Zac, which also specifically recognizes Arabidopsis H2A.Zac (Crevillén et al.[17]; Supplementary Fig. 8c). **b** Scatter plots showing differences in the H3, H2A.Z, H3K9ac, H4K5ac and H2A.Zac occupancy between *Atepl1-2* and WT. Each dot represents the log$_2$-transformed ChIP-seq signal averaged over the first 500 bp downstream of the TSS (shaded in **a**) of a single gene in WT and *Atepl1-2*. The number of dots per pixel is color-coded. **c** A fragment of chromosome 1 showing spike-in-normalized H3, H2A.Z, H3K9ac, H4K5ac, or H2A.Zac occupancy across several loci. The particular chromosomal locations were chosen to illustrate changes in the chromatin environment coinciding with increased (green) or decreased (red) transcript levels vs. transcript levels with no significant change (gray). The values on the vertical axes are relative. **d** Scatter plots showing interrelationships between H2A.Zac, H4K5ac, H3K9ac and total H2A.Z occupancy in WT. Each dot represents the log$_2$-transformed ChIP-seq signal averaged over the first 500 bp downstream of the TSS (shaded in **a**) of a single gene. The color of dots indicates gene transcriptional response in *Atepl1-2* (upregulation, green, downregulation, red, no change, blue). Spearman rank correlation values are shown below the plots. **e** As in **d**, but for *Atepl1-2*.

perturbations observed in the mutant and not as a direct consequence of NuA4 inactivation. Consequently, genes that did not show reduced H3K9ac but displayed large reductions in H4 and H2A.Z acetylation and reduced transcript levels could be viewed as NuA4-dependent. To strictly define this group of genes, we set the following arbitrary criteria: (i) Log$_2$ FC in the transcript level $<-1.5$, (ii) Log$_2$ FC in H4K5ac $< -2$, and (iii) Log$_2$ FC in H3K9ac $> -0.5$ (Supplementary Fig. 13a and Supplementary Data 17). We plotted the log$_2$-transformed FC values of the two acetylation marks against each other separately for genes down- and upregulated in *Atepl1-2* (Fig. 7a). Application of the acetylation fold change criteria (ii) and (iii) to genes downregulated in *Atepl1-2* resulted in the identification of 347 potentially NuA4-dependent genes (Supplementary Data 17).

Gene ontology analysis of the NuA4-dependent gene group vs. all genes downregulated in *Atepl1-2* showed significant enrichment of categories related to growth and photosynthesis, including genes encoding components of the chloroplast stroma, thylakoid, and chloroplast envelope (Supplementary Fig. 13b, Supplementary Table 9, and Supplementary Data 18). Interestingly, nearly all nuclear genes coding for plastid ribosomal proteins[53] clustered within or near the NuA4-dependent sector (Fig. 7a and Supplementary Data 18). Furthermore, the median log$_2$ FC in the expression of plastid translation- and ribosome-related NuA4-dependent genes was significantly lower than that of all downregulated genes, as could be expected for the genes that directly depend on NuA4 for efficient transcription (Supplementary Fig. 13b). This finding indicates that the NuA4 complex specifically stimulates the expression of growth- and photosynthesis-related genes in plants.

**The presence of H2A.Z determines NuA4 dependency.** NuA4-dependent genes were identified as a subgroup of 347 genes that showed significant loss of H4 acetylation. We sought to determine how these genes differ from other H4K5ac-depleted genes that did not display decreased transcript levels in the mutant. Studies in *S. cerevisiae* have suggested that H4 acetylation plays an important role in the recruitment of the general transcription factor TFIID, especially to genes that lack a TATA box in their core promoters[54], which is the case for approximately 80% of Arabidopsis npRPGs[55]. The interaction between TFIID and H4ac is mediated in yeast by the double bromodomain protein BDF1[56]. Animals and plants do not have BDF1 homologs and instead harbor bromodomains within the TAF1 subunits of their TFIIDs[57]. We used Cas9-gRNA to eliminate the bromodomains from both Arabidopsis TAF1 homologs to test whether this alteration has an effect on gene expression similar to loss of H4ac in the NuA4 mutants (Fig. 7b). Intriguingly, double *taf1Δbromo* homozygotes showed no obvious phenotypic effects (Fig. 7c). RT-qPCR analysis of the relative transcript levels of several

NuA4-dependent genes also showed no similarity to those in *Atepl1* (Fig. 7d). In fact, some transcripts showed significant changes in the opposite direction, consistent with a previous study of a single *haf2* mutant (*taf1b*) (Fig. 7d)[37]. Assuming functional conservation of the relationship between TFIID and H4 acetylation, these results demonstrate that recruitment of TFIID via TAF1 bromodomains is not the primary mode of action of H4ac at NuA4-dependent loci in plants.

Next, we sought to identify chromatin states that distinguish NuA4-dependent genes from other genes that were downregulated in *Atepl1-2*. First, we observed that this gene class showed a very high level of H2A.Z and H2A.Zac in the $+1$ nucleosome, unlike the broader group of all genes that were downregulated in *Atepl1-2* (Fig. 7e). In *Atepl1-2*, NuA4-dependent genes showed a significantly greater decrease in the level of H2A.Zac than did all expressed genes, while the two groups of genes did not differ in the decrease in the level of total H2A.Z (Fig. 7e, f). This pattern may suggest that in the mutant background, the remaining unacetylated H2A.Z in the $+1$ nucleosome in NuA4-dependent genes inhibits the expression of these genes. In addition, we confirmed these observations by comparing different groups of plastid NuA4-dependent genes with all the genes in their GO groups (Supplementary Fig. 14). We speculate that elevated H2A.Zac levels on $+1$ nucleosomes determine the NuA4 dependency of gene transcriptional activity.

**Upregulation of genes in *Atepl1-2* results from H2A.Z loss.** Gene body H2A.Z can act as a transcriptional repressor of responsive genes[52,58,59]. Since many of the genes upregulated in *Atepl1-2* could be classified as responsive (Fig. 3a and Supplementary Data 11), we hypothesized that their increased activity may be a consequence of decreased H2A.Z occupancy. To test this hypothesis, we compared the averaged H2A.Z enrichment profiles in *Atepl1-2* and WT for genes upregulated or downregulated in the mutant and for all expressed genes. This analysis showed that genes upregulated in *Atepl1-2* were relatively rich in gene body H2A.Z but had little H2A.Zac in the $+1$ nucleosome in WT plants (Fig. 8a). The averaged H2A.Z profiles of these genes were flattened in the mutant, overlapping other gene classes (Fig. 8a and Supplementary Fig. 15a). Consequently, the changes in gene body H2A.Z levels between WT and *Atepl1-2* were greater for the upregulated genes than for all expressed genes (Fig. 8b). A similar comparison for acetylated H2A.Z in gene bodies showed no significant difference. This pattern indicates stimulation of H2A.Z deposition and not H2A.Z acetylation per se as a key contribution of NuA4 to H2A.Z-dependent gene silencing (Fig. 8b).

If loss of gene body H2A.Z was the cause of gene upregulation in the NuA4 mutants, a similar set of genes should be upregulated in mutants that lack H2A.Z in nucleosomes. We found significant

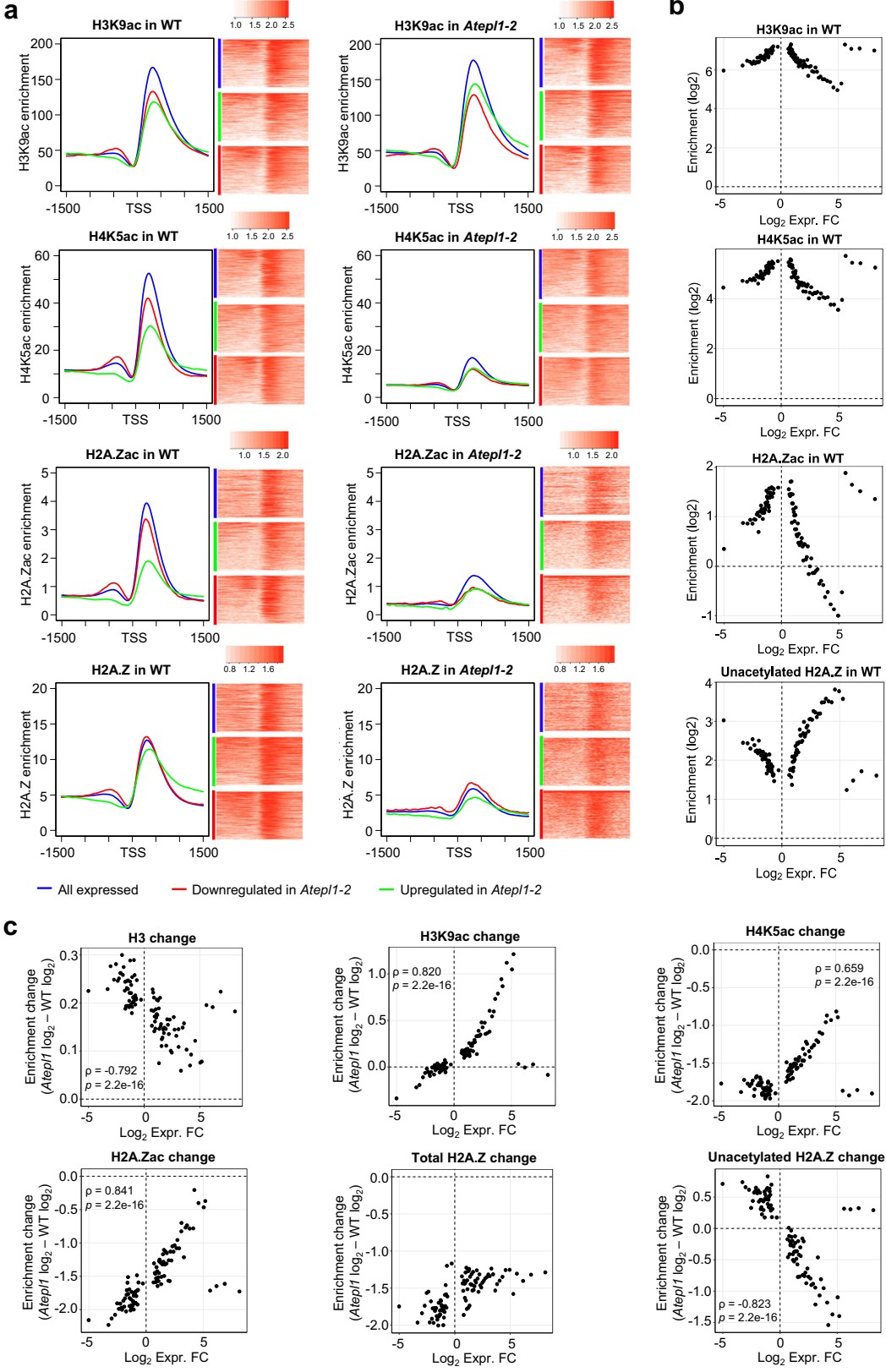

overlap between genes upregulated in both NuA4 mutants described here and those upregulated in the *pie1-2* mutant, which lacks the catalytic subunit of the SWI2/SNF2-Related 1 chromatin remodeling complex (SWR1-C) responsible for H2A.Z deposition in nucleosomes[60,61] (Fig. 8c). How do the genes upregulated in both *Atepl1-2* and *pie1* differ from these

upregulated only in one of these mutants? To answer this question, we investigated the levels of gene body H2A.Z and H2A.Z/H4K5/H3K9 histone acetylation. We found that gene body H2A.Z enrichment in the three groups of genes followed the order of upregulated in *Atepl1-2* and *pie1* > upregulated only in *pie1* > upregulated only in *Atepl1-2*, but gene body H2A.Z

**Fig. 6 Gain of H3ac and loss of H4ac and H2A.Zac shape the *Atepl1-2* transcriptome. a** Spike-in-normalized H3K9ac, H4K5ac, H2A.Zac, and total H2A.Z profiles at the TSS for all genes expressed in the WT ($n = 18{,}620$; blue) as well as for genes downregulated ($n = 941$; red) and upregulated ($n = 1746$; green) in *Atepl1-2*. For each pair of plots, the plot on the left shows the acetylation profiles as detected in the WT, while the plot on the right shows the modifications in the *Atepl1-2* mutant. The values on the vertical axes are relative. The values on the horizontal axes indicate the chromosomal position relative to the TSS. The heatmaps show the corresponding three groups of genes ordered according to the average enrichment level on a logarithmic scale. For the all expressed genes category, 2500 randomly selected genes are shown. **b** Scatter plots showing correlations between chromatin marks (H3K9ac, H4K5ac, H2A.Zac, and unacetylated H2A.Z occupancy in WT) and gene expression change ($\log_2$ fold change) as calculated by percentiles of expression change. Genes dysregulated in *Atepl1-2* ($n = 5135$) were sorted based on their fold change and divided into percentiles. For each percentile, the mean change in expression and the mean $\log_2$-transformed spike-in normalized ChIP-seq signal were calculated and plotted. To estimate values for non-acetylated H2A.Z, the ChIP-seq signal for acetylated H2A.Z were subtracted from the signal obtained for total H2A.Z. **c** Scatter plots showing correlations between changes in chromatin marks (H3, H3K9ac, H4K5ac, H2A.Zac, total H2A.Z, and unacetylated H2A.Z occupancy) and gene expression change ($\log_2$ fold change) as calculated by percentiles of expression change. Genes dysregulated in *Atepl1-2* ($n = 5135$) were divided into percentiles as in **b**. For each percentile, the mean change in expression and the mean difference in the level of the $\log_2$-transformed spike-in normalized ChIP-seq signal between WT and *Atepl1-2* were calculated and plotted. To estimate values for non-acetylated H2A.Z, the differences for acetylated H2A.Z were subtracted from the differences obtained for total H2A.Z. Spearman rank correlation values are shown below the plots.

enrichment in all groups was much higher than the average for all expressed genes (Fig. 8d and Supplementary Fig. 15b). This finding further supports our hypothesis that H2A.Z loss from gene bodies is largely responsible for gene upregulation in *Atepl1-2*. Conversely, the average histone acetylation levels were lower in the group of genes upregulated in both *Atepl1-2* and *pie1* than in the other groups (Fig. 8d and Supplementary Fig. 15). In the *Atepl1-2* mutant, the levels of H2A.Z, H2A.Zac and H4K5ac in all gene groups decreased in a similar manner (Supplementary Fig. 15c). However, while the H3K9ac level was significantly increased in the *Atepl1-2* background in genes upregulated in both mutants and for those upregulated only in *Atepl1-2*, it was not altered in genes upregulated only in *pie1* (Fig. 8e). This pattern again points to H3 acetylation as the driver of gene activation in *Atepl1-2*. Consequently, those genes that did not gain H3 acetylation upon loss of H2A.Z remained silent in *Atepl1-2* despite their activation in *pie1* driven by intact NuA4-dependent H4 acetylation. In conclusion, a significant proportion of genes upregulated in the NuA4 mutants require cooperation between NuA4, SWR1 and possibly histone deacetylases for transcriptional repression mediated by deacetylated gene body H2A.Z (Fig. 9).

## Discussion

The Cas9-induced loss-of-function mutant presented here provides evidence supporting a central role of AtEAF1 in plant NuA4 (Figs. 1 and 2). First, the phenotype of the *Ateaf1-1* mutant is significantly more severe than the phenotypes of the *mrg*, *eaf7* and *yaf9* mutants described previously[17,25,27]. We note that lethal effects have been reported for the *Atswc4* mutant as well as the double *ham* mutant[24,62]. These effects are probably not NuA4 specific, however, due to the role of AtSWC4 in the SWR1 complex and the potential NuA4-independent roles of HAM in the acetylation of nonhistone substrates[62,63]. Furthermore, the *Ateaf1-1* mutant displays extreme enhancement of the photosynthetic and reproductive phenotypes previously reported in a double mutant of AtYAF9, a confirmed interacting protein of AtEAF1[16,17].

The severe phenotype of *Ateaf1-1*, which is most likely caused by the dissociation of the regulatory module of NuA4, suggests that proper targeting of picNuA4 activity is critically important in plants. The phenotype of *Ateaf1-1* is not copied by any mutation disrupting a nuclear transcription factor described to date in Arabidopsis, although some aspects were reminiscent of the phenotype of the well-characterized double *glk* mutant (Fig. 2)[30,35]. Our genetic and transcriptomic analyses provide evidence that NuA4 and GLK operate mostly independently.

Since the phenotypes and transcriptomes of the *Ateaf1* and *Atepl1* mutants differed quantitatively rather than qualitatively (Figs. 1 and 2), we assumed that similarly widespread chromatin changes occurred in these two mutants. Consequently, we focused on the *Atepl1-2* mutant, which allowed us to investigate the impact of complete NuA4 inactivation. Interestingly, despite the roughly equal numbers of down- and upregulated genes in *Atepl1-2*, the levels of total H2A.Z, H2A.Zac, and H4K5ac but not H3K9ac were globally decreased (Figs. 3 and 5). Investigating the reasons for this pattern led us to discover that only those genes with low levels of all three modifications in WT plants show dysregulation in the *Atepl1-2* mutant (Fig. 6b). However, the degree of change in their expression is strongly correlated with the degree of change in the level of H2A.Z, H4K5 and H3K9 acetylation in *Atepl1-2* (Fig. 6c). These findings indicate that H3K9ac may effectively stimulate transcription in the absence of nucleosomal H4 acetylation in plants. Remarkably, an inverse relationship exists for the level of non-acetylated H2A.Z (Fig. 6c). This shows that the previously described dual role of H2A.Z in the regulation of gene expression[17,64,65] results from its presence in the +1 nucleosome in two forms: acetylated, which activates, and non-acetylated, possibly monoubiquitinylated[66], which represses transcription.

Several characteristics indicate that the genes that are down-regulated in *Atepl1-2* can be both directly and indirectly regulated by NuA4. Therefore, we defined genes downregulated as a direct consequence of NuA4 loss in *Atepl1-2*. We found that these genes were enriched in GO categories related to plastid translation machinery (Supplementary Fig. 14), which is in line with the NuA4 mutant phenotype. We observed that this group of genes showed strong enrichment of the H2A.Z histone variant in the +1 nucleosome (Fig. 7e). The presence of H2A.Z in the +1 nucleosome is known to prepare genes for transcriptional activation[52,67,68], and H2A.Z acetylation is likely required in this process[17,64,65]. Hypoacetylated H2A.Z in the +1 nucleosome may constitute a major barrier for RNAPII during transcription[69]. As we showed that NuA4 is responsible for H2A.Z acetylation in plants (Figs. 4g–6), the absence of the complex may specifically affect the expression of genes enriched with H2A.Z: in this scenario, deficiency of both H4 and H2A.Z acetylation cannot be compensated by increased H3 acetylation, leading to transcriptional repression (Fig. 9a).

Although histone acetylation has long been considered a general transcriptional coactivator, our research shows that this view is greatly simplified. Similar to findings in yeast[49], we found that plant NuA4 enables H2A.Z to be deposited efficiently on chromatin (Figs. 5–6). It is important to note that the role of NuA4 in this process is not limited to the +1 nucleosome but also extends

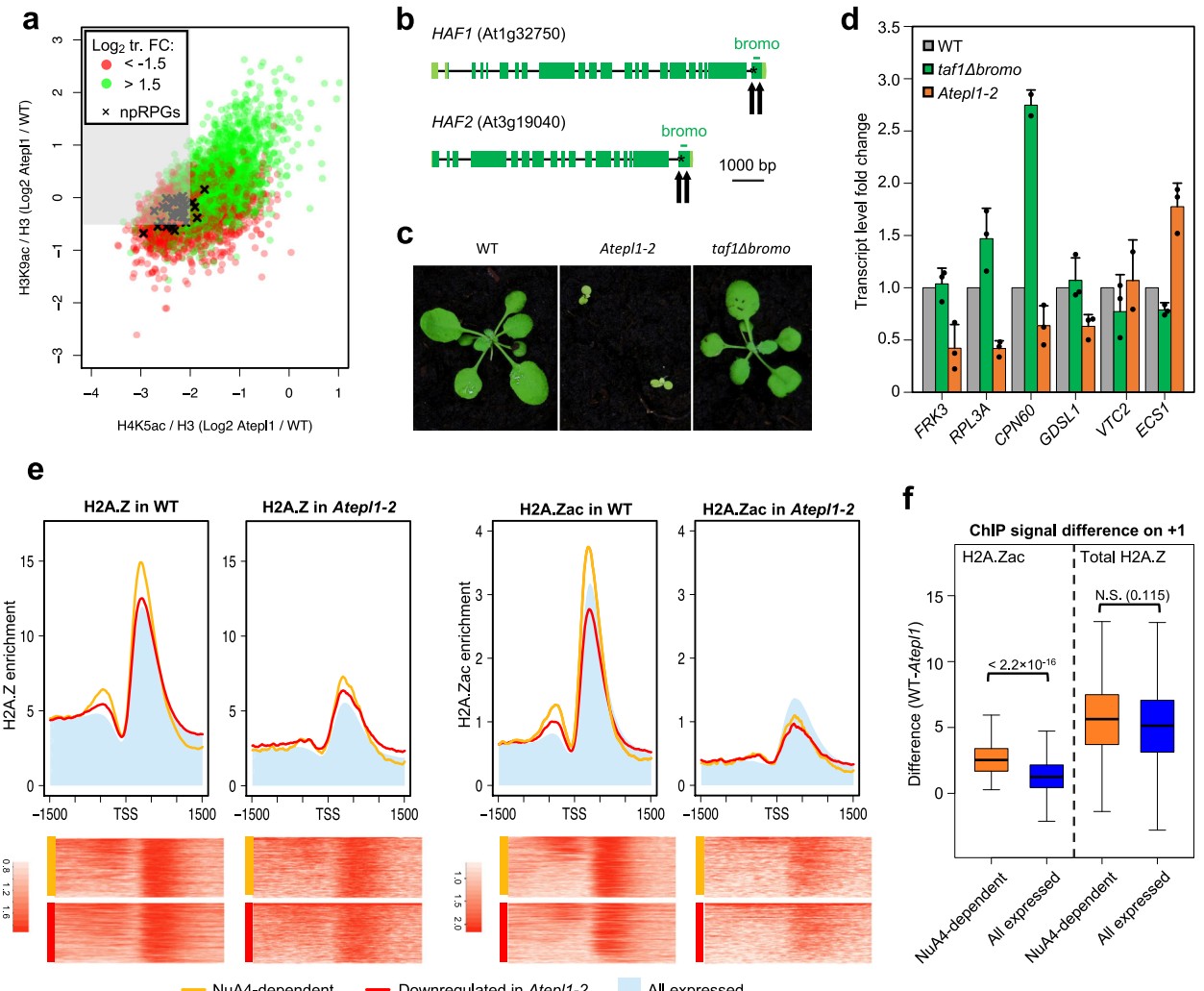

**Fig. 7 H2A.Z enrichment on +1 nucleosomes and not TAF1 bromodomain-mediated TFIID recruitment determines the NuA4 dependence of npRPGs.**
**a** Scatter plot showing the relationships between the log$_2$-transformed fold changes in the spike-in- and H3-normalized H3K9ac and H4K5ac occupancies and the directionality of the transcript level change. Genes downregulated in *Atepl1* shown within the gray area fulfill all criteria for NuA4 dependency. **b** Gene structures of the two *Arabidopsis thaliana* genes encoding the TAF1 subunit of TFIID. The black arrows indicate the Cas9-gRNA cut sites, and the black asterisks mark the positions of premature stop codons in the resulting mutant alleles. **c** Phenotypes of *taf1Δbromo*, *epl1-1* and WT plants. **d** RT-qPCR analysis of the relative transcript abundance in the double bromodomain mutant (*taf1Δbromo*) and the double *Atepl1a-1 Atepl1b-1* mutant (*Atepl1-1*). The genes were selected on the basis of their distinct transcriptional behavior in the *haf2-1* mutant (as reported in Bertrand et al.[37]) and in the *Atepl1-1* mutant, as shown in this work by RNA-seq. Average expression levels from three independent replicates ±standard deviations are shown. **e** Spike-in-normalized H2A.Zac and H2A.Z profiles at the TSS for NuA4-dependent genes (*n* = 347; orange) as compared to genes downregulated in *Atepl1-2* (*n* = 941; red) and all genes expressed in the WT (*n* = 18,620; blue shading). The values on the vertical axes are relative. The values on the horizontal axes indicate the chromosomal position relative to the TSS. The heatmaps show the corresponding NuA4-dependent and Downregulated in *Atepl1-2* groups of genes ordered according to the average enrichment level on a logarithmic scale. **f** Boxplots showing the difference between the average H2A.Zac and H2A.Z enrichment in the first 500 bp downstream of the TSS for NuA4-dependent genes (*n* = 347; orange) and all genes expressed in the WT (*n* = 18,620; blue). The center line indicates the median; upper and lower bounds indicate the 75th and 25th percentiles, respectively; and the whiskers indicate the minimum and maximum. An unpaired two-sample Wilcoxon test was used to determine significance. Source data are provided as a Source Data file.

to H2A.Z in gene bodies, which generally has a repressive function in preventing spurious activation of responsive genes[52,58,59]. Consequently, NuA4 inactivation in plants leads to a global decrease in H2A.Z deposition and spurious expression of stress-responsive genes (Figs. 8a, b and 9a). The expected overlap between genes upregulated in the absence of NuA4 and those upregulated in the absence of the H2A.Z-depositing complex SWR1 is relatively modest (24.4% and 26.1% for *Atepl1-2* and *Ateaf1-1*, respectively; Fig. 8c). The reason for this may be the

simultaneous decrease in histone acetylation in the +1 nucleosome, which seems to be necessary for the transcriptional activation of many responsive genes (Fig. 8d, e). Therefore, while most genes displaying a high gene body H2A.Z level are upregulated in the *pie1* mutant, only genes that are able to compensate for the lack of H4/H2A.Z acetylation by H3 hyperacetylation are transcriptionally activated in the NuA4 mutants (Fig. 8e).

Transcriptional inhibition of genes related to photosynthesis frequently coincides with activation of the stress response in

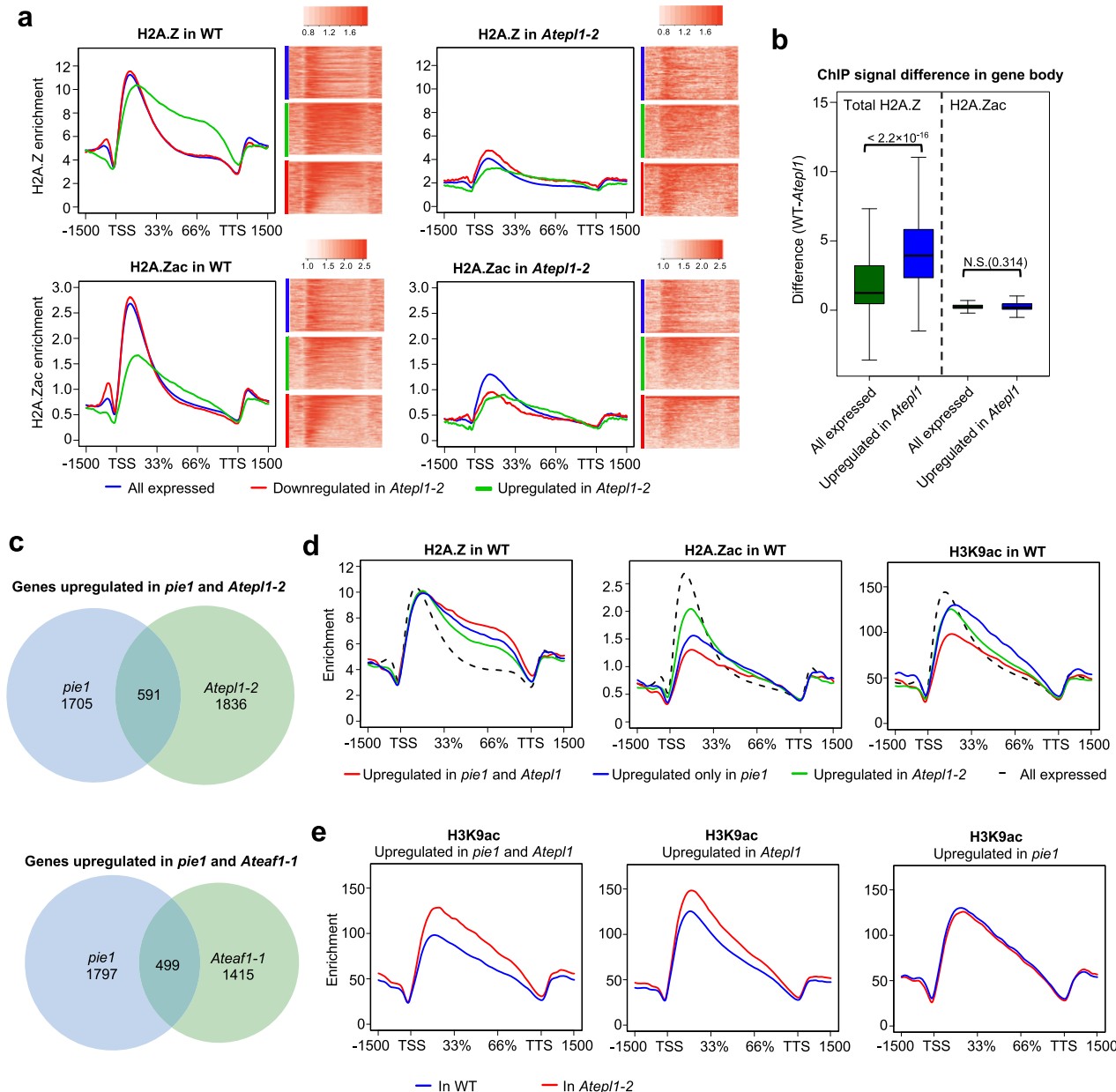

**Fig. 8 Upregulation of genes in NuA4 mutants results from both H2A.Z loss from gene bodies and increased levels of H3K9ac. a** Spike-in-normalized H2A.Z and H2A.Zac profiles in gene bodies for all genes expressed in the WT ($n = 18{,}620$; blue) as well as for genes downregulated ($n = 941$; red) and upregulated ($n = 1746$; green) in *Atepl1-2*. TSS, transcription start site; TTS, transcription termination site. **b** Boxplots showing the differences between the average H2A.Z and H2A.Zac enrichment in gene bodies for all genes expressed in the WT ($n = 18{,}620$; green) and for genes upregulated in *Atepl1-2* ($n = 1746$; blue). The center line indicates the median; the upper and lower bounds indicate the 75th and 25th percentiles, respectively; and the whiskers indicate the minimum and maximum. An unpaired two-sample Wilcoxon test was used to determine significance. **c** Area-proportional Venn diagrams showing the overlaps between genes exhibiting upregulation in the *Atepl1-1* or *Ateaf1-1* mutant and the *pie1* mutant. **d** Spike-in-normalized H2A.Z, H2A.Zac and H3K9ac profiles across gene bodies for genes upregulated in both *pie1* and *Atepl1-2* ($n = 591$; red), genes upregulated only in *Atepl1-2* ($n = 1836$; green), and genes upregulated only in *pie1* ($n = 1705$; blue), as measured by ChIP in WT plants. The corresponding profiles for all genes expressed in the WT are shown as a reference (dashed lines). **e** Comparison of spike-in-normalized H3K9ac profiles across gene bodies for gene groups as described in **d**, as measured by ChIP-seq in WT (blue) and *Atepl1-2* (red) plants.

plants, allowing survival under conditions that prevent normal development[70]. Our analyses demonstrate upregulation of stress-responsive genes and downregulation of genes related to auto-trophic growth in NuA4 mutants deficient in deposition and acetylation of H2A.Z (Fig. 9). We predict that future studies may reveal how the regulation of NuA4 activity switches between growth and stress responses in plants.

## Methods

**Plant material**. All *A. thaliana* mutants used in this work are in Col-0 background. T-DNA insertion mutants for AtEPL1A (At1g16690, SAIL_239_D12[71], *Atepl1a-1*) and AtEPL1B (At1g79020, SALK_094941[72], *Atepl1b-1*) were obtained from NASC and crossed to produce the double mutant *Atepl1-1*. The previously characterized double homozygous transposon-insertion mutant *glk1-1 glk2-1*[30] also comes from the NASC collection (NASC ID: N9807). The *arp6* and *mbd9-1* mutants were kindly provided by Kyuha Choi and Roger Deal, respectively. All other mutants

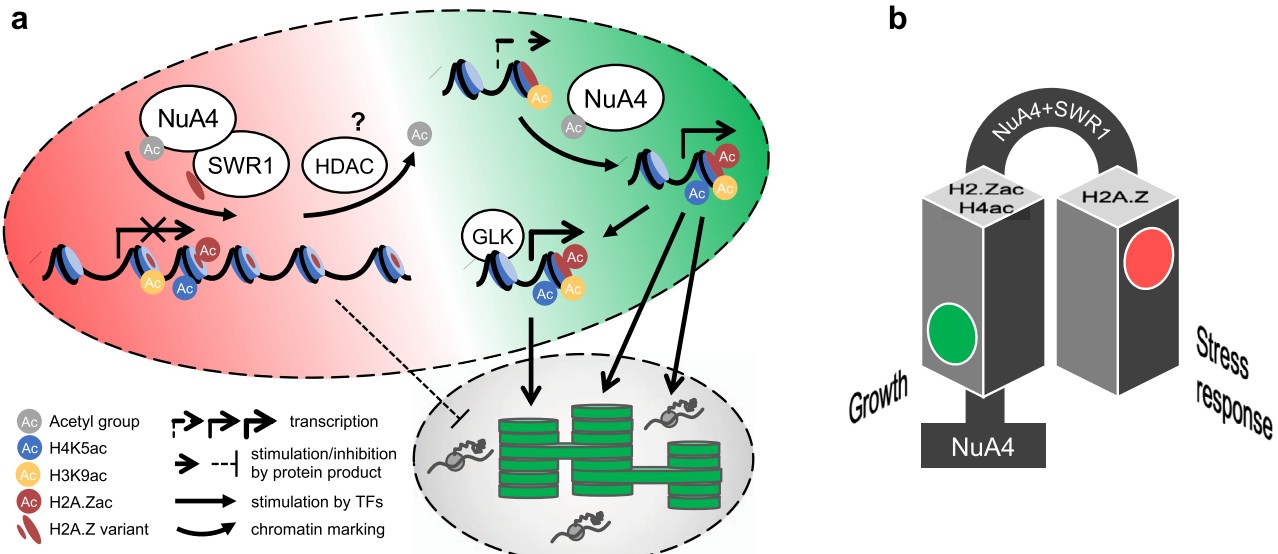

**Fig. 9 Model showing the role of NuA4 in the switch between autotrophic growth and the stress response. a** Regulation of gene expression involving NuA4. Large dashed-line oval represents nucleus, while small dashed-line oval represents chloroplast. Plastid and growth-related genes (green portion of the nucleus) depend on H2A.Zac and H4ac for their high transcriptional activity. Loss of H2A.Zac/H4ac from the +1 nucleosome at these loci cannot be compensated by H3K9ac and results in gene repression. GLK transcription factors function in parallel with NuA4 to activate genes involved in photosynthesis but not RPGs. Stress response genes (red portion of the nucleus) are repressed by hypoacetylated H2A.Z in their gene bodies (represented by +2 nucleosome). Deposition of H2A.Z by the SWR1 complex in the gene body requires NuA4, although it is not known whether acetyl groups are later removed by an unknown HDAC or not deposited at all (not shown). Loss of NuA4 activity triggers transcriptional release of stress genes that have sufficient H3ac levels in the +1 nucleosome. **b** By controlling H2A.Z acetylation and deposition, NuA4 contributes to the switch between autotrophic growth and the stress response.

were generated for this work through Cas9-gRNA mediated genome editing of WT Col-0 plants and are described in detail in Supplementary Figs 1–2.

**Preparation of genetic constructs for gene editing.** The Cas9 expression cassette was created through fusion of the *ICU2* promoter sequence cloned from genomic DNA following published instructions[73] with the 3 × Flag-NLS-hSpCas9-NLS::NosT cassette which was a gift from Dr Jian-Kang Zhu[74]. Following insertion of the Cas9 expression cassette into the pFGC vector (a gift of Dr Jian-Feng Li[75]), the *BamHI* site was moved from the 5′- to the 3′-end of the Nos terminator. The *BamHI* site was subsequently used for linearization of the plasmid and insertion of gRNA cassettes by Gibson assembly-based cloning. The gRNA expression cassette containing U6 promoter was a gift from Dr Jian-Kang Zhu[74]. The U3 expression cassette was created by replacing the U6 promoter with the *AtU3C* promoter cloned from genomic DNA isolated from *A. thaliana* Ler ecotype following published sequence data[76]. The target sequences were found with CRISPR-P[77] (*Ateaf1-1, Atepl1-2*) or CRISPOR[78] (*Atepl1-3*). Targeted mutagenesis of the gRNA expression cassettes was followed by PCR-amplification and recircularization of the pJET1.2-based gRNA vectors. The sequences of the primers used for the mutagenesis can be found in the Supplementary Data 19.

**Editing strategy and isolation of the mutants.** In order to generate genomic deletions within coding regions of the target genes, double-gRNA-Cas9 constructs were integrated with the genome through Agrobacterium-mediated transformation[79]. Transformed plants were selected on soil by spraying with Basta (Bayer). Two primers flanking the expected deletion were used to identify plants that accumulated somatic mutations apparent from the presence of short PCR products corresponding to deletions between two neighboring cut sites. T2 plants were screened for the absence of Cas9-gRNA T-DNA by PCR and genotyped again for the presence of desired genomic deletions. Any mutations detected in this way were assumed heritable. Subsequently, PCR amplification products were cloned into pJET1.2 vector and plasmids isolated from multiple single colonies were sequenced to determine the sequences of both homologous loci. If plants were confirmed as frame-shift- (*Ateaf1-1, Atepl1-2*) or in-frame-deletion mutants (*Atepl1-3*), they were backcrossed to Col-0 and propagated. Sequences of all primers used for genotyping can be found in Supplementary Data 19.

**Phenotypic analyses.** For all phenotypic analyses plants were grown in peat pellets (Jiffy) in controlled conditions: 22 °C, 80–90 μmol/(m²s) PAR, 60% RH. In course of the growth rate analyses, plants were imaged in one-week intervals with a stereomicroscope or a camera, depending on the size. ImageJ software was used for

the image analyses to extract the distance between the tip of the longest rosette leaf and the rosette center[80].

As an approximation for chloroplast and cell size, we used the surface projection areas of the organelles or cells obtained through live confocal imaging of the palisade mesophyll cell layer of rosette leaves. To eliminate variability resulting from changes in chloroplast size during leaf growth, we measured chloroplasts in the fully expanded 4th leaves of 10-true-leaf rosettes. For plastid- and cell-size measurements, leaves #4 were cut from 10-true-leaf rosettes, vacuum-infiltrated with perfluoroperhydrophenanthrene[81], mounted on a slide and imaged with the A1Rsi confocal microscope (Nikon). Z-stacks spanning the palisade mesophyll cell-layer, acquired in transmitted-light and chlorophyll autofluorescence channels, were used for analysis in ImageJ. Contours of cells and plastids were tracked manually and the surface area of the resulting regions of interest (ROI) was treated as the surface projection area of the measured structures. All calculations and statistical analysis were done using Microsoft Excel.

Chlorophyll was extracted from a single plant for each biological replicate. A 10-leaf rosette was separated from the root, weighed and ground in liquid nitrogen. After obtaining a fine powder, 3 or 5 ml (depending on a rosette size) of Tris-HCl-buffered 80% acetone was added and grinding was continued until homogeneity. The thawed mixture was then transferred to a 10 ml glass tube wrapped in aluminum foil to prevent light access, sealed and kept at 4 °C overnight. Directly before spectroscopic measurements, 1.5 ml of the mixture was transferred to 1.5 ml microtubes (Axygen) and centrifuged at 10,000 × g for 5 min to pellet any debris. After centrifugation, the extract was transferred to a spectrophotometer cuvette and the absorbance was measured at 647 nm and 664 nm using BioSpectrometer kinetic (Eppendorf) spectrophotometer against buffered acetone as blank. The number of samples processed in parallel was limited to 4 in order to reduce the influence of the processing time on the measurements. Total chlorophyll content, as well as chlorophyll a/b ratio, were calculated according to published formulas[82] without correction for the absorbance at 750 nm.

**Photomorphogenic response measurement.** For analyses of photomorphogenic response, wild-type and *Atepl1-2* seeds were stratified and grown in ½ Murashige-Skoog medium for three (apical curvature) or five (hypocotyl length, cotyledon angle and surface) days in the darkness and then transferred to constant light and controlled conditions: 22 °C, 80–90 μmol/(m²s) PAR, 60% RH. Apical curvature and cotyledon angle measurements have been conducted as previously described[83,84]. Briefly, plants that were analyzed for hypocotyl length were grown vertically. Pictures have been taken in 0 h, 6 h, 24 h, and 56 h intervals using stereomicroscope. ImageJ software was used for the analyses of captured images.

**Pull-down.** Recombinant proteins were expressed in *E. coli* strains BL21-CodonPlus-RP or Lemo21(DE3, New England Biolabs). Bacterial lysates were prepared as previously described[85] with the following modifications. GST-tag and His-tag fusion proteins were resuspended in PBS or equilibration and wash buffer (50 mM sodium phosphate, pH 8.0, with 0.3 M sodium chloride, 10 mM imidazole) respectively, with addition of 1 mM phenylmethylsulfonyl fluoride (PMSF) and cOmplete™ Mini EDTA-free Protease Inhibitor Cocktail (Roche). His-GFP or His-fusion proteins were bound to HIS-Select HF Nickel Affinity Gel (Sigma) for 1 hour, then washed and incubated with the recombinant GST-fusion proteins for additional 1 h at 4 °C. The beads were washed five times with wash buffer containing 0.5% NP40. The protein complexes were analyzed by SDS/PAGE followed by WB analysis with anti-GST (B14; #sc-138; Santa Cruz; 1:500) and anti-His (#R940-25; Life Technology, Waltham, MA, USA; 1:6000) antibodies.

**Far western blotting.** In Far Western Blotting (FWB), one of the recombinant proteins is subject to SDS–PAGE and western blotting. After renaturation, the bait protein attached to the PVDF membrane is incubated with the potential interactor. Overlap between the bait and prey signals on the membrane after the immuno-detection of the prey indicates an interaction. This experiment was performed as previously described[86] with some modification. Briefly, GST, GST-AtEPL1A and GST-AtEPL1B were purified on Glutathione Sepharose 4 Fast Flow beads (GE Healthcare) for 1 h at 4 °C. The beads were washed five times with PBS containing 1% Triton X-100. The GST-fusion proteins were eluted by heating at 95 °C in SDS/PAGE loading buffer, resolved by SDS/PAGE and transferred onto PVDF membrane. Proteins were renatured by incubation of the membrane in AC buffer (20 mM Tris pH 7.5, 1 mM EDTA, 0.1 M NaCl, 10% glycerol, 0.1% Tween-20, 2% nonfat milk, 1 mM DTT) containing 6 M guanidine-HCl for 30 min at RT. Subsequently, the membrane was washed with AC buffer containing 3 M guanidine-HCl for 30 min at RT. This step was followed by washing with AC buffer containing 0.1 M- and no-guanidine-HCl AC buffer at 4 °C, for 30 min and overnight respectively. The membrane was blocked for 8 h at 4 °C in 5% nonfat milk in the PBST buffer and subsequently incubated with His-HAM1 or His-HAM2. Next, the membrane was washed with 5% nonfat milk in the PBST buffer followed by three washes with PBST. To detect bait proteins bound to the bait proteins (GST-fusions) on the membrane, standard immunodetection with anti-His antibody (#R940-25; Life Technology, Waltham, MA, USA; 1:6000) was performed.

**In vitro acetylation assay.** Coding sequences of AtEPL1A, HAM1, AtING2, and AtEAF6 were cloned to the pST44 polycistronic vector for protein expression in *E. coli*, according to the original pST44 documentation[87]. Three different constructs were prepared: (1) with Flag-tag added to the N-terminus of AtEPL1A and 6×His-tag added to the C-terminus of HAM1, (2) with Flag-tag added to N-terminus of HAM1 and 6×His-tag added to C-terminus of AtING2, and (3) with Flag-HAM1 and ING2-6×His in a pST44 vector without the AtEPL1A sequence. Protein expression was conducted in BL21(DE3)pLysS cells after 0.3 mM IPTG induction and the complex was purified over Talon (Clontech) cobalt affinity resin[88] via HAM1-6×His, using 150 mM imidazole for the elution. The presence of all subunits was visible on Coomassie-stained SDS-PA gel and the presence of HAM1-His and Flag-AtEPL1A was confirmed by Western blot. For the Flag-HAM1, purification was first performed with cobalt resin (based on ING2-6×His), followed by incubation with anti-Flag M2 resin and elution with 3×FLAG peptides. Human K562 cells were engineered to express FLAG-tagged Arabidopsis HTA9 (a 3×Flag fused to HTA9 C-terminus) from the *AAVS1* safe harbor loci using Zinc-Finger Nuclease-mediated recombination as described[41]. Chromatin enriched extracts were prepared with micrococcal nuclease as described[46] and HTA9-containing nucleosomes were purified by incubation with anti-Flag M2 resin, followed by washes and elution with 3×Flag peptides. The gel presented in Fig. 4f shows efficient incorporation of HTA9 into human nucleosomes. HAT assays were performed as described previously[89] with some changes. Briefly, in liquid HAT assays 0.5 µg of purified human free core histones (CHs) or native H1-depleted short oligonucleotides (SONs) were incubated together with bacterially expressed plant picNuA4 (or yeast picNuA4 as a positive control) complexes with ³H-labeled acetyl-CoA (0.125 µCi) in HAT buffer (50 mM Tris-HCl pH 8.0, 50 mM [KCl +NaCl], 5% glycerol, 0.1 mM EDTA, 1 mM DTT, 1 mM PMSF, 10 mM sodium butyrate) at 30 °C for 30 min. Subsequently, the reactions were spotted on P81 membrane (Whatman), which were then air-dried, washed three times with 50 mM carbonate buffer pH 9.2, rinsed in acetone and soaked in scintillation cocktail, which was used to count ³H signal. Alternatively, the reactions were resolved by SDS–PAGE. After staining the electrophoresis gel in Coomassie to visualize proteins, it was destained and incubated in EN3HANCE solution (Perkin Elmer). After washing and drying, the gel was exposed to a film. For western blotting, the following antibodies were used: α-Flag (Santa Cruz sc-807, 1:5000), α-acH2A.Z (Diagenode C15410173, 1:1000), and α-H4 (Abcam ab7311, 1:3000).

**Tandem affinity purification.** The construct used for purification of proteins interacting with AtEPL1B was created by replacing the Gateway cassette in pMDC99 vector with the *AtEPL1B* expression cassette. The *AtEPL1B* cassette consisting of the genomic sequence of *AtEPL1B* (At1g79020) including 969 bp of the native upstream regulatory sequence, C-terminal GSrhino tag[90], native 3′ UTR

of *AtEPL1B* and Nos terminator was assembled in the pMDC99 backbone through Gibson assembly-based cloning. Growing of seedlings and TAP purifications using 50 g of 6-day-old seedlings as input per experiment, were performed as described in[90]. Bound proteins were digested on-bead after a final wash with 500 µL 50 mM NH₄HCO₃ (pH 8.0). Beads were incubated with 1 µg Trypsin/Lys-C in 50 µL 50 mM NH₄OH and incubated at 37 °C overnight in a thermomixer at 800 rpm. Next day, an extra 0.5 µg Trypsin/Lys-C was added and the digest was further incubated for 2 hours at 37 °C. Finally, the digest was separated from the beads and centrifuged at $20,800 \times g$ in an Eppendorf centrifuge for 5 min, the supernatant was transferred to a new 1.5 mL Eppendorf tube. Peptides were purified on C18 Bond Elut OMIX tips (Agilent) and later dried in a Speedvac and stored at −20 °C until MS analysis. Co-purified proteins were identified by mass spectrometry using a Q Exactive mass spectrometer (ThermoFisher Scientific) using standard procedures[91]. Proteins with at least two matched high confident peptides were retained. Background proteins were filtered out based on frequency of occurrence of the co-purified proteins in a large dataset containing 543 TAP experiments using 115 different baits[90]. True interactors that might have been filtered out because of their presence in the list of non-specific proteins were retained by means of semi-quantitative analysis using the average normalized spectral abundance factors (NSAF) of the identified proteins in the EPL1B TAPs. Proteins identified with at least two peptides in at least two experiments, showing high (at least 10-fold) and significant [-log10($p$-value(T-test)) ≥10] enrichment compared to calculated average NSAF values from a large dataset of TAP experiments with non-related bait proteins, were retained[92].

**RT-qPCR.** For RT-qPCR WT, *Atepl1-1*, *glk* and *Ateaf1-2 glk* plants were grown under LD photoperiod (16 h light and 8 h darkness) with constant conditions: 22 °C, 80/90 µmol/(m²s) PAR, 70% RH. The material was collected at the developmental stage of 10 leaves.

For each biological repeat, RNA was extracted from 100 mg of rosette leaves with TRI Reagent (Merck) and two rounds of phenol-chloroform extraction, followed by ethanol precipitation. RNA was treated with gDNA wiper (Vazyme) in order to digest genomic DNA. First strand synthesis was carried out with HiScript III 1ˢᵗ Strand cDNA Synthesis Kit (Vazyme). Sequences of primers used for RT-qPCR are shown in Supplementary Data 19.

**RNA-seq.** For RNA-seq WT, *Atepl1-2*, *Ateaf1-1*, and *glk* plants were grown under LD conditions (16 h light and 8 h darkness), under 80–90 µmol/(m²s) of photosynthetically active radiation. The material was collected at the developmental stage of ca. 10 leaves about 5 h after light onset.

RNA for each biological repeat was extracted from 110 mg of rosette leaves number 5–6 (from at least six plants) with TRI Reagent (Merck) and rounds of phenol-chloroform and chloroform extractions followed by isopropanol precipitation. RNA was treated with RQ1 DNase I (Promega), then extracted with phenol-chloroform and precipitated with ethanol. Pure RNA solution was sent to Macrogen Europe (South Korea), where libraries and sequencing were performed using Illumina chemistry. RNA-seq experiment was also validated by RT-qPCR (Supplementary Figs. 5 and 6).

Adapters, poor-quality regions (mean Phred quality <5) and rRNA sequences were removed from short reads using BBDuk2 program from BBTools suite (https://jgi.doe.gov/data-and-tools/bbtools). Filtered reads shorter than 50 bp were excluded from the analysis. Expression values for genes and transcripts (Araport11 annotations) were calculated using Salmon[93] with sequence-specific bias and GC content bias correction. Differential expression analysis on both gene and transcript level was performed using limma package[94,95]. False discovery rate (FDR) threshold of 0.01 and fold change threshold of 1.5 (Fig. 3) or Log₂ FC = 1.5 (Figs. 6–8) were used in the analyses.

**ChIP.** ChIP-seq analysis was carried out on WT and *Atepl1-2* plants grown in the same conditions as the plants used for RNA-seq and harvested at the same developmental stage. The isolated chromatin (from 2.1 g of fresh tissue per sample) was enriched with 2% chromatin isolated from mouse embryonic fibroblasts as spike-in control and incubated with 6 µg of antibody (α-H3, Abcam ab1791, α-H4K5ac, Millipore 07327, α-H3K9ac Millipore 07352, α-HTA9, Agrisera AS10718, α-H2A.Zac, Diagenode C15410173, pAb-173-050) overnight at 4 °C[52]. 25 µL of Dynabeads Protein A (Thermo Fisher Scientific) were added to the samples and incubated for another 2 hours at 4 °C. The slurry was washed twice with low salt buffer (with 150 mM NaCl), twice with high salt buffer (with 500 mM NaCl) and once with LiCl wash buffer (10 mM Tris-HCl, pH 8.0, 1 mM EDTA, 0.25 M LiCl, 1% Nonidet P-40, 0.5% sodium deoxycholate, 1 mM PMSF and protease inhibitor cocktail). Chromatin was eluted through incubation in 1% SDS and 0.8 M NaHCO3 at 65 °C and the cross-links were reversed by overnight incubation at 65 °C. Following Proteinase K treatment and phenol/chloroform extraction, DNA was precipitated and DNA concentration was measured with Qubit dsDNA HS Kit (Thermo Fisher Scientific). 10 ng (H3) or 2 ng (other antibodies) of ChIP-ed DNA was used to prepare DNA libraries with MicroPlex Library Preparation Kit v2 (Diagenode).

Adapters and poor-quality regions (mean Phred quality < 5) were removed from short reads using BBDuk2 program from BBTools suite (https://jgi.doe.gov/data-and-tools/bbtools). Filtered reads shorter than 35 bp were excluded from the

analysis. Reads were then aligned to the *A. thaliana* (TAIR10) and mouse (GRCm38) genomes using BWA-MEM aligner[96]. Sambamba[97] was used to process the aligned data and remove duplicated reads. Mapping statistics, including the percentages of mapped reads and duplicates, were calculated separately for *A. thaliana* genome (ChIP-Seq data) and mouse genome (spike-in data) using BamTools[98]. Normalized and smoothed coverage data used for ChIP-Seq plots was calculated using DANPOS2[99] and corrected using the fraction of deduplicated spike-in reads mapped to the mouse genome. Three biological replicates were performed for H3, H3K9ac and H4K5ac ChIP-seq, while two biological replicates were used for H2A.Z and H2A.Zac. In case of four samples one replicate was excluded from further analysis due to significantly lower coverage (Supplementary Table 10). The summary of ChIP-seq data is shown in Supplementary Table 10. The ChIP-seq profiles and the heatmaps were generated using DANPOS2[99].

ChIP was combined with quantitative PCR (ChIP-qPCR) to check the relative enrichment within NuA4-dependent genes and At5g15710, At5g16410. Sequences of primers used for ChIP-qPCR can be found in Supplementary Data 19. Real-time qPCR was carried out with Maxima SYBR Green qPCR Master Mix (Bio-Rad) according to the manufacture's instruction. Relative enrichment was calculated from the relative amount of input samples obtained with standard curves for each primer set. The antibodies used in ChIP-qPCR assays were: α-H3 (Abcam ab1791), α-H4K5ac (Millipore 07327), α-HTA9 (Agrisera AS10718) and α-H2A.Zac (Diagenode C15410173, pAb-173-050), H4K8ac (#07-328, Millipore), H4K12ac (#07-595, Millipore), H4K16ac (#07-329, Millipore). For ChIP-qPCR, 2 µg of antibody was combined with 100 µL of chromatin isolated from 0.7 g of fresh tissue.

**Statistics**. Calculations and statistical tests not described in RNA-seq or ChIP-seq sections were run and plotted in R (www.R-project.org), except Mann–Whitney U tests run with an online calculator (http://www.statskingdom.com). For creation of the area proportional Venn diagrams presented in Fig. 3b, the gene sets included in Supplementary Data 1-6 were first analyzed for overlaps by BioVenn[100], then plotted with Euler*APE*[101]. Gene set enrichment analyses presented in Fig. 3 were performed with PANTHER Overrepresentation Test[48] (Supplementary Data 7–12). Gene set enrichment analyses presented in Fig. 3c were performed with DAVID Functional Annotation Tool[102] (Supplementary Data 18).

**Reporting summary**. Further information on research design is available in the Nature Research Reporting Summary linked to this article.

## Data availability

All sequencing data that support the findings of this study have been deposited in the National Center for Biotechnology Information Gene Expression Omnibus (GEO) database under accession number GSE152940. The mass spectrometry proteomics data have been deposited to the ProteomeXchange Consortium via the PRIDE[103] partner repository with the dataset identifier PXD030302. Source data are provided with this paper.

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

## Acknowledgements

This work was supported by the Polish National Science Center (NCN) grants 2016/21/B/NZ2/01757 and 2016/22/E/NZ2/00455 to P.A.Z., 2015/17/N/NZ1/00028 to W.S. and the Foundation for Polish Science grant (POIR.04.04.00-00-5C0F/17-00) to P.A.Z. The PhD fellowship of TB was covered by the International PhD Programme of the Foundation for Polish Science (MPD/2010/3) and KNOW RNA Research Centre in Poznan (01/KNOW2/2014). Research in JC lab was supported by a Foundation grant from the Canadian Institutes of Health Research (FDN-143314). J.C. holds a Canada Research Chair in Chromatin Biology and Molecular Epigenetics. The four-month stay of W.S. in J.C. group was funded by NCN grant 2017/24/T/NZ1/00165. We thank Kyuha Choi (Postech Biotech Center, Korea) and Roger Deal (Emory University, US) for the *arp6* and *mbd9-1* mutants, respectively.

## Author contributions

Conceptualization: T.B., J.C., J.S., and P.A.Z.; Experimental work: preparation of mutant lines: T.B., W.S., A.B., M.A., P.W., W.D., and M.Sz-L., phenotypic analyses: T.B., W.D., and M.Sz-L., RNA-seq/RT-qPCR: W.S., W.D., and T.B., ChIP-seq/ChIP-qPCR: W.S., W.D., T.B., and M.Sz-L., protein–protein interactions/purification/enzymatic assays: A.B., W.S., C.L., and T.B., AP-MS: N.D.W. and T.B.; Data analysis: T.B., M.K., W.D., and P.A.Z.; Project administration: P.A.Z. and J.S.; Supervision: P.A.Z., J.C. (HAT assays), and G.D.J. (AP-MS); Writing and Editing: T.B., P.A.Z. with contributions from all authors.

## Competing interests

The authors declare no competing interests.
