## [Peer Review File · Nature Communications]

NuA4 and H2A.Z control environmental responses and autotrophic growth in ArabidopsisReviewers' Comments:

Reviewer #1:

Remarks to the Author:

In this manuscript, Bieluszewski, et al, explore the importance of the yet poorly understood connection between NuA4 acetylation and H2A.Z function in gene expression in Arabidopsis. The authors generate a nice repertoire of genetic mutants targeting two specific components of the putative NuA4 complex in Arabidopsis, the EAF1 and EPL1 proteins. Phenotypically, these mutants affect chloroplast development and resemble a mutation in GLK transcription factor involved in chloroplast gene activation. However this study falls short on some of the genomic and biochemical characterization listed below and requires further clarification/deeper characterization.

General point:

Although the authors generate loss-of-function mutations for both EAF1 and EPL1, they only characterize EPL1 mutants in their genomic study. It is not exactly clear why. It would be nice to include EAF1 mutants in RNA-Seq and ChIP-Seq studies to better understand the specificity of the NuA4 complex.

RNA-Seq analysis:

The authors have performed RNA-Seq of the Atepl1-1 mutant (Fig 3), but the ChIP-Seq has been done for the Atepl1-2 allele. In the ChIP-Seq section the authors refer to RNA-seq of Atepl1-2 allele. So is this just a typo, has RNA-Seq in Fig 3 been done for Atepl1-2 and not for Atepl1-1? If a different allele has been used for RNA-Seq and ChIP-seq, than conclusions regarding transcriptional changes as a result of chromatin changes may not be appropriate.

In Figure 3, the authors compare transcriptional misregulation of genes in Atepl1-1 to that of GLK transcription factor, where very little overlap was observed and conclude that NuA4 and GLK work independently. However, glk RNA-seq is from publically available data in seedlings, while the Atepl1-1 is done in rosette leaves. The RNA-Seq should be done side-by side to reach a convincing conclusion about the relationship of these mutants. Furthermore, the authors generated a Atepl1 glk double mutant which is viable. Therefore, in addition to the phenotypic characterization of the Atepl1 glk mutant, I recommend that the authors perform RNA-seq in parallel with the single mutants to clearly determine genetic relationship.

Biochemistry:

Using in-vitro expression in bacteria of several key components of the Arabidopsis NuA4 complex, the authors show that AtEPL1 interacts with the NuA4 catalytic subunit HAM1. This interaction parallels and verifies IP-mass spec studies generated in this and previous studies. The authors also perform an in-vitro HAT activity assay of the picNuA4 components (HAM1, AtEPL1A, ING and AtEAF6) purified from bacteria, showing that it is capable of acetylating H4 and H2A histone particles in-vitro. This is rather an expected outcome, a more interesting experiment would be to determine if/how AtEPL1 is required for HAM1 function. For instance, could the authors perform HAT activity of the picNuA4 complex in the presence and absence of AtEPL1 or with certain mutant versions?

ChIP-Seq:

The authors generated ChIP-seq in triplicates for H3K9Ac, H4K5Ac and H3 and convincingly show that a major change in acetylation in Atepl1-2 mutant involves a loss in H4K5Ac and not H3K9Ac. Even though majority of the Atepl1-2 characterization in this study has been done in relation to H4K5Ac changes, the connection between NuA4 and H2A.Z is the most interesting conclusion of the paper (also reflected by the title), and therefore deserves a deeper characterization. The authors identify a class of Atepl1-2 down-regulated and NuA4 dependent genes that contain high levels of H2AZ at +1 nucleosomes. Their acetylation is dependent on Atepl1-2 activity at a subset of loci. It would be of high interest to generate genome-wide H2A.Z and H2A.Z Ac maps in Atepl1-2 mutant to better understand the relationship of NuA4 acetylation and H2A.Z. In a previous study involving the

characterization of YAF9A/9B, a putative NuA4 components in Arabidopsis, Crevillen, P et al, 2019 concluded that at FLC locus, YAF9A/B are required for H2A.Z acetylation but not deposition. The data presented in his paper also suggests similar conclusion (Fig. 6). This is in contrast to models presented in yeast (Lu, P, Y, T et al, 2009). Exploring this relationship genome-wide will capture an interest of wider audience as it may highlight interesting differences between NuA4-H2A.Z connection in plants and other eukaryotic organisms.

Furthermore, the connection between NuA4 and H2A.Z could be strengthened genetically. It would be interesting to see the effect of H2A.Z deposition, acetylation and gene expression in higher order mutants of Atepl1-2 crossed with SWR1 mutants. For instance, MBD9, a newly characterized component of the SWR1 complex (Sijacic, P et al, 2019, Potok, M et al, 2019), has been shown to be specifically required for H2A.Z deposition at genes with H2A.Z localized to +1 nucleosome. So presumably these may be the same sets of genes that are NuA4 dependent identified in this study. In general, in MBD9 mutant, the reduction in H2AZ at +1 nucleosome does not lead to drastic transcriptional changes. This was similarly observed in this study upon depletion of H4K5Ac. It would be interesting to see if removal of H2AZ along with NuA4 depended acetylation could reveal some redundancies in the control of gene expression.

Minor point:

Regarding some of the H3K9Ac conclusions, its not clear how the authors came to the following conclusion: "H3K9ac levels were significantly lower among the genes that were downregulated in Atepl1-2 than among all expressed genes (..Fig 5D)". The levels of H3K9Ac seems to be the same in WT and Atepl1-2 mutant when looking at the y-Axis. Perhaps this point can be explained more clearly.

Reviewer #2:

Remarks to the Author:

The manuscript by Tomasz Bieluszewski and co-authors describes a novel role for the plant NuA4 HAT complex in controlling photosynthesis genes through H4K5ac and H2A.Z acetylation In Arabidopsis. The authors generate series of CRISPR-Cas9 mutants affecting Arabidopsis homologs of the yeast the NuA4 scaffold proteins EPL1 and EAF1. In knock-out plants, they observe dwarf pale green phenotypes connected to smaller mesophyll cell and chloroplast size and less chlorophyll. They use the Atepl1 mutants to carry out transcriptomic analysis, detecting photosynthesis-related genes downregulated and stress response genes upregulated. They perform a series of in-vitro experiments to confirm interaction with the HAT subunits HAM1 and HAM2 and to confirm the ability of the recombinant picNuA4 to acetylate H4. This is followed by ChIP-seq-based detection of reduced H4K5ac (but not K3K9ac) in the Atepl1 mutants and identification of genes that depend on NuA4 for their expression. Finally, the authors suggest that NuA4 dependent genes are marked by the presence of H2A.Z that is also acetylated by NuA4.

I appreciated that the manuscript is very well and carefully written and the experiments are well documented and described. The manuscript brings valuable novel information about the NuA4 complex in Arabidopsis, well combining biochemical, epigenomic and developmental evidence. What I think need to be improved is setting the introduction and interpretation in the context of what is known of the PLANT NuA4 complex. In addition, I don't find the evidence of the SPECIFIC requirement of NuA4 for the regulation of photosynthetic genes very convincing as the interpretation of the observed molecular and developmental phenotypes may be also compatible with the possibility that the effect on photosynthetic genes/developmental phenotypes is secondary but correlative. In this regard, I also raise a few questions/comments. This does not depreciate the value of the findings but rather questions the take-home message (and title) of the manuscript.

1. Multiple analogies are drawn from the yeast model and applied to interpret plant-model data. More plant-specific information about the NuA4 should be given in the introduction and interpretation made in context of plant-related knowledge of the NuA4.

- e.g. exclusive involvement of EAF1 and EPL1 in NuA4 is stated (Line 95-96) – this information however seems to relate to yeast models. How well is the exclusive involvement of these two subunits in plant NuA4 supported and can the statement be made considering this study uses plants as a model? Although some support is shown by the TAP-tag in-vivo pulldown presented here (results line 250 – 273, Table 1), the statement in the introduction may be misleading. This may be key in interpreting the developmental (and ultimately the molecular) phenotypes.
- 2. Approximately the same numbers of genes were up- and down-regulated in the Atepl1 mutants, which contrasts with the general function of NuA4 in transcriptional activation (that would presume predominating down-regulation if only primary effects were observed). This suggests that transcriptional effects are combined of primary and secondary effects. Photosynthesis is generally down-regulated under stress and stress-response categories are seen as enriched in the Atepl1 mutants. May downregulation of photosynthetic genes be in fact a secondary effect of elevated stress?
- 3. The pull-down and far-WB experiments use GFP or tags as negative controls. As all the tested interactions were positive, a negative control protein that would provide more support for the specificity of the interactions should be used.
- 4. It is not well explained why the two histone acetylation marks were selected to be profiled. Has this been preceded by establishing which OTHER modifications could be affected in the epl1-1 mutants, e.g. by WB?
 - Atepl1 mutants are affected by general depletion of H4K5ac (Fig. 5E), in line with the requirement for NuA4 for general H4K5ac establishment. Some sets of genes are identified as being specifically dependent on the NuA4 for their expression which include the photosynthesis-related genes. Some clarification is needed to support the conclusion.
 - The depletion of H4K5ac in Atepl1 mutants seems to be more pronounced on the down-regulated genes. Based on this observation, criteria for defining NuA4-dependent genes are set (Lines 331-334). Can it be however excluded that the more pronounced reduction of the modification on these genes is caused by their downregulation – i.e. that the reduction of the modification specifically at these genes would be a consequence of the transcription change rather than an indication of specific requirement of NuA4?
 - Can the authors demonstrate targeting of AtEPL1 to these genes?
- 5. It is unclear whether the H2A.Z profiling (ChIP-seq) data was generated here or used from another study – can you clarify? Is the material (growth/developmental stage of plants) the same as used for the H4K5ac profiling?
- 6. For the ChIP-qPCR experiments, a commercial H2A.Z antibody raised against mammalian H2A.Z is used, despite the fact that the authors point out the different N-terminus sequence (Fig S7) which was the reason for producing plant-specific H2A.Z for the in-vitro experiments (Fig 6G,H). How has the antibody functionality been tested in plants and can the ChIP experimental controls (IgG, positive target gene) be shown? Can you specify the number of replicates especially (separately) for the H2A.Z ChIP that seems to be of lower quality but should convey an important message?
- 7. Fig S5D and Line 385 – I do not see evidence from the metagene plot that the gene body of the “NuA4-dependent genes” would be depleted of H2A.Z and H4K5ac compared to “all expressed genes” (comparing blue and brown track).
- 8. (Line 223) The statement “HAM1 and HAM2 are widely accepted as ESA1 orthologs” is missing a reference.
- 9. The hypothesis formulation and message in lines 320-322 is unclear (what are “H4K5ac-reach genes”).
- 10. Please change commas for points in graphs.

Reviewer #3:

Remarks to the Author:

In the manuscript entitled “Nuclear-encoded photosynthesis genes are specifically controlled by the NuA4 complex and H2A.Z acetylation”, Bieluszewski and colleagues uncovered the profound impact of the conserved Nua4 histone H4 acetyltransferase complex on plant growth. They further explore its

impact on gene expression and chromatin status and found a propensity of nuclear genes encoding plastid proteins to be impacted by Nua4, which they propose to rely on histone H2A.Z acetylation. Most aspects in epigenomics and biochemical analyses of this work are of high quality, of great interest and appear timely in the field of plant research. They also provide new aspects on the function of the important and conserved Nua4 protein complex that is of general interest in the epigenomics/chromatin field of research. In my opinion the scope and the novelty of this study suit the editorial line of Nature Communications. I would recommend its publication provided that authors make satisfactory responses and revise the manuscript to address several weaker aspects or imprecisions detailed below.

More precisely, Bieluszewski and colleagues were able, for the first time, to generate a complete loss of function of the nucleosomal acetyltransferase of H4 (NuA4) activity in the model plant Arabidopsis, and used that genetic tool to investigate more its influence on chromatin modification and plant development. This was notably achieved through the use of CRISPR-Cas9 mutagenesis to produce null mutants for two scaffolding Nua4 subunits, AtEAF1 and AtEPL1, which were so far prevented by the presence of two tandem copies of AtEAF1 and the suspicion of non-null T-DNA alleles for AtEPL1 genes in previous studies. Those new mutant lines allowed establishing a more drastic role for Nua4 in plant growth and chloroplast development than anticipated, since other Nua4 mutants supposedly retained part of its function. Biochemical analyses are presented to confirm EAF1 and EPL1 as Nua4 subunits and verify their ability to acetylate nucleosomal histone as part of Nua4 complex. Searching by which mechanism Nua4 regulates growth and plastid development, the authors searched for transcription factors recruiting Nua4 to its target genes. Despite the Nua4 phenotypes observed being reminiscent of the ones observed for the well-known GLKs TFs involved in chloroplast retrograde signaling, the epistasis analysis and the transcriptomes performed concluded that GLKs and Nua4 function independently.

Nua4 is known to acetylate both H4 and H2A in different eukaryotic species. The authors first focus on H4K5ac and showed by quantitative ChIP-seq a very low level of the mark genome-wide in the absence of a functional Nua4.

Finally, to understand why, despite this global effect, specific sets of genes were affected transcriptionally upon Nua4 loss-of-function, the authors looked for specific chromatin signatures. They identify that downregulated genes have low levels of H3K9ac, suggesting that H3K9ac dynamics could stimulate transcription redundantly to H4K5ac. They also observe that the genes with expression level most strongly affected by a decrease in H4K5ac tend to be highly enriched in the H2A.Z histone variant. They tested whether plant NuA4 is capable of acetylating H2A.Z using ChIP-qPCR analysis of a few candidate genes, and observed a strong loss of H2A.Zac in the Atepl1 mutant on three downregulated genes but not on an unchanged gene. This led the authors to propose that acetylation of histone H4 and H2A.Z together influences gene expression.

Major comments:

The search for TFs recruiting Nua4 could have been investigated more in depth after GLKs being spotted as good candidates. For example, searching for cis-regulating motifs and TF binding sites in the promoter of the most affected genes (such as the plastidial ribosomal proteins npRPGs described in the manuscript as privileged targets of Nua4). Indeed, there are numerous TFs involved in the induction of plastidial genes, such as HY5/HYH, TCPs, BBXs, BZR1 etc. Maybe other potential TF recruiting Nua4 could be identified this way.

Also, Arabidopsis nuclear genes encoding cytosolic ribosomal proteins are thought to be commonly regulated using a specific DNA sequence motif known as Telobox, possibly through TRB transcription factor (see Gaspin et al, BMC Plant Biol 2010 doi: 10.1186/1471-2229-10-283). This also applies in yeast, where ribosomal protein genes are regulated both by Nua4 and by Rap1 that recognizes the telobox sequence motif in their promoter regions. Is-there any evidence that 1) this type of ribosomal protein genes of the cytosol and/or of plastids could similarly be regulated by Nua4 in Arabidopsis, and 2) that plastid localized npRPGs also display such a telobox motif linked to Nua4 co-regulation?

In that regard, the following question asked by the authors has not been addressed at all: "we thought to identify chromatin states that distinguish Nua4-dependent genes from other genes that were downregulated in Atepl1-2". The following section hints at the link between Nua4 and H2A.Z enrichment on +1 nucleosomes, but this co-occurrence is subsequently shown to be likely a consequence of Nua4 function on H2A.Z chromatin incorporation/acetylation. If the authors want to push in that direction, they should test whether H2A.Z enrichment is a pre-requisite for Nua4 recruitment. Hence, either the authors provide a ChIP-seq analysis of Nua4 chromatin landscape (in wild-type and possibly also in H2A.Z mutant plants) or the whole presentation of that question of Nua4 chromatin recruitment should be rearranged.

An important experimental aspect to fulfill is a better characterization of Nua4 effect on H2A.Z acetylation. While the ChIP-qPCR results on H2A.Zac are very promising, to really conclude that H2A.Zac defect is making the difference for triggering expression changes in Atepl1, a ChIP-seq of H2A.Zac in both genotypes is required. This would allow testing for a correlation between H4K5ac, H2A.Zac, mRNA level and photosynthetic or other GO functions at the genome wide scale and test whether the claims made by the authors are actually supported when tested more generally. This sounds critical for a manuscript having the current title. Also, as Nua4 is known to acetylate canonical H2A, a ChIP-seq of H2Aac in parallel is also required to establish a specific effect of H2A.Zac rather than indifferent acetylation of most H2A isoforms.

Given the transcriptomic defects identified, another important and easily amenable experiment is the determination of the phenotypic adaptation to light of Nua4 mutant plants during dark-to-light transitions. Typical photomorphogenic traits such as seedling hypocotyl size, apical hook and cotyledon development (size, angle) should be measured in autotrophic conditions and discussed with regard to other mutant plants affected in chromatin machineries or light signaling.

Is there any explanation why EPL1b is found similarly associated to HAM1 and HAM2 in the in vitro assay but HAM2 is not significantly detected in the in vivo EPL1b TAP assays?

Finally, as mentioned in the title, Nua4 is proposed to affect strongly nuclear-encoded photosynthesis genes (e.g., title and line 212). This is reasonable considering the strong phenotype of the mutants, and at the molecular level in the genomics analysis this is supported by a GO analysis. Yet, more details should be provided on light-harvesting components, plastid membrane proteins or composing the photosynthetic apparatus. Also, for the H2A.Zac aspect, only npRPGs are analyzed, generating two problems in my opinion:

- those plastidial ribosomal proteins are not "photosynthesis genes", as they do not function directly into photosynthesis, which is in disagreement with the title of the manuscript.
- a more systematic analysis of all kind of plastidial nuclear encoded genes in term of expression changes and histone acetylation changes could be conducted to support and maybe refine this idea.

Minor comments:

To my opinion, the biochemical analyses of the Nua4 complex subunits could come first in the manuscript along with panel 1A. This way, the conservation and position of the studied subunits in the complex would be established before diving in the phenotypes and chromatin impact of the different mutants.

Line 37 – 44: This paragraph presents a view that is "animal-centered". Plants have been a model of choice to enable seminal discoveries in the chromatin field (about transposition and about silencing through small RNAs as just some examples) and do not just serve as complementary species to test for discoveries previously made on animal models. Second, evolutionarily conserved enzymes were not "adopted" in plants, they first evolved in common ancestors and then either evolved similarly or differentially during the evolution of animal, yeast and plant or other eukaryotic organisms.

Line 45 – 57: 1) The meaning of the abbreviated names GCN5, HAF2, TAF1 BRM must be provided. 2) the field of research has been extensively abounded by dozens of studies over the last decade, a dimension that is hidden in the introduction. The authors could for example refer to recent reviews of the field such as:

- Plant Chromatin Catches the Sun.

Bourbousse et al Front Plant Sci. 2020. doi: 10.3389/fpls.2019.01728.

- Light in the transcription landscape: chromatin, RNA polymerase II and splicing throughout Arabidopsis thaliana's life cycle. Tognacca et al. Transcription. 2020 doi: 10.1080/21541264.2020.1796473.

- The impact of light and temperature on chromatin organization and plant adaptation.

Perrella G, et al 2020 J Exp Bot. doi: 10.1093/jxb/eraa154.

Line 54: there is no citation for BRM. Given the focus on photosynthetic genes, the authors could cite Jégu et al Genome Biol. 2017 doi: 10.1186/s13059-017-1246-7.

Line 82: About the statement that "we show that the loss of NuA4 integrity leads to a dramatic global decrease in H4K5 acetylation with almost no impact on H3K9ac and to a widespread disruption of nuclear transcription". What is meant by a defect in nuclear transcription? The study only assesses transcript steady-state levels, and do not present any investigation of transcription of single genes or more globally in whole nuclei.

Line 106: For my understanding, the construct targeting the downstream region of the Swi3, Ada2, N-Cor and TFIIIB (SANT) is not precisely described in the schematic representation of the protein. A better illustration would help the readership.

Line 142: the "prediction" that Nua4 loss-of-function mutations would lead to Arabidopsis lethality as in yeast species is farfetched, as most chromatin-based or epigenetic processes known to be essential in metazoan species can actually be knocked out in plants without triggering complete sterility or lethality during development. The mentions of Nua4 being specific in that way therefore appears overstated. For the precise case of transcription-coupled chromatin machineries, see Grasser et al, 2020 doi: 10.1016/j.bbagr.2020.194613.

Line 175: "of most, but not all" is quite imprecise. Is it implied a cooperation for chloroplast and less so for cell/organ size?

Line 182 – 183: The aggravation of the phenotype in the quadruple mutant is not aggravated in "all phenotypic assays" as explained in the next sequence, please rephrase.

Line 186 – 187: In none of the assays Nua4 and GLK mutants have opposite effect. The case of the a/b chlorophyll ratio is a suppression of the glk phenotype by the epl1 mutation, it is not an opposite effect as the epl1 alone has no phenotype in that assay. Please rephrase.

Line 295 – 298: A reduction of the strength of nucleosome–DNA interaction by H4 acetylation is not the only possible explanation for an increase in H3 occupancy in the mutant, H4 acetylation could destabilize nucleosomes or change their positioning or facilitate its eviction by remodelers. Plus, it could also result from H2A.Z hypoacetylation in the mutant and not H4 hypoacetylation. Also, the effect on H3 is along the whole gene body while H4k5ac is restricted mainly to the +1 nucleosome, indicating that this is not the sole explanation. This should be rephrased in a more hypothetical way or reference articles showing the effect of H4 acetylation on nucleosome–DNA interaction and nucleosome occupancy.

Line 322: rich instead of reach

Line 321 – 322: “the level of H4K5ac drops below a certain “threshold” level for genes downregulated more rapidly than for the H4K5ac-reach genes.” While I understand the arguments about the threshold that might explain the downregulation of genes in the mutant, I found the wording “drops more rapidly” confusing, given that authors do not examine a temporal process but compare steady state levels in different plant lines.

Line 344: I find the wording “Nua4-dependent genes” a bit confusing because what is done here is selecting, among the Nua4-dependent genes for H4K5ac, the genes that are impacted in term of transcription by Nua4 presumably because they don’t have a H3K9ac “back-up” system.

Line 354: rephrase “has been adopted to”.

Figure 6D and related text: This panel is not very informative as the fact that “Nua4 dependent genes have very high level of H4K5ac at the +1 nucleosome, differing from the broader group of all genes that were downregulated in Atepl1-2” might only reflect that those genes were chosen in the first place for having a LogFC of H4K5ac < -2, so highly marked genes were initially pre-selected.

Figure 6E and related text: It is not mentioned where the H2A.Z ChIP-seq data were retrieved.

Line 373: About the statement “ These results clearly demonstrate that the recruitment of TFIID via TAF1 bromodomains is not the primary mode of action of H4ac at NuA4-dependent loci in plants”: what is the evidence for a conservation between animals and plants of the interaction between TFIID and H4ac by the bromodomain of TAF1? I would down-tune the statement and condition the conclusion to future studies examining that predicted functional conservation.

Line 395 – 397: The higher level of H2A.Z in the mutant observed by ChIP-qPCR in figure 6F is neither discussed here nor in the discussion. Can we envision that more free SWC4 is available in Atepl1 due to the disruption of Nua4, and that could favor the formation of the SWR1 complex and so favor H2A.Z incorporation into chromatin?

Line 425 – 426: The present manuscript does not include any mention of a reproductive phenotype for Ateaf1-1. Even though this is out of the main scope, essential information on the Nua4 mutant plants life cycle is required.

Line 490 – 492: About the sentence “To the best of our knowledge, similar photosynthesis-related phenotypes are not observed when any of the chromatin factors involved in transcription control characterized to date are inactivated in plants.” This clearly appears as an over-statement. I acknowledge that a few of them have been described, but the authors cannot ignore and, for the sake of acknowledging prior work, they should cite at least 3 other Arabidopsis chromatin factors involved in transcription control influencing plant adaptation to light:

1) the histone acetyltransferase GCN5 (see for example Benhamed et al 2006 Plant Cell doi:

10.1105/tpc.106.043489) or a recent review in Grasser et al BBA 2020 doi:

10.1016/j.bbagrm.2020.194613);

2) HUB1-HUB2 histone H2B ubiquitin ligase that shares with Nua4 mutant plant some defects in pigment biogenesis and misregulation of numerous photosynthetic genes (see Bourbousse et al. 2012 PLoS Genet. doi:10.1371/journal.pgen.1002825);

3) SDG8 histone methyltransferase with similarly established global effects on photosynthetic genes (see Li et al. Genome Biology 2015 doi 10.1186/s13059-015-0640-2).

Line 497 – 499: Why not look at the list of plastidial and mitochondrial genes to see their enrichment in H4K5ac, H2A.Zac and misregulation in epl1 to see if they are particularly affected?

REVIEWER COMMENTS

Reviewer #1 (Remarks to the Author):

In this manuscript, Bielszowski, et al, explore the importance of the yet poorly understood connection between NuA4 acetylation and H2A.Z function in gene expression in Arabidopsis. The authors generate a nice repertoire of genetic mutants targeting two specific components of the putative NuA4 complex in Arabidopsis, the EAF1 and EPL1 proteins. Phenotypically, these mutants affect chloroplast development and resemble a mutation in GLK transcription factor involved in chloroplast gene activation. However this study falls short on some of the genomic and biochemical characterization listed below and requires further clarification/deeper characterization.

We thank the Reviewer for appreciating our work. In the revised version of the manuscript, we included most of the additional analyzes suggested by the reviewer, including genome-scale ChIP-seq and RNA-seq experiments.

General point:

Although the authors generate loss-of-function mutations for both EAF1 and EPL1, they only characterize EPL1 mutants in their genomic study. It is not exactly clear why. It would be nice to include EAF1 mutants in RNA-Seq and ChIP-Seq studies to better understand the specificity of the NuA4 complex.

The overall aim of this work was to understand the consequences of the loss of functionality of the NuA4 complex in plants at the transcriptional and physiological level. While the *Atepl1* and *Ateaf1* mutants differ significantly in yeast, our research shows that in plants these differences, if any, are minimal. Phenotypically, the *Atepl1* and *Ateaf1* mutants are practically indistinguishable (Fig. 1d and Fig. 2) clearly indicating that the NuA4 complex is inactive in both AtEPL1 and AEA1 absence. It should also be noted that working with both *Atepl1* and *Ateaf1* mutants is technically very difficult. Moreover, AtEAF1 is a very big protein of 212 kDa, which makes it unsuitable for biochemical approaches. Therefore, instead of superficial analyzing the mutants for both components, we decided to focus mainly on *Atepl1*, which allowed us to conduct a more detailed characterization.

However, to finally confirm that the observed developmental differences are also reflected at the molecular level, we performed comparative RNA-seq analyzes for the *Atepl1-2* and *Ateaf1-2* mutants. Both mutants showed very similar expression profiles: of 1,593 genes upregulated in *Ateaf1-1*, 1,212 (77.6%) were also upregulated in

Atepl1-2 (Fig. 3b). Similarly, 1,771 genes were downregulated in *Ateaf1-1*, 1,287 (72.7%) were downregulated also in *Atepl1-2* (Fig. 3b). This further supports our view that the differences between these two mutants are rather quantitative than qualitative.

RNA-Seq analysis:

The authors have performed RNA-Seq of the *Atepl1-1* mutant (Fig 3), but the ChIP-Seq has been done for the *Atepl1-2* allele. In the ChIP-Seq section the authors refer to RNA-seq of *Atepl1-2* allele. So is this just a typo, has RNA-Seq in Fig 3 been done for *Atepl1-2* and not for *Atepl1-1*? If a different allele has been used for RNA-Seq and ChIP-seq, than conclusions regarding transcriptional changes as a result of chromatin changes may not be appropriate.

Indeed, RNA-seq analysis was performed on the *Atepl1-1* allele while ChIP-seq analyzes were performed on the *Atepl1-2* allele, which is due to "historical" reasons: we were in possession of the *Atepl1-1* allele long before the *Atepl1-2* allele. Both alleles are indistinguishable phenotypically (Fig. 1 and 2). However, according to this reviewer's suggestion, we repeated the RNA-seq experiment on the *Atepl1-2* allele, thus ensuring that the conclusions drawn were relevant. In connection with the request to perform RNA-seq analyzes of *Ateaf1-1* and *glk* mutants, we present the RNA-seq results of *Atepl1-2* in the new version of the manuscript.

In Figure 3, the authors compare transcriptional misregulation of genes in *Atepl1-1* to that of GLK transcription factor, where very little overlap was observed and conclude that NuA4 and GLK work independently. However, *glk* RNA-seq is from publically available data in seedlings, while the *Atepl1-1* is done in rosette leaves. The RNA-Seq should be done side-by side to reach a convincing conclusion about the relationship of these mutants. Furthermore, the authors generated a *Atepl1 glk* double mutant which is viable. Therefore, in addition to the phenotypic characterization of the *Atepl1 glk* mutant, I recommend that the authors perform RNA-seq in parallel with the single mutants to clearly determine genetic relationship.

We thank the reviewer for this comment with which we agree. Therefore, we repeated RNA-seq experiments, growing all three genotypes (*Ateaf1-1*, *Atepl1-2* and *glk*) side-by-side. The results clearly show that while *Ateaf1-1* and *Atepl1-2* are very similar in terms of deregulated genes, the expression profiles of other genes changes in the *glk* mutant (Fig. 3b, c). In particular, only 121 out of the 4,144 genes falling under the GO "chloroplast" category, undergo simultaneous downregulation in *Atepl1-2* and *glk*, which is in line with our earlier observations. Importantly, npRPGs were strongly downregulated in both NuA4 mutants, while no change in their expression was observed for *glk* (Fig. 3c).

For technical reasons, we were not able to perform the experiment on the quadruple mutant *Atepl1-2 glk (Atepl1a Atepl1b glk1 glk2)*: apart from the fact that this mutant shows extremely retarded growth, obtaining this mutant requires selection from a heterozygote, which significantly reduces the availability of the material. However, we included RT-qPCR analysis for several photosynthesis-related genes, which shows a decrease of their expression in *Atepl1-2 glk* when compared to the *Atepl1-2* or *glk* double mutants (Fig. S5). Therefore, we think that the new data presented for the *Atepl1-2*, *Ateaf1-1*, and *glk* double mutants, together with the RT-qPCR results for the quadruple mutant, is strong enough to argue against NuA4 and GLKs acting in the same pathway.

Biochemistry:

Using in-vitro expression in bacteria of several key components of the Arabidopsis NuA4 complex, the authors show that AtEPL1 interacts with the NuA4 catalytic subunit HAM1. This interaction parallels and verifies IP-mass spec studies generated in this and previous studies. The authors also perform an in-vitro HAT activity assay of the picNuA4 components (HAM1, AtEPL1A, ING and AtEAF6) purified from bacteria, showing that it is capable of acetylating H4 and H2A histone particles in-vitro. This is rather an expected outcome, a more interesting experiment would be to determine if/how AtEPL1 is required for HAM1 function. For instance, could the authors perform HAT activity of the picNuA4 complex in the presence and absence of AtEPL1 or with certain mutant versions?

We agree with the reviewer that showing the dependence of picNuA4 acetylation on the presence of AtEPL1 would be an important result. Therefore, we prepared new genetic constructs to conduct this experiment, performed the purification of the complex in different combinations (to be able to restore the complex with and without AtEPL1) and performed new HAT assays (Fig. 4c-g and S8). We found that in the absence of AtEpl1, the AtHAM1-containing complex did not show any acetyltransferase activity towards short nucleosomes (At-FLAG), although it did show trace activity when the substrate was core histones. This result is consistent with what was observed in yeast (Boudreault et al., Genes & Dev. 2003). Interestingly, WB using anti-FLAG showed that in the absence of AtEPL1, the complex still contains similar if not greater amounts of AtHAM1. This suggests that, unlike yeast EPL1, plant EPL1 is not essential for HAM1 binding to the complex, but is essential for its acetyltransferase activity.

ChIP-Seq:

The authors generated ChIP-seq in triplicates for H3K9Ac, H4K5Ac and H3 and convincingly show that a major change in acetylation in *Atepl1-2* mutant involves a loss in H4K5Ac and not H3K9Ac. Even though majority of the *Atepl1-2* characterization in this study has been done in relation to H4K5Ac changes, the connection between NuA4 and H2A.Z is the most interesting conclusion of the paper (also reflected by the title), and therefore deserves a deeper characterization. The authors identify a class of *Atepl1-2* down-regulated and NuA4 dependent genes that contain high levels of H2AZ at +1 nucleosomes. Their acetylation is dependent on *Atepl1-2* activity at a subset of loci. It would be of high interest to generate genome-wide H2A.Z and H2A.Z Ac maps in *Atepl1-2* mutant to better understand the relationship of NuA4 acetylation and H2A.Z. In a previous study involving the characterization of YAF9A/9B, a putative NuA4 components in Arabidopsis, Crevillen, P et al, 2019

concluded that at FLC locus, YAF9A/B are required for H2A.Z acetylation but not deposition. The data presented in his paper also suggests similar conclusion (Fig. 6). This is in contrast to models presented in yeast (Lu, P, Y, T et al, 2009). Exploring this relationship genome-wide will capture an interest of wider audience as it may highlight interesting differences between NuA4-H2A.Z connection in plants and other eukaryotic organisms.

In connection with the reviewer's request, we performed genome-wide ChIP-seq analyzes for total H2A.Z and H2A.Zac in the wild-type background and the *Atepl1-2* mutant. The results for ChIP-seq total H2A.Z which were normalized using spike-in showed that the incorporation of this histone variant is limited in the *Atepl1-2* mutant (Fig. 5 & 6). We performed additional ChIP-qPCR control experiments which also showed a significant decrease in the H2A.Z deposition (Fig. S10). We have revised our previous results in this regard, which were probably caused by too mild washes following ChIP. It should be noted that the available antiHTA9 antibody shows little enrichment and is therefore technically difficult to use. We are now convinced that, similarly to yeast and mammals, reducing the activity of the NuA4 complex causes a significant decrease in the amount of H2A.Z incorporated into chromatin. This result does not contradict the result cited by the reviewer for the *Atyaf9* mutant as YAF9 is only a helper protein for the SWR1 and NuA4 complexes.

New ChIP-seq analyzes confirmed our earlier observations that NuA4-dependent genes have significantly higher levels of the total H2A.Z in +1 nucleosome than the other gene groups. Even more intriguingly, we found that genes upregulated in the *Atepl1-2* mutant show high levels of the H2A.Z in their gene body regions when measured in the WT plants (new Fig. 8). Previously, we and others showed that the gene body H2A.Z represses gene expression in plants (Coleman-Derr & Zilberman, PLoS Genet. 2012; Sura et al. Plant Cell 2017; Dai et al. Mol Plant 2017), and by comparing our data with that for the mutant of PIE1, a catalytic subunit of SWR1C, we proposed that H2A.Z loss from gene body regions is a cause of the gene upregulation observed in *Atepl1-2*. Taken together, our new ChIP-seq data provide a comprehensive explanation for gene dysregulation observed in the NuA4 mutants.

Furthermore, the connection between NuA4 and H2A.Z could be strengthened genetically. It would be interesting to see the effect of H2A.Z deposition, acetylation and gene expression in higher order mutants of *Atepl1-2* crossed with SWR1 mutants. For instance, MBD9, a newly characterized component of the SWR1 complex (Sijacic, P et al, 2019, Potok, M et al, 2019), has been shown to be specifically required for H2A.Z deposition at genes with H2A.Z localized to +1 nucleosome. So presumably these may be the same sets of genes that are NuA4 dependent identified in this study. In general, in MBD9 mutant, the reduction in H2AZ at +1 nucleosome does not lead to drastic transcriptional changes. This was similarly observed in this study upon depletion of H4K5Ac. It would be interesting to see if removal of H2AZ along with NuA4 dependent acetylation could reveal some redundancies in the control of gene expression.

We agree with the reviewer that it would be interesting to study the genetic interaction between the *Atepl1-2* mutant and the mutants of genes involved in H2A.Z deposition. Therefore, we crossed the double *Atepl1-2* with *arp6* and *mbd9* to obtain heterozygotes. These genes are not linked, therefore in the F₂ generation the expected number of homozygous triple mutants is 1/64 ($1/4 \times 1/4 \times 1/4$). Despite genotyping 285 and 267 plants for *xarp6* and *xmbd9* respectively, we did not find a single triple mutant. From the 2.5x mutants obtained (with only *Atepl1a* in a heterozygous state) we sowed seeds of the F₃ generation, but we did not obtain triple mutants also in this generation (see Table S2 for details). Therefore, we concluded that the *Atepl1 arp6* and *Atepl1 mbd9* mutants are synthetically lethal, presumably due to hyperactivation of stress genes caused by a complete lack of H2A.Z.

Minor point:

Regarding some of the H3K9Ac conclusions, it's not clear how the authors came to the following conclusion: "H3K9ac levels were significantly lower among the genes that were downregulated in *Atepl1-2* than among all expressed genes (..Fig 5D)". The levels of H3K9Ac seem to be the same in WT and *Atepl1-2* mutant when looking at the y-axis. Perhaps this point can be explained more clearly.

Here we were referring to the comparison of H3K9ac levels measured in the wild-type plants, and not to the comparison between the wild-type and the mutant. We apologize that this has not been made clear enough in this paper, we have tried to improve it in the revised manuscript.

Reviewer #2 (Remarks to the Author):

The manuscript by Tomasz Bieluszewski and co-authors describes a novel role for the plant NuA4 HAT complex in controlling photosynthesis genes through H4K5ac and H2A.Z acetylation in Arabidopsis. The authors generate series of CRISPR-Cas9 mutants affecting Arabidopsis homologs of the yeast NuA4 scaffold proteins EPL1 and EAF1. In knock-out plants, they observe dwarf pale green phenotypes connected to smaller mesophyll cell and chloroplast size and less chlorophyll. They use the *Atepl1* mutants to carry out transcriptomic analysis, detecting photosynthesis-related genes downregulated and stress response genes upregulated. They perform a series of in-vitro experiments to confirm interaction with the HAT subunits HAM1 and HAM2 and to confirm the ability of the recombinant picNuA4 to acetylate H4. This is followed by ChIP-seq-based detection of reduced H4K5ac (but not H3K9ac) in the *Atepl1* mutants and identification of genes that depend on NuA4 for their expression. Finally, the authors suggest that NuA4 dependent genes are marked by the presence of H2A.Z that is also acetylated by NuA4.

I appreciated that the manuscript is very well and carefully written and the experiments are well documented and described. The manuscript brings valuable novel information about the NuA4 complex in Arabidopsis, well combining biochemical, epigenomic and developmental evidence. What I think need to be improved is setting the introduction and interpretation in the context of what is known of the PLANT NuA4 complex. In addition, I don't find the evidence of the SPECIFIC requirement of NuA4 for the regulation of photosynthetic genes very convincing as the interpretation of the observed molecular and developmental phenotypes may be also compatible with the possibility that the effect on photosynthetic genes/developmental phenotypes is secondary but correlative. In this regard, I also raise a few questions/comments. This does not depreciate the value of the findings but rather questions the take-home message (and title) of the manuscript.

We are pleased that the reviewer appreciates the research presented in the study. In the revised version of the manuscript, we included, among other things, the whole genome ChIP-seq analyzes for the total H2A.Z and H2A.Zac, which we believe allow us to present an even more complete picture of the NuA4 function in plants.

1. Multiple analogies are drawn from the yeast model and applied to interpret plant-model data. More plant-specific information about the NuA4 should be given in the introduction and interpretation made in context of plant-related knowledge of the NuA4.
– e.g. exclusive involvement of EAF1 and EPL1 in NuA4 is stated (Line 95-96) – this information however seems to relate to yeast models. How well is the exclusive involvement of these two subunits in plant NuA4 supported and can the statement be made considering this study uses plants as a model? Although some support is shown by the TAP-tag in-vivo pulldown presented here (results line 250 – 273, Table 1), the statement in the introduction may be misleading. This may be key in interpreting the developmental (and ultimately the molecular) phenotypes.

We thank the reviewer for this comment. We modified significantly both the introduction and the discussion part of the manuscript to meet these suggestions. Specifically, we added more information about the plant NuA4 complex composition (line 52-64).

Unlike many other subunits, AtEAF1A/B and AtEPL1A/B were not copurified with any other chromatin remodeling or chromatin modifying complex. In contrast, AtTRAP and numerous auxiliary proteins, including AtSWR4, AtARP4, AtYAF9, AtEAF1, AtING2, ACT1 were found in AP-MS/MS analyzes in SAGA, SWI/SNF, SWR1 or INO80 complexes. Therefore, it seems unlikely that AtEAF1 and AtEPL1 will function in other complexes, especially as they are assembly platforms. This issue is quite well discussed in the recent in-depth review of Espinosa-Cores et al. (Front. Plant Sci., 2020), which we refer to in our manuscript. In mammals, the counterparts of the NuA4 and SWR1C complexes are physically and functionally linked to form TIP60-p400. There is no evidence that this physical association also occurs in plants, although our new results presented in the revised manuscript clearly show a functional association of SWR1C with NuA4 in plants (similar to yeast). Given all of this, it seems safe to assume that AtEPL1 and AtEAF1 are key to understand the NuA4 function in plants.

2. Approximately the same numbers of genes were up- and down-regulated in the *Atepl1* mutants, which contrasts with the general function of NuA4 in transcriptional activation (that would presume predominating down-regulation if only primary effects were observed). This suggests that transcriptional effects are combined of primary and secondary effects. Photosynthesis is generally down-regulated under stress and stress-response categories are seen as enriched in the *Atepl1* mutants. May downregulation of photosynthetic genes be in fact a secondary effect of elevated stress?

Initially, we were also surprised by the large number of genes upregulated in the *Atepl1* and *Ateaf1* mutants, which seemingly contrasts with the general function of NuA4 as a transcriptional coactivator. The newly included spike-in normalized ChIP-seq analyzes for both total H2A.Z and acetylated H2A.Z and show that the genes upregulated in NuA4 mutants have very high levels of unacetylated H2A.Z in their gene body regions (Fig. 8a). Furthermore, we found that most of H2A.Z is lost from chromatin in the *Atepl1-2* mutant (Fig. 5 & 6). Our previous research as well as data from others have indicated that H2A.Z in gene body helps to suppress the expression of responsive genes under normal (non-stress) conditions (Coleman-Derr & Zilberman, PLoS Genet. 2012; Sura et al. Plant Cell, 2017; Dai et al. Mol Plant, 2017). Thus, we argue that this reduction in gene body H2A.Z is the main cause of gene upregulation in the NuA4 mutants. This hypothesis was further supported by our new extensive comparisons with the mutant of PIE1, a catalytic subunit of SWR1C, which is deficient in H2A.Z deposition (Fig. 8c-e, Fig. S14). Based on these findings we propose that the NuA4 is required to H2A.Z deposition, and in gene body regions H2A.Z is deacetylated and likely ubiquitinated by other enzymes, leading to their transcriptional suppression (Fig. 9b).

On the other hand, clear differences in chromatin profiles were also observed for genes downregulated in *Atepl1-2* that we defined as NuA4-dependent when compared to all expressed genes (high H2A.Z and H2A.Zac/H4K5ac occupancy in the +1 nucleosome, dramatic decrease in H2A.Zac/H4ac in the *Atepl1-2* mutant; Fig. 7, Fig. S13). Also, the reduction of chloroplast size is clearly cell autonomous as shown in our analyzes of *Ateaf1* mosaic plants (Fig. 1c), while stress responses are generally cell non-autonomous. Therefore, we conclude that the chloroplast-related phenotypes observed in our NuA4 mutants are at least partially independent on stress signaling induced in these mutants (Fig. 9a). We propose that NuA4, by controlling both the acetylation and deposition of H2A.Z, contributes to the balance between the stress response and the autotrophic growth (Fig. 9). We predict that future studies may reveal how the regulation of NuA4 activity balances growth and stress responses in plants.

3. The pull-down and far-WB experiments use GFP or tags as negative controls. As all the tested interactions were positive, a negative control protein that would provide more support for the specificity of the interactions should be used.

The GFP and GST proteins have been used in our protein-protein interaction assays as negative controls. They meet all the criteria for the negative control: they are proteins and are not expected to interact specifically with the NuA4 complex proteins. Both of these proteins are commonly used as negative controls in this type of experiment.

4. It is not well explained why the two histone acetylation marks were selected to be profiled. Has this been preceded by establishing which OTHER modifications could be affected in the *epl1-1* mutants, e.g. by WB?

We thank the reviewer for this remark and agree that the choice of modifications for profiling should be better documented. In the revised manuscript, we included the ChIP-qPCR experiments that were used to select the tested modifications. As NuA4 is by definition a histone H4 acetyltransferase, we studied H4K5ac, H4K8ac, H4K12ac and H4K16ac. All these modifications produced a similar profile for the tested genes and showed a strong relationship of AtEPL1 (Fig. S10), however the highest enrichment was observed for H4K5ac, hence our choice. We decided that we needed to profile an additional histone acetylation, which is NuA4-independent. Our choice was H3K9ac, which is the best characterized form of histone acetylation when it comes to transcriptional regulation in plants. H3K9ac is dependent on the second key histone acetyltransferase, i.e. the SAGA complex.

- *Atepl1* mutants are affected by general depletion of H4K5ac (Fig. 5E), in line with the requirement for NuA4 for general H4K5ac establishment. Some sets of genes are identified as being specifically dependent on the NuA4 for their expression which include the photosynthesis-related genes. Some clarification is needed to support the conclusion.

We believe that the conclusions regarding the NuA4-dependent genes are better presented in the revised version of the manuscript.

- The depletion of H4K5ac in *Atepl1* mutants seems to be more pronounced on the down-regulated genes. Based on this observation, criteria for defining NuA4-dependent genes are set (Lines 331-334). Can it be however excluded that the more pronounced reduction of the modification on these genes is caused by their downregulation – i.e. that the reduction of the modification specifically at these genes would be a consequence of the transcription change rather than an indication of specific requirement of NuA4?

The reviewer touches on the long-lasting debate as to whether histone acetylation is the cause or the effect of gene transcriptional activity. Unfortunately, it cannot be resolved using modern research methods. However, our new results regarding the comparison between genes upregulated in *Atepl1-2* and those upregulated in *pie1* seem to support the hypothesis that some level of histone acetylation at the nucleosome +1 is necessary for transcriptional activation: As genes upregulated in *pie1* and *Atepl1-2* show very high levels of gene body H2A.Z as measured in the WT plants (Fig. 8d), we assume that in *pie1* and *Atepl1-2*, the loss of H2A.Z allows them to be transcribed. However, not all genes upregulated in *pie1* are also upregulated in *Atepl1-2*: the difference lies in the level of histone acetylation in the nucleosome +1. The genes upregulated in *Atepl1-2* have a relatively high level of the acetylation in +1, or the level of H3 acetylation increases significantly in the background of the *Atepl1-2* mutant. We do not see such a relationship in the *pie1* mutant, where there is no decrease in H4ac/H2A.Zac acetylation (Fig. 8d).

- Can the authors demonstrate targeting of AtEPL1 to these genes?

We agree with the reviewer that such an experiment would be important. Therefore, we generated the transgenic lines carrying AtEPL1A-GFP, AtEPL1A-6xmyc or AtEPL1A-TAP fusions under endogenous promoters. All these lines complemented the *Atepl1-2* mutant phenotype which proved that they were functional. Based on the material isolated from these lines, we conducted ChIP-qPCR experiments, which, however, were unsuccessful. Only for AtEPL1A-GFP we obtained some enrichment in complemented line (included below FYI), however as the enrichment was very small and a strong unspecific signal was obtained also for TA3 transposon, we resigned from including this result in the ms. We think that AtEPL1 does not interact directly with the chromatin, and the interaction of the entire complex is very transient, which together with very low expression of AtEPL1 under endogenous promoters causes the ChIP experiments being very challenging.

5. It is unclear whether the H2A.Z profiling (ChIP-seq) data was generated here or used from another study – can you clarify? Is the material (growth/developmental stage of plants) the same as used for the H4K5ac profiling?

Initially we used our previously published data (Sura et al., Plant Cell 2017) produced from the material at the same developmental stage (rosette leaves), grown in similar conditions but extracted using MNase. However, due to the doubts of this and other reviewers, to what extent these data can be matched, the new version of the manuscript contains the ChIP-seq results for total H2A.Z and H2A.Zac for the wild-type and the *Atepl1-2* mutant under the same conditions, and performed in the very same way including spike-in controls as other analyzes ChIP-seq.

6. For the ChIP-qPCR experiments, a commercial H2A.Zac antibody raised against mammalian H2A.Z is used, despite the fact that the authors point out the different N-terminus sequence (Fig S7) which was the reason for producing plant-specific H2A.Z for the in-vitro experiments (Fig 6G,H). How has the antibody functionality been tested in plants and can the ChIP experimental controls (IgG, positive target gene) be shown? Can you specify the number of replicates especially (separately) for the H2A.Zac ChIP that seems to be of lower quality but should convey an important message?

For the experiments presented in this work, we used an antibody that had previously been thoroughly checked for specificity for H2A.Zac in *A. thaliana* (Crevillen et al. New Phytologist 2019 doi: 10.1111 / nph.15737; Fig. S12). We additionally performed a WB to see if the antibody recognizes HTA9ac, the major H2A.Z isoform in *A. thaliana*. This is now included in the manuscript as Fig. S8c). As the antibody specifically recognizes HTA9ac, we used it in ChIP-qPCR and ChIP-seq analyses.

In all ChIP-qPCR experiments presented in this paper, we used at least 3 biological replicates (with three technical replicates each). This antibody gives much lower enrichment than H4K5ac, therefore for the ChIP-seq analysis we used two replicates.

7. Fig S5D and Line 385 – I do not see evidence from the metagene plot that the gene body of the “NuA4-dependent genes” would be depleted of H2A.Z and H4K5ac compared to “all expressed genes” (comparing blue and brown track).

This fragment has been changed and the conclusion has been removed. However, we used the term “gene body” to describe the profile of a given modification along the gene outside the +1 nucleosome (the +1 nucleosome is analyzed separately, as it is usually much better positioned than other nucleosomes and, being the main barrier for RNA polymerase II, it often behaves differently). As genes differ in length, the DANPOS software that we applied to analyze ChIP-seq results, uses “percentage” instead of “position in kb” when plotting gene body enrichment. For boxplot visualization of gene body enrichment, we used the segment between 50% and 75% on the X axis.

8. (Line 223) The statement “HAM1 and HAM2 are widely accepted as ESA1 orthologs” is missing a reference. We added the lacking reference.

9. The hypothesis formulation and message in lines 320-322 is unclear (what are “H4K5ac-reach genes”). This has been rephrased.

10. Please change commas for points in graphs.

We changed as requested.

Reviewer #3 (Remarks to the Author):

In the manuscript entitled “Nuclear-encoded photosynthesis genes are specifically controlled by the NuA4 complex and H2A.Z acetylation”, Bieluszewski and colleagues uncovered the profound impact of the conserved Nua4 histone H4 acetyltransferase complex on plant growth. They further explore its impact on gene expression and chromatin status and found a propensity of nuclear genes encoding plastid proteins to be impacted by Nua4,

which they propose to rely on histone H2A.Z acetylation. Most aspects in epigenomics and biochemical analyses of this work are of high quality, of great interest and appear timely in the field of plant research. They also provide new aspects on the function of the important and conserved Nua4 protein complex that is of general interest in the epigenomics/chromatin field of research. In my opinion the scope and the novelty of this study suit the editorial line of Nature Communications. I would recommend its publication provided that authors make satisfactory responses and revise the manuscript to address several weaker aspects or imprecisions detailed below.

We thank the reviewer for appreciating our work.

More precisely, Bieluszewski and colleagues were able, for the first time, to generate a complete loss of function of the nucleosomal acetyltransferase of H4 (NuA4) activity in the model plant *Arabidopsis*, and used that genetic tool to investigate more its influence on chromatin modification and plant development. This was notably achieved through the use of CRISPR-Cas9 mutagenesis to produce null mutants for two scaffolding Nua4 subunits, AtEAF1 and AtEPL1, which were so far prevented by the presence of two tandem copies of AtEAF1 and the suspicion of non-null T-DNA alleles for AtEPL1 genes in previous studies. Those new mutant lines allowed establishing a more drastic role for Nua4 in plant growth and chloroplast development than anticipated, since other Nua4 mutants supposedly retained part of its function. Biochemical analyses are presented to confirm EAF1 and EPL1 as Nua4 subunits and verify their ability to acetylate nucleosomal histone as part of Nua4 complex.

Searching by which mechanism Nua4 regulates growth and plastid development, the authors searched for transcription factors recruiting Nua4 to its target genes. Despite the Nua4 phenotypes observed being reminiscent of the ones observed for the well-known GLKs TFs involved in chloroplast retrograde signaling, the epistasis analysis and the transcriptomes performed concluded that GLKs and Nua4 function independently. Nua4 is known to acetylate both H4 and H2A in different eukaryotic species. The authors first focus on H4K5ac and showed by quantitative ChIP-seq a very low level of the mark genome-wide in the absence of a functional Nua4.

Finally, to understand why, despite this global effect, specific sets of genes were affected transcriptionally upon Nua4 loss-of-function, the authors looked for specific chromatin signatures. They identify that downregulated genes have low levels of H3K9ac, suggesting that H3K9ac dynamics could stimulate transcription redundantly to H4K5ac. They also observe that the genes with expression level most strongly affected by a decrease in H4K5ac tend to be highly enriched in the H2A.Z histone variant. They tested whether plant NuA4 is capable of acetylating H2A.Z using ChIP-qPCR analysis of a few candidate genes, and observed a strong loss of H2A.Zac in the *Atepl1* mutant on three downregulated genes but not on an unchanged gene. This led the authors to propose that acetylation of histone H4 and H2A.Z together influences gene expression.

Major comments:

The search for TFs recruiting Nua4 could have been investigated more in depth after GLKs being spotted as good candidates. For example, searching for cis-regulating motifs and TF binding sites in the promoter of the most affected genes (such as the plastidial ribosomal proteins npRPGs described in the manuscript as privileged targets of Nua4). Indeed, there are numerous TFs involved in the induction of plastidial genes, such as HY5/HYH, TCPs, BBXs, BZR1 etc. Maybe other potential TF recruiting Nua4 could be identified this way.

We thank the reviewer for this remark. In fact, we have performed very thorough analyzes of the cis-regulatory motifs for the genes defined by us as NuA4-dependent. We used MEME, *de novo* DNA motif search algorithm, to test for sequences enriched within 500-bp next to TSS. This resulted in identification of a long A/G-rich and a shorter GCCCA motifs in 228 (66%) and 116 (33%) of NuA4-dependent genes ($n = 347$), respectively. Interestingly, the GCCCA motif corresponds to DNA sequence recognized by TCP transcription factors (Li et al., PNAS 2005). However, when we performed a comparison with all genes expressed in wild type plants, none of those motifs turned to be significantly enriched in NuA4-dependent genes. For this reason, we resigned from including this analysis in the manuscript.

However, as the reviewer will see later in our responses, the new analyzes incorporated into the manuscript, especially those suggested by this reviewer, show a much more complete picture of the NuA4 functionality. In this new light also the genes upregulated in NuA4 seem to be directly controlled by this complex, and the transcriptional activity of the individual genes can be largely explained by the combination of the individual chromatin states.

Also, *Arabidopsis* nuclear genes encoding cytosolic ribosomal proteins are thought to be commonly regulated using a specific DNA sequence motif known as Telobox, possibly through TRB transcription factor (see Gaspin et al, BMC Plant Biol 2010 doi: 10.1186/1471-2229-10-283). This also applies in yeast, where ribosomal protein genes are regulated both by Nua4 and by Rap1 that recognizes the telobox sequence motif in their promoter regions. Is-there any evidence that 1) this type of ribosomal protein genes of the cytosol and/or of plastids could similarly be regulated by Nua4 in *Arabidopsis*, and 2) that plastid localized npRPGs also display such a telobox motif linked to Nua4 co-regulation?

Since in our analysis the Telobox motif is not enriched in NuA4-dependent genes, we have no grounds to argue that such regulation takes place. Additionally, we compared motifs present in the 5'UTR of cytoplasmic ribosomal protein genes with that of plastid ribosomal protein genes. In the first group we found the GGGCY motif that is probably recognized by TCP transcription factors (Li et al. PNAS 2005). However, no specific motif was revealed in the analysis of plastid ribosomal protein genes.

In that regard, the following question asked by the authors has not been addressed at all: "we thought to identify chromatin states that distinguish NuA4-dependent genes from other genes that were downregulated in *Atepl1-2*". The following section hints at the link between Nua4 and H2A.Z enrichment on +1 nucleosomes, but this co-occurrence is subsequently shown to be likely a consequence of Nua4 function on H2A.Z chromatin incorporation/acetylation. If the authors want to push in that direction, they should test whether H2A.Z enrichment is a pre-requisite for Nua4 recruitment. Hence, either the authors provide a ChIP-seq analysis of Nua4 chromatin landscape (in wild-type and possibly also in H2A.Z mutant plants) or the whole presentation of that question of Nua4 chromatin recruitment should be rearranged.

In light of the new results, all work has been significantly revised as explained in a more detailed way below.

An important experimental aspect to fulfill is a better characterization of Nua4 effect on H2A.Z acetylation. While the ChIP-qPCR results on H2A.Zac are very promising, to really conclude that H2A.Zac defect is making the difference for triggering expression changes in *Atepl1*, a ChIP-seq of H2A.Zac in both genotypes is required. This would allow testing for a correlation between H4K5ac, H2A.Zac, mRNA level and photosynthetic or other GO functions at the genome wide scale and test whether the claims made by the authors are actually supported when tested more generally. This sounds critical for a manuscript having the current title. Also, as Nua4 is known to acetylate canonical H2A, a ChIP-seq of H2Aac in parallel is also required to establish a specific effect of H2A.Zac rather than indifferent acetylation of most H2A isoforms.

As suggested by the reviewer, we performed genome-wide ChIP-seq analyzes for H2A.Z and H2A.Zac. Due to the lack of available antibodies for H2Aac, we were not able to conduct such an experiment, but the high correlation between the H4K5ac and H2A.Zac profiles suggests that this modification would largely follow H4K5ac.

The results for ChIP-seq total H2A.Z which were normalized using spike-in showed that the incorporation of this histone variant is limited in the *Atepl1-2* mutant (Fig. 5 & 6). We performed additional ChIP-qPCR control experiments which also showed a significant decrease in the H2A.Z deposition (Fig. S9). We have revised our previous results in this regard, which were probably caused by too mild washes following ChIP. It should be noted that the available antiHTA9 antibody shows little enrichment and is therefore technically difficult to use. We are now convinced that, similarly to yeast and mammals, reducing the activity of the NuA4 complex causes a significant decrease in the amount of H2A.Z incorporated into chromatin.

New ChIP-seq analyzes confirmed our earlier observations that NuA4-dependent genes have significantly higher levels of the total H2A.Z in the +1 nucleosome than the other gene groups. Even more intriguingly, we found that genes upregulated in the *Atepl1-2* mutant show high levels of H2A.Z in their gene body regions when measured in the wild type plants (new Fig.8 and S14). The gene body H2A.Z was shown to repress gene expression in plants (eg. Coleman-Derr & Zilberman, PLoS Genet. 2012; Sura et al. Plant Cell 2017, Dai et al. Mol Plant 2017), and by comparing our data with that for the mutant of PIE1, a catalytic subunit of SWR1C, we proposed that H2A.Z loss from gene body regions is a cause of gene upregulation observed in *Atepl1-2*. Taken together, our new ChIP-seq data provide a comprehensive explanation for gene dysregulation observed in the NuA4 mutants.

Given the transcriptomic defects identified, another important and easily amenable experiment is the determination of the phenotypic adaptation to light of Nua4 mutant plants during dark-to-light transitions. Typical photomorphogenic traits such as seedling hypocotyl size, apical hook and cotyledon development (size, angle) should be measured in autotrophic conditions and discussed with regard to other mutant plants affected in chromatin machineries or light signaling.

As suggested, we measured the hypocotyl length, apical curvature, and cotyledon development (surface and angle) after 0, 6, 24 and 56 hours following dark-to-light transition. We expected that similarly to other histone acetyltransferase mutants, we will observe elongated hypocotyls when grown in light (Benhamed et al., Plant Cell 2006; Bertrand et al., J. Bioch. Chem. 2005). Surprisingly, *Atepl1-2* showed reduced hypocotyl length and cotyledon surface (Fig. S4). However, our NuA4 mutants has very strong growth phenotypes when compared to the abovementioned mutants and as this could affect the photomorphogenic phenotype, we were careful to draw far-reaching conclusions. It is likely that this unexpected photomorphogenic phenotype reflects the abovementioned changes in the gene body H2A.Z.

Is there any explanation why EPL1b is found similarly associated to HAM1 and HAM2 in the in vitro assay but HAM2 is not significantly detected in the in vivo EPL1b TAP assays?

We do not have RNA-seq data for our *AtEpl1b*-TAP assay, however our RNA-seq results from rosette leaves show almost three times higher expression of HAM1 than HAM2. Perhaps, then, the difference in the expression of both genes is relevant here.

Finally, as mentioned in the title, Nua4 is proposed to affect strongly nuclear-encoded photosynthesis genes (e.g., title and line 212). This is reasonable considering the strong phenotype of the mutants, and at the molecular level in the genomics analysis this is supported by a GO analysis. Yet, more details should be provided on light-harvesting components, plastid membrane proteins or composing the photosynthetic apparatus. Also, for the H2A.Zac aspect, only npRPGs are analyzed, generating two problems in my opinion:
- those plastidial ribosomal proteins are not "photosynthesis genes", as they do not function directly into photosynthesis, which is in disagreement with the title of the manuscript.

- a more systematic analysis of all kind of plastidial nuclear encoded genes in term of expression changes and histone acetylation changes could be conducted to support and maybe refine this idea.

We agree with the reviewer that a deeper analysis of the genes encoding proteins directly related to photosynthesis is needed. The paper contains now an analysis of functional enrichment and relative expression of the NuA4-dependent genes (Fig. S7). In the new version, we have also added an extensive analysis of the chromatin states for individual GO categories with a breakdown into individual chloroplast compartments, as well as a comparison of cytoplasmic and plastid ribosomal protein genes (Fig. S7). These results confirm that a high level of H2A.Zac at +1 is the main determinant of NuA4-dependency. Most plastid genes including plastid ribosomal proteins, unlike cytoplasmic ribosomal protein genes, are unable to increase the level of H3K9ac in the *Atepl1-2* background. This may be why this gene group is overrepresented in the NuA4-dependent gene class.

Minor comments:

To my opinion, the biochemical analyses of the Nua4 complex subunits could come first in the manuscript along with panel 1A. This way, the conservation and position of the studied subunits in the complex would be established before diving in the phenotypes and chromatin impact of the different mutants.

While we agree with the reviewer that biochemical analyzes could be early on, they only complement the genetic analysis that was at the center of our research. The unexpected phenotype of the mutants we obtained was the starting point for all of our work. Therefore, we decided to stick to our original manuscript structure.

Line 37 – 44: This paragraph presents a view that is “animal-centered”. Plants have been a model of choice to enable seminal discoveries in the chromatin field (about transposition and about silencing through small RNAs as just some examples) and do not just serve as complementary species to test for discoveries previously made on animal models. Second, evolutionarily conserved enzymes were not “adopted” in plants, they first evolved in common ancestors and then either evolved similarly or differentially during the evolution of animal, yeast and plant or other eukaryotic organisms.

We thank the reviewer for this comment, with which we fully agree. Hence this paragraph has been amended in the revision of the manuscript.

Line 45 – 57: 1) The meaning of the abbreviated names GCN5, HAF2, TAF1 BRM must be provided. Corrected as requested.

2) the field of research has been extensively abounded by dozens of studies over the last decade, a dimension that is hidden in the introduction. The authors could for example refer to recent reviews of the field such as:

- Plant Chromatin Catches the Sun.

Bourbousse et al Front Plant Sci. 2020. doi: 10.3389/fpls.2019.01728.

- Light in the transcription landscape: chromatin, RNA polymerase II and splicing throughout Arabidopsis thaliana's life cycle. Tognacca et al. Transcription. 2020 doi: 10.1080/21541264.2020.1796473.

- The impact of light and temperature on chromatin organization and plant adaptation.

Perrella G, et al 2020 J Exp Bot. doi: 10.1093/jxb/eraa154.

Although we were limited with space, we tried to emphasize the relationship between light and chromatin, including the suggested references, in the introduction.

Line 54: there is no citation for BRM. Given the focus on photosynthetic genes, the authors could cite Jégu et al Genome Biol. 2017 doi: 10.1186/s13059-017-1246-7.

The reference added as requested.

Line 82: About the statement that “we show that the loss of NuA4 integrity leads to a dramatic global decrease in H4K5 acetylation with almost no impact on H3K9ac and to a widespread disruption of nuclear transcription”.

What is meant by a defect in nuclear transcription? The study only assesses transcript steady-state levels, and do not present any investigation of transcription of single genes or more globally in whole nuclei.

We agree with the reviewer – this has been amended.

Line 106: For my understanding, the construct targeting the downstream region of the Swi3, Ada2, N-Cor and TFIIIB (SANT) is not precisely described in the schematic representation of the protein. A better illustration would help the readership.

First of all, we would like to apologize for a writing error through which the order of words “downstream” and “region” has been reversed (should be “region downstream of the SANT domain”, line 109). As shown in Fig. 1b, the two gRNAs of pair #3 were aimed approximately halfway between the SANT domain and the STOP codon. As for the representation of the SANT domain in the cartoon shown in Fig. 1a, we purposefully placed labels above the EAF1 contour to indicate only approximate positions of the SANT and HSA domains since the idea behind the drawing was only to illustrate the scaffold function of EAF1 and EPL1 and not the shape of the NuA4 complex. According to published cryo-EM structures of the yeast NuA4 complex, picNuA4 and TRA1 contact EAF1 close to HSA and SANT domain, respectively (Setiaputra et al., Mol Cell Biol. 2018, Wang et al., Nature Commun. 2018), which in our view provides sufficient support for positioning of the labels in the cartoon. Since we were not

interested in dissecting the role of individual domains of EAF1, we did not determine if truncation of the protein at position targeted by the gRNA pair #3 or loss of the SANT domain could result in separation of AtTRA1 from NuA4.

Line 142: the "prediction" that Nua4 loss-of-function mutations would lead to Arabidopsis lethality as in yeast species is farfetched, as most chromatin-based or epigenetic processes known to be essential in metazoan species can actually be knocked out in plants without triggering complete sterility or lethality during development. The mentions of Nua4 being specific in that way therefore appears overstated. For the precise case of transcription-coupled chromatin machineries, see Grasser et al, 2020 doi: 10.1016/j.bbasm.2020.194613.

This has been corrected.

Line 175: "of most, but not all" is quite imprecise. Is it implied a cooperation for chloroplast and less so for cell/organ size?

Yes, we meant the different degree of influence of both mutants on chloroplast development and cell size. But as such detailing would repeat the previous sentence in the paragraph, this has been rephrased.

Line 182 – 183: The aggravation of the phenotype in the quadruple mutant is not aggravated in "all phenotypic assays" as explained in the next sentence, please rephrase.

This has been corrected.

Line 186 – 187: In none of the assays Nua4 and GLK mutants have opposite effect. The case of the a/b chlorophyll ratio is a suppression of the glk phenotype by the epl1 mutation, it is not an opposite effect as the epl1 alone has no phenotype in that assay. Please rephrase.

We agree, we have changed accordingly.

Line 295 – 298: A reduction of the strength of nucleosome–DNA interaction by H4 acetylation is not the only possible explanation for an increase in H3 occupancy in the mutant, H4 acetylation could destabilize nucleosomes or change their positioning or facilitate its eviction by remodelers. Plus, it could also result from H2A.Z hypoacetylation in the mutant and not H4 hypoacetylation. Also, the effect on H3 is along the whole gene body while H4K5ac is restricted mainly to the +1 nucleosome, indicating that this is not the sole explanation. This should be rephrased in a more hypothetical way or reference articles showing the effect of H4 acetylation on nucleosome–DNA interaction and nucleosome occupancy.

We agree with the reviewer that our interpretation is one of many. The sentence was changed accordingly, taking into account also new observations for individual groups of genes.

Line 322: rich instead of reach

Corrected.

Line 321 – 322: "the level of H4K5ac drops below a certain "threshold" level for genes downregulated more rapidly than for the H4K5ac-reach genes." While I understand the arguments about the threshold that might explain the downregulation of genes in the mutant, I found the wording "drops more rapidly" confusing, given that authors do not examine a temporal process but compare steady state levels in different plant lines.

In the revised manuscript this part is rewritten to reflect new observations and this expression is no longer present.

Line 344: I find the wording "Nua4-dependent genes" a bit confusing because what is done here is selecting, among the Nua4-dependent genes for H4K5ac, the genes that are impacted in term of transcription by Nua4 presumably because they don't have a H3K9ac "back-up" system.

While we agree with the reviewer on this point, we have not found a better term for this gene group that is sufficiently succinct.

Line 354: rephrase "has been adopted to".

This has been rephrased.

Figure 6D and related text: This panel is not very informative as the fact that "Nua4 dependent genes have very high level of H4K5ac at the +1 nucleosome, differing from the broader group of all genes that were downregulated in Atep1-2" might only reflect that those genes were chosen in the first place for having a LogFC of H4K5ac < -2, so highly marked genes were initially pre-selected.

We agree with the reviewer. In the revised manuscript we focused on H2A.Z and H2A.Zac in NuA4-dependent genes.

Figure 6E and related text: It is not mentioned where the H2A.Z ChIP-seq data were retrieved.

The new ChIP-seq data is now presented.

Line 373: About the statement " These results clearly demonstrate that the recruitment of TFIID via TAF1 bromodomains is not the primary mode of action of H4ac at NuA4-dependent loci in plants": what is the evidence for a conservation between animals and plants of the interaction between TFIID and H4ac by the bromodomain of TAF1? I would down-tune the statement and condition the conclusion to future studies examining that predicted functional conservation.

We rephrased the conclusion as suggested.

Line 395 – 397: The higher level of H2A.Z in the mutant observed by ChIP-qPCR in figure 6F is neither discussed here nor in the discussion. Can we envision that more free SWC4 is available in *Atepl1* due to the disruption of *Nua4*, and that could favor the formation of the SWR1 complex and so favor H2A.Z incorporation into chromatin?

Following the genome-wide analyzes of H2A.Z in the revised manuscript, the preliminary conclusion about H2A.Z enrichment in the *Atepl1-2* mutant has been revised. We sincerely apologize for our earlier mistake which was probably due to the use of too gentle washes and poor antibody. Currently, we have made every effort to ensure that the new data on this subject is absolutely certain.

Line 425 – 426: The present manuscript does not include any mention of a reproductive phenotype for *Ateaf1-1*. Even though this is out of the main scope, essential information on the *Nua4* mutant plants life cycle is required.

We thank the reviewer for this suggestion. Actually, Fig. S3c documented the complete sterility of the mutants and now we referred to this also in the text of the manuscript.

Line 490 – 492: About the sentence "To the best of our knowledge, similar photosynthesis-related phenotypes are not observed when any of the chromatin factors involved in transcription control characterized to date are inactivated in plants." This clearly appears as an over-statement. I acknowledge that a few of them have been described, but the authors cannot ignore and, for the sake of acknowledging prior work, they should cite at least 3 other Arabidopsis chromatin factors involved in transcription control influencing plant adaptation to light:
1) the histone acetyltransferase GCN5 (see for example Benhamed et al 2006 Plant Cell doi: 10.1105/tpc.106.043489) or a recent review in Grasser et al BBA 2020 doi: 10.1016/j.bbagrm.2020.194613);
2) HUB1-HUB2 histone H2B ubiquitin ligase that shares with *Nua4* mutant plant some defects in pigment biogenesis and misregulation of numerous photosynthetic genes (see Bourbousse et al. 2012 PLoS Genet. doi:10.1371/journal.pgen.1002825);
3) SDG8 histone methyltransferase with similarly established global effects on photosynthetic genes (see Li et al. Genome Biology 2015 doi 10.1186/s13059-015-0640-2).

In the excerpt cited by the reviewer, we wanted to state that such a strong phenotype related to the disturbance of chloroplast development was not described in other chromatin mutants. We therefore apologize for this lack of precision. In the new version of the manuscript, we briefly refer to the works mentioned by the reviewer, however this fragment was now placed in the introduction.

Line 497 – 499: Why not look at the list of plastid and mitochondrial genes to see their enrichment in H4K5ac, H2A.Zac and misregulation in *epl1* to see if they are particularly affected?

We thank for this suggestion: as mentioned above, the more in-depth characterization of these gene groups has been now included and presented in Fig. S13.

Reviewers' Comments:

Reviewer #1:

Remarks to the Author:

Re-revision comments for "Nuclear-encoded photosynthesis genes are specifically controlled by the NuA4 complex and H2A.Z acetylation" by Bieluszewski et al.

I am very pleased with the revisions of the manuscript "Nuclear-encoded photosynthesis genes are specifically controlled by the NuA4 complex and H2A.Z acetylation" by Bieluszewski et al. The authors have satisfactorily addressed all concerns such as the comparison of RNA-seq performed side-by-side for all the mutants and most importantly generated H2AZ and H2AZ.ac genome wide in WT and Atepl1-1 mutants. They provide a thorough analysis of H2AZ changes in relation to changes in gene expression and histone acetylation. The authors conclude that NuA4 is required for H2AZ deposition and acetylation. Interestingly, they also provide insights into why certain genes become upregulated or downregulated upon a loss of H2AZ, and this may be due to the deposition of histone H3 acetylation on certain genes by the SAGA complex. The authors also relate the changes of H2AZ to those observed in the *pie1* mutants of the SWR1 complex. It is unfortunate that the higher order mutants between *Nua4* mutants, *arp6* and *mbd9* were unable to be recovered due to lethality. Overall this is a very thorough biochemical, genetic, and genomic characterization of the *Nua4* complex in plants, which up until now has been very poorly described in *Arabidopsis*. This study also provides interesting insights in the H2AZ biology and therefore would be of great interest to the general community.

Minor issues

Figure 7e legend seems to be a copy and paste error since this figure is not a boxplot.

The model is a little difficult to follow, for instance, you could change three arrows for transcription in the legend to one arrow, since only one arrow is used in the model. Also, maybe consider adding SAGA complex for the H3K9ac deposition?

Reviewer #2:

Remarks to the Author:

In the revised version of the manuscript, the authors have added a significant body of additional data, evidencing the major conclusions. In particular, transcriptome analysis of *Atepl1-2*, *Ateaf1-1* and double *glk* mutants has been carried out, supporting the similarities and/or differences of the *NuA4* mutants or *NuA4* and *glk* mutants, resp. The ChIP-seq analysis of H4K5ac, H2A.Zac and H3K9ac has brought convincing support for the involvement of *NuA4* in H2A.Z deposition and acetylation, and has also supported the specific direct (primary?) effect of *NuA4* on photosynthesis-related genes.

All the points of the reviewers have been carefully considered and where possible, addressed. The suggested experiments have been carried out. For successfully experiments, all results are included in the revised version of the manuscript. Where technically unsuccessful, evidence of being attempted is provided in the rebuttal. The text of the manuscript has been revised accordingly, addressing issues that concerned formulations, phrasing and content.

In conclusion, the manuscript has been thoroughly revised according to the issues raised by the reviewers in a way that strengthens the final conclusions and I would support its publication in the present form.

Reviewer #3:

Remarks to the Author:

The manuscript by Bieluszewski and colleagues has been largely improved according to the 3 reviewers' suggestions and the points previously raised have been addressed very seriously. Following my requests identification that Nua4 is required on a genome-wide scale for H2A.Z incorporation at +1 nucleosomes and gene body, and for H2A.Z acetylation, constitute a strong improvement. I have a few new concerns about the analysis of the series of ChIP-seq introduced in the revised version (H2A.Z and H2A.Zac), which come short in linking differential marking with differential expression (as expressed in the response to the authors). In my opinion the manuscript should be accepted for publication following minor revision of the issues detailed below:

Main issue

Line 319/Fig. 6: Analysis of the new ChIP-seq provided (H2A.Z and H2A.Zac) that comes short in linking differential marking with differential expression:

- Gains and losses in Fig. 6 are hard to see because WT and epl1-2 are not on the same plots.
 - o "H4/H2A.Zac loss shapes the Atepl1-2 transcriptome" is hard to interpret because it seems that most marked genes lose those 2 marks, while 3474 genes are down and 1235 genes are upregulated. A more refined analysis would be more informative to understand what makes hypomarked genes more susceptible to other ones in term of expression outputs (e.g., how many genes are marked by H4K5ac/H2A.Zac and how much do they overlap? What are the genes losing/gaining each or both marks in the mutant? (Scrolling over H2A.Z and H2A.Zac tracks provided on GEO I have seen many genes gaining the marks, probably corresponding to the dots above the diagonal in figure 5b and that are not analyzed in the manuscript); What are the transcript levels of these differentially marked genes in the WT and the mutant? Are downregulated genes more marked than other genes by one or both marked in the WT to start with?
- Second, the gain of H3ac is not directly mediated by Nua4 loss of function, and so this could on the contrary be a consequence of the transcriptomic changes induced by Nua4, as stated by the authors just below lines 340-343, so these cases totally contradict the title. This should be considered in the text.

Text and figure corrections

Abstract: "Many chloroplast-related genes depend on NuA4": replace by "Expression of many chloroplast-related genes depends on NuA4"

Figure S4B: There must be an issue for the two samples at 56h. The hypocotyl of dark grown seedlings should stop elongating upon light exposure, which is no consistent with the graph data.

Figure S9/S10 & Figure 5: The diagenode antibody C15410173 was raised against acetylated human H2A.Z at positions K4, K7 and K11. As shown in Fig. S9 and written in the legend, this region is not conserved between Arabidopsis and human. When first using this antibody, the authors should cite Crevillen et al. New Phytologist 2019 (reference 17) and their own figure S8C that show that this antibody recognized acetylated Arabidopsis HTA9 (Line 294/Legend Figure 5).

Line 303: "strong" is overstated, and not easily supported by a visual inspection of Fig5b. Please correct or provide a quantitative assessment (e.g., r calculation) for each scatter plot.

Line 311 to 318: This is sustained only by a snapshot on 2 example genes in 5C and is largely repeated in the next section with Fig. S11. These lines should be removed.

Figure S13: a and b labels are missing.

Figure 9: Is the specific drawing of H4/H2A.Zac at the +2 nucleosome of stress genes a mistake? How do we know acetylation on those genes are later removed versus not deposited at all? What is the evidence that GLK genes get H2A.Z and H3K9/H2A.Z/H4ac, and that deposition of these marks does not depends on Nua4 as shown in the scheme? Or maybe it depends on Nua4 but GLKs are still able to promote their expression despite this?

Methods: Only 2 biological replicates of H2A.Z and H2A.Zac profiling have been provided and not 3 as stated in the methods, please correct.

REVIEWER COMMENTS

Reviewer #1 (Remarks to the Author):

Re-revision comments for "Nuclear-encoded photosynthesis genes are specifically controlled by the NuA4 complex and H2A.Z acetylation" by Bieluszewski et al.

I am very pleased with the revisions of the manuscript "Nuclear-encoded photosynthesis genes are specifically controlled by the NuA4 complex and H2A.Z acetylation" by Bieluszewski et al. The authors have satisfactorily addressed all concerns such as the comparison of RNA-seq performed side-by-side for all the mutants and most importantly generated H2AZ and H2AZ.ac genome wide in WT and Atepl1-1 mutants. They provide a thorough analysis of H2AZ changes in relation to changes in gene expression and histone acetylation. The authors conclude that NuA4 is required for H2AZ deposition and acetylation. Interestingly, they also provide insights into why certain genes become upregulated or downregulated upon a loss of H2AZ, and this may be due to the deposition of histone H3 acetylation on certain genes by the SAGA complex. The authors also relate the changes of H2AZ to those observed in the pie1 mutants of the SWR1 complex. It is unfortunate that the higher order mutants between Nua4

mutants, arp6 and mbd9 were unable to be recovered due to lethality.

Overall this is a very thorough biochemical, genetic, and genomic characterization of the Nua4 complex in plants, which up until now has been very poorly described in Arabidopsis. This study also provides interesting insights in the H2AZ biology and therefore would be of great interest to the general community.

Minor issues

Figure 7e legend seems to be a copy and paste error since this figure is not a boxplot.

We apologize for this oversight. In the new version of the manuscript, we have corrected the caption accordingly.

The model is a little difficult to follow, for instance, you could change three arrows for transcription in the legend to one arrow, since only one arrow is used in the model. Also, maybe consider adding SAGA complex for the H3K9ac deposition?

In the revised version we simplified the model and we tried to explain it better. The three arrows represented different levels of gene transcription, but there was an issue with this figure conversion, so they looked the same on the model. We will make sure that in the final version of the paper the arrows will be correctly presented. We also considered adding SAGA complex to the model, but felt that this would make the model even more difficult to follow. Moreover, we do not directly investigate SAGA in this research.

Reviewer #2 (Remarks to the Author):

In the revised version of the manuscript, the authors have added a significant body of additional data, evidencing the major conclusions. In particular, transcriptome analysis of Atepl1-2, Ateaf1-1 and double glk mutants has been carried out, supporting the similarities and/or differences of the NuA4 mutants or NuA4 and glk mutants,

resp. The ChIP-seq analysis of H4K5ac, H2A.Zac and H3K9ac has brought convincing support for the involvement of NuA4 in H2A.Z deposition and acetylation, and has also supported the specific direct (primary?) effect of NuA4 on photosynthesis-related genes.

All the points of the reviewers have been carefully considered and where possible, addressed. The suggested experiments have been carried out. For successfully experiments, all results are included in the revised version of the manuscript. Where technically unsuccessful, evidence of being attempted is provided in the rebuttal. The text of the manuscript has been revised accordingly, addressing issues that concerned formulations, phrasing and content.

In conclusion, the manuscript has been thoroughly revised according to the issues raised by the reviewers in a way that strengthens the final conclusions and I would support its publication in the present form.

We thank the Reviewer for appreciating our work on the manuscript revision.

Reviewer #3 (Remarks to the Author):

The manuscript by Bieluszewski and colleagues has been largely improved according to the 3 reviewers' suggestions and the points previously raised have been addressed very seriously. Following my requests identification that Nua4 is required on a genome-wide scale for H2A.Z incorporation at +1 nucleosomes and gene body, and for H2A.Z acetylation, constitute a strong improvement. I have a few new concerns about the analysis of the series of ChIP-seq introduced in the revised version (H2A.Z and H2A.Zac), which come short in linking differential marking with differential expression (as expressed in the response to the authors). In my opinion the manuscript should be accepted for publication following minor revision of the issues detailed below:

Main issue

Line 319/Fig. 6: Analysis of the new ChIP-seq provided (H2A.Z and H2A.Zac) that comes short in linking differential marking with differential expression:

- Gains and losses in Fig. 6 are hard to see because WT and *epl1-2* are not on the same plots.

In Fig. 6, we decided to compare the profiles of the analyzed chromatin modifications for individual gene groups separately for WT and for *Atepl1-2*. We found these differences between gene groups to be more interesting than the general decreases observed in *Atepl1-2*. However, we draw your attention to the fact that Fig. S11 compares the level of each modification in WT and *Atepl1-2* in the same plots, i.e. in the way that the Reviewer wanted.

o "H4/H2A.Zac loss shapes the *Atepl1-2* transcriptome" is hard to interpret because it seems that most marked genes lose those 2 marks, while 3474 genes are down and 1235 genes are upregulated.

In the new version of the manuscript, thanks to the further analysis suggested by the reviewer, we largely answer these questions (see below).

A more refined analysis would be more informative to understand what makes hypomarked genes more susceptible to other ones in term of expression outputs (e.g., how many genes are marked by H4K5ac/H2A.Zac and how much do they overlap? What are the genes losing/gaining each or both marks in the mutant? (Scrolling over H2A.Z and H2A.Zac tracks provided on GEO I have seen many genes gaining the marks, probably corresponding to the dots above the diagonal in figure 5b and that are not analyzed in the manuscript); What are the transcript levels of these differentially marked genes in the WT and the mutant? Are downregulated genes more marked than other genes by one or both marked in the WT to start with?

We thank the reviewer for provoking us to a more in-depth analysis of ChIP-seq data in the context of changes in gene expression. While attempts to directly correlate these two data sets through single-gene comparisons were unsuccessful, the group analysis was much more fruitful. We split all genes that change expression in *Atepl1* into percentiles depending on expression fold change. For these percentiles, we computed the mean values of the ChIP-seq signal at 500 bp downstream of the TSS. This analysis showed two important things: 1) in the *Atepl1* mutant, only those genes that have low basal levels of all acetylations (new Fig. 6b, new Fig. S12) are dysregulated, and 2) the degree of change in expression in *Atepl1* positively correlates with the change of acetylation level (new Fig. 6c). We believe that this largely answers the reviewer's questions.

In addition, we conducted other analyses suggested by the reviewer. In new Figs 5d and 5e we have shown to what extent the different types of chromatin modifications correlate with each other. This analysis showed that the acetylations catalysed by NuA4, i.e. H4K5ac and H2A.Zac, correlate strongly with each other, but also with H3K9ac.

We also examined genes that lose only one of the NuA4-dependent acetylations. Only 36 genes lose H2A.Zac while retaining H4K5ac. The levels of H4K5ac and H3K9ac in these genes, however, were very low, suggesting that these observations were false-positive. Many more genes, i.e. 1029, lose H4K5ac without changing the level of H2A.Zac. However, as many as 671 of these genes showed no change in expression level in the mutant, while 56 genes were down- (GO insignificant) and 302 (GO mainly biotic stress) were upregulated. As suggested by the reviewer, we also checked the genes that showed an increase in H4K5ac or H2A.Zac in *Atepl1* in relation to WT (above the diagonal lines on Fig. 5b). For H4K5ac, there are 110 such genes and the vast majority of them (87) show no changes in expression in *Atepl1* (the remaining 23 genes are upregulated). In turn, 1102 genes have

more H2A.Zac in *Atep1* than in WT. Interestingly, they contain all the 1029 genes that lose H4K5ac without changing their H2A.Zac levels, so it can be said that they are largely the same group of genes. Based on these analyzes, the conclusion is that the transcriptomic response to the decrease in NuA4-dependent acetylation depends mainly on the basal level of all acetylations in the WT: only those genes that have a low level of all acetylation respond to NuA4 loss. As this conclusion already appears in the manuscript (based on Fig. 6b and Fig. S12), and the above-mentioned analyzes do not lead to other interesting conclusions, we decided not to include them in the manuscript.

- Second, the gain of H3ac is not directly mediated by Nua4 loss of function, and so this could on the contrary be a consequence of the transcriptomic changes induced by Nua4, as stated by the authors just below lines 340343, so these cases totally contradict the title. This should be considered in the text.

The reviewer is right that the H3ac level changes are not a direct result of the NuA4 loss. As suggested by the reviewer, they may be a consequence of transcriptomic changes, although they may also appear as an indirect response to the lack of H4ac and H2A.Zac, and themselves affect gene transcription. Therefore, we changed the title of this subsection replacing "Gain of H3ac and loss of H4/H2A.Zac shape the *Atep1-2* transcriptome" with the title "Loss of H4/H2A.Zac and gain of H3ac correlate with the *Atep1-2* transcriptome changes".

Text and figure corrections

Abstract: "Many chloroplast-related genes depend on NuA4": replace by "Expression of many chloroplast-related genes depends on NuA4"

We corrected the abstract as requested.

Figure S4B: There must be an issue for the two samples at 56h. The hypocotyl of dark grown seedlings should stop elongating upon light exposure, which is no consistent with the graph data.

Indeed, there must have been some error in the measurements. Therefore, we removed the data point at 56 hours.

Figure S9/S10 & Figure 5: The diagenode antibody C15410173 was raised against acetylated human H2A.Z at positions K4, K7 and K11. As shown in Fig. S9 and written in the legend, this region is not conserved between Arabidopsis and human. When first using this antibody, the authors should cite Crevillen et al. New Phytologist 2019 (reference 17) and their own figure S8C that show that this antibody recognized acetylated Arabidopsis HTA9 (Line 294/Legend Figure 5).

In the revised ms, we introduced information on the use of this antibody by Crevillen et al. 2019, where we first mention ChIP-seq for H2A.Zac in the text and also in the Fig. 5 caption.

Line 303: "strong" is overstated, and not easily supported by a visual inspection of Fig5b. Please correct or provide a quantitative assessment (e.g., r calculation) for each scatter plot.

The text was corrected accordingly.

Line 311 to 318: This is sustained only by a snapshot on 2 example genes in 5C and is largely repeated in the next section with Fig. S11. These lines should be removed.

We agree with the reviewer and thus we removed this part as requested.

Figure S13: a and b labels are missing.

We apologize for this mistake, which we fixed in the new ms version.

Figure 9:

Is the specific drawing of H4/H2A.Zac at the +2 nucleosome of stress genes a mistake?

This was intended to show gene body H2A.Z deposition and was explained in the revised caption of this figure.

How do we know acetylation on those genes are later removed versus not deposited at all?

This is a good point, we explained in the revised figure caption that it is currently not known.

What is the evidence that GLK genes get H2A.Z and H3K9/H2A.Z/H4ac, and that deposition of these marks does not depends on Nua4 as shown in the scheme? Or maybe it depends on Nua4 but GLKs are still able to promote their expression despite this?

This was a misunderstanding, as we wanted to show cooperation between nucleosome acetylation and TF activity. This was simplified in the new version of the model.

Methods: Only 2 biological replicates of H2A.Z and H2A.Zac profiling have been provided and not 3 as stated in the methods, please correct.

Corrected as requested.

Reviewers' Comments:

Reviewer #3:

Remarks to the Author:

The manuscript by Bieluszewski and colleagues has been largely improved according to the 3 reviewers' suggestions and the points previously raised have been addressed very seriously. Following my requests identification that Nua4 is required on a genome-wide scale for H2A.Z incorporation at +1 nucleosomes and gene body, and for H2A.Z acetylation, constitute a strong improvement. I have a few new concerns about the analysis of the series of ChIP-seq introduced in the revised version (H2A.Z and H2A.Zac), which come short in linking differential marking with differential expression (as expressed in the response to the authors). In my opinion the manuscript should be accepted for publication following minor revision of the issues detailed below:

Main issue

Line 319/Fig. 6: Analysis of the new ChIP-seq provided (H2A.Z and H2A.Zac) that comes short in linking differential marking with differential expression:

- Gains and losses in Fig. 6 are hard to see because WT and epl1-2 are not on the same plots.
 - o "H4/H2A.Zac loss shapes the Atepl1-2 transcriptome" is hard to interpret because it seems that most marked genes lose those 2 marks, while 3474 genes are down and 1235 genes are upregulated. A more refined analysis would be more informative to understand what makes hypomarked genes more susceptible to other ones in term of expression outputs (e.g., how many genes are marked by H4K5ac/H2A.Zac and how much do they overlap? What are the genes losing/gaining each or both marks in the mutant? (Scrolling over H2A.Z and H2A.Zac tracks provided on GEO I have seen many genes gaining the marks, probably corresponding to the dots above the diagonal in figure 5b and that are not analyzed in the manuscript); What are the transcript levels of these differentially marked genes in the WT and the mutant? Are downregulated genes more marked than other genes by one or both marked in the WT to start with?
- Second, the gain of H3ac is not directly mediated by Nua4 loss of function, and so this could on the contrary be a consequence of the transcriptomic changes induced by Nua4, as stated by the authors just below lines 340-343, so these cases totally contradict the title. This should be considered in the text.

Text and figure corrections

Abstract: "Many chloroplast-related genes depend on NuA4": replace by "Expression of many chloroplast-related genes depends on NuA4"

Figure S4B: There must be an issue for the two samples at 56h. The hypocotyl of dark grown seedlings should stop elongating upon light exposure, which is no consistent with the graph data.

Figure S9/S10 & Figure 5: The diagenode antibody C15410173 was raised against acetylated human H2A.Z at positions K4, K7 and K11. As shown in Fig. S9 and written in the legend, this region is not conserved between Arabidopsis and human. When first using this antibody, the authors should cite Crevillen et al. New Phytologist 2019 (reference 17) and their own figure S8C that show that this antibody recognized acetylated Arabidopsis HTA9 (Line 294/Legend Figure 5).

Line 303: "strong" is overstated, and not easily supported by a visual inspection of Fig5b. Please correct or provide a quantitative assessment (e.g., r calculation) for each scatter plot.

Line 311 to 318: This is sustained only by a snapshot on 2 example genes in 5C and is largely repeated in the next section with Fig. S11. These lines should be removed.

Figure S13: a and b labels are missing.

Figure 9: Is the specific drawing of H4/H2A.Zac at the +2 nucleosome of stress genes a mistake? How

do we know acetylation on those genes are later removed versus not deposited at all? What is the evidence that GLK genes get H2A.Z and H3K9/H2A.Z/H4ac, and that deposition of these marks does not depend on Nua4 as shown in the scheme? Or maybe it depends on Nua4 but GLKs are still able to promote their expression despite this?

Methods: Only 2 biological replicates of H2A.Z and H2A.Zac profiling have been provided and not 3 as stated in the methods, please correct.